# Long-read sequencing for 29 immune cell subsets reveals disease-linked isoforms

Jun Inamo [1,2], Akari Suzuki[3], Mahoko Takahashi Ueda [1], Kensuke Yamaguchi[1,3,4], Hiroshi Nishida [5], Katsuya Suzuki[2], Yuko Kaneko[2], Tsutomu Takeuchi [2,6], Hiroaki Hatano[7], Kazuyoshi Ishigaki [7], Yasushi Ishihama [5,8], Kazuhiko Yamamoto[3] & Yuta Kochi [1,3] ✉

Alternative splicing events are a major causal mechanism for complex traits, but they have been understudied due to the limitation of short-read sequencing. Here, we generate a full-length isoform annotation of human immune cells from an individual by long-read sequencing for 29 cell subsets. This contains a number of unannotated transcripts and isoforms such as a read-through transcript of *TOMM40-APOE* in the Alzheimer's disease locus. We profile characteristics of isoforms and show that repetitive elements significantly explain the diversity of unannotated isoforms, providing insight into the human genome evolution. In addition, some of the isoforms are expressed in a cell-type specific manner, whose alternative 3'-UTRs usage contributes to their specificity. Further, we identify disease-associated isoforms by isoform switch analysis and by integration of several quantitative trait loci analyses with genome-wide association study data. Our findings will promote the elucidation of the mechanism of complex diseases via alternative splicing.

Over 90% of human genes undergo alternative splicing, resulting in hundreds of thousands of transcript isoforms[1–3]. Alternative splicing can generate isoforms that differ in coding sequences through mechanisms that include exon skipping, a choice between mutually exclusive exons, the use of alternative splice sites, and intron retention, causing diversity in the open reading frame (ORF) and function of proteins[4]. In addition, different 5'- and 3'-untranslated regions (UTRs) can quantitatively affect cellular functions[5,6]. These splicing events can trigger human diseases, as genetic variants that affect alternative splicing, defined as splicing quantitative trait loci (sQTL), are enriched in the loci discovered by genome-wide association studies (GWAS) for complex diseases including immune-mediated diseases (IMDs)[7–9].

Indeed, in the GTEx project, 23% of the GWAS loci were co-localized with sQTL.

Generally, sQTL can be identified by testing the association of a variant's genotype with the junction read counts of isoforms[10], or alternatively, with the transcript-ratio of isoform expressions[8]. The latter method, also known as trQTL analysis, enables a direct understanding of which isoform expression is altered, that is, what kind of changes in the coding sequences or UTRs occur. However, the accuracy of quantification of junction reads and isoform expressions is susceptible to the credibility of isoform annotation. Therefore, accurate isoform annotation containing full-length sequences, even for those with low expression levels, can reveal novel disease pathogenetic

[1]Department of Genomic Function and Diversity, Medical Research Institute, Tokyo Medical and Dental University (TMDU), Tokyo 113-8510, Japan. [2]Division of Rheumatology, Department of Internal Medicine, Keio University School of Medicine, Tokyo 160-8582, Japan. [3]Laboratory for Autoimmune Diseases, RIKEN Center for Integrative Medical Sciences, Yokohama, Kanagawa 230-0045, Japan. [4]Biomedical Engineering Research Innovation Center, Institute of Biomaterials and Bioengineering, Tokyo Medical and Dental University (TMDU), Tokyo 113-8510, Japan. [5]Department of Molecular Systems Bioanalysis, Graduate School of Pharmaceutical Sciences, Kyoto University, Kyoto 606-8501, Japan. [6]Saitama Medical University, 38 Morohongo, Moroyama, Iruma, Saitama 350-0495, Japan. [7]Laboratory for Human Immunogenetics, RIKEN Center for Integrative Medical Sciences, Yokohama, Kanagawa 230-0045, Japan. [8]Laboratory of Proteomics for Drug Discovery, National Institute of Biomedical Innovation, Health and Nutrition, Ibaraki, Osaka 567-0085, Japan. ✉e-mail: y-kochi.gfd@mri.tmd.ac.jp

mechanisms via alternative splicing. In fact, we have recently demonstrated that some low-expression isoforms that are disease-causing have coding sequences that were incomplete in the GENCODE annotation[11].

The emergence of long-read sequencing that can generate reads of 10,000 bases or more has revolutionized genomic studies; it has improved the mapping accuracy of reads, de novo assembly of genomes, and the detection of structural variations including repetitive elements[12]. It has also advanced the precise evaluation of transcript structures as well as the discovery of novel transcripts that had been missed by short-read sequencing[13–15]. Motivated by these features of long-read sequencing, large-scale projects aiming to reconstruct full-length transcripts using multi-tissue samples are ongoing[16,17]. However, to our knowledge, there have been no studies focusing on immune cell subsets other than whole blood cells or lymphoblastoid cell-lines (LCL). Each of the diverse immune cell subsets has critical and specific functions in response to external stimuli[18,19], the metabolic system[20], and the nervous system[21]. In addition, the importance of cell-type-specific alternative splicing in the immune system is known for a variety of genes, and cell-type-specific profiling of isoforms will help elucidate complicated immune system networks[21–23].

Here, we generated a full-length isoform annotation, which we named the Transcriptomic Resource of Immune Cells using Long-read Sequencing (TRAILS), by sequencing 29 immune cell subsets using long-read sequencing. Through profiling the characteristics of isoforms including inserted transposable elements (TEs) and cell-type specificity, we sought the evolutionary origins of human genome function as well as the human immune system. Furthermore, we showed that TRAILS would bridge the gap between genomic and functional analysis and help elucidate the pathogenesis of IMDs.

## Results
### Overview of the TRAILS
To clarify the transcriptome profiles of immune cells at full-length levels, we isolated 29 immune cell subsets from the peripheral blood cells of a 42-year-old healthy female. cDNA libraries were made from the poly(A) mRNA, PCR amplified, and subjected to long-read sequencing using the MinION Oxford Nanopore Technologies platform (Methods; Fig. 1A; Supplementary Data 1, 2). We identified a total of 159,369 isoforms transcribed from 17,496 genomic loci (Supplementary Fig. 1A−C). As a validation, we additionally sequenced the transcripts of PBMC from a 40-year-old healthy male using the latest PromethION platform (Methods). We validated 85.0% ($n = 6399$) of isoforms expressed in PBMC used in the TRAILS. We also validated 30.6% ($n = 48,757$) of total isoforms in all cell types; of the isoforms validated by this independent data set, 29.6% were full splice match (FSM), 33.4% had a novel combination of known splice sites (NIC), and 28.2% had novel splicing site (NNC) in comparison with GENCODE version 38 (hereafter GENCODE). We made them publicly available, considering both cases where users want more exploratory data (TRAILS) and where they want validated data. From here, downstream analysis was performed using TRAILS to focus on cell type diversity in the landscape of transcriptome. We compared TRAILS with those of GM12878 cell lines obtained by long-read sequencing[24], and 7.8% of our isoforms were matched. The median length of isoforms in our database was 1752 nucleotides (maximum 9,933 nucleotides), which was significantly longer than the isoforms registered in the comprehensive gene annotation of GENCODE (median 929 nucleotides, maximum 347,561 nucleotides) (Wilcoxon test, $p < 0.001$, Fig. 1B). To assess the capability of our dataset in discovering isoforms, we conducted a sensitivity analysis by downsampling reads. Our results revealed that our dataset exhibited sufficient power in detecting transcripts with high expression (Supplementary Fig. 2). However, we observed that increasing the number of reads would enable the identification of

additional transcripts with low expression. At the gene loci level, we found 3006 genomic loci in the TRAILS where transcripts are not annotated in GENCODE (Fig. 1C). Further, we predicted coding potential of all isoforms in both TRAILS and GENCODE using GeneMark-ST[25] and compared the number of them. As a result, we found 129,708 isoforms with sequences not registered in GENCODE (Fig. 1C). The number of isoforms per genomic locus was higher in the TRAILS than in GENCODE, with 44% of the total genomic loci transcribing more than 10 different isoforms (Fig. 1D). A comparison of the percentage of transcriptional support level categories for isoforms common to GENCODE and TRAILS showed that the highest percentage was in the most reliable category (all splice junctions are supported by at least one non-suspect mRNA) defined by GENCODE (Supplementary Fig. 1D). This supports the reliability of isoforms identified by long-read sequencing in our database.

We then examined what classes of alternative splicing occurred in our isoforms compared to those registered in GENCODE (Fig. 1E). In view of splicing junctions, we found that 78% of the isoforms had either (NNC) or (NIC). Regarding the type of splicing events, intron retention, alternative 5′ splice sites, and alternative 3′ splice sites were more common in the TRAILS[26] (Fig. 1F). Since genes with higher expression levels had a greater number of alternatively spliced isoforms (Supplementary Fig. 1E), we corrected the number of alternative isoforms by the expression level of each gene. We then examined the characteristics of genes having the highest and lowest numbers of isoforms (top 5% and bottom 5%, respectively) (Methods). As a result, we found that genes involved in IFN signaling and mitotic spindles were enriched, respectively, suggesting that humans have acquired diverse isoforms related to immunological activity through alternative splicing, while cellular homeostasis was maintained by genes with a limited diversity of isoforms (Fig. 1G).

### Predicted-coding transcripts in the TRAILS
Next, we predicted the coding potential of isoforms and identified 145,523 coding isoforms. To determine whether proteins are actually translated from the predicted ORFs, we referred to the data of ongoing proteome analysis using the LCL and a monocytic leukemia cell line (THP-1) (Nishida et al., manuscript in preparation). To reduce false positives and maximize the identification of novel proteins, we selected 16,190 isoforms expressed in LCL or THP-1 and did not exactly match the amino acid sequences registered in GENCODE. As a result, we confirmed peptides for 276 isoforms (Supplementary Data 3, 4), of which 139 peptides were not also registered in Swiss-Prot[27].

Then, of predicted-coding isoforms that differ from coding sequences (CDS) in GENCODE (Fig. 1C), we focused on 1365 (40%) loci encoded so-called read-through isoforms, in which the mRNA extended through the conventional polyadenylation signal (PAS) but stopped at the PAS of adjacent genes or genomic loci (Fig. 2A). Although read-through isoforms are known to be transcribed in particular situations in cells, such as in malignancy and infection[28,29], many read-through isoforms, such as *TOMM40_APOE* (Fig. 2B), were also transcribed in cells under normal physiological conditions. Interestingly, the read-through isoform transcribed from the *TOMM40_APOE* locus was predicted to harbor conserved domains and conformational structure characteristics to both *TOMM40* and *APOE* (Fig. 2B, C). Using an independent PBMC sample from a male individual (Methods), we verified the expression of 74.2% (291 out of a total of 392) read-through transcripts expressed in female PBMC used for our database, including the transcript of *TOMM40_APOE*. To rule out the possibility of artifacts in the sequencing, we further investigated their expressions using direct RNA sequencing data obtained from the LCL (GM12878)[24]. Among the read-through loci identified in 29 cell types ($n = 1365$) in our TRAILS, we validated that 437 loci were truly expressed in LCL, including those from *TOMM40_APOE* and *IFNAR2_IL10RB* (Supplementary Fig. 3).

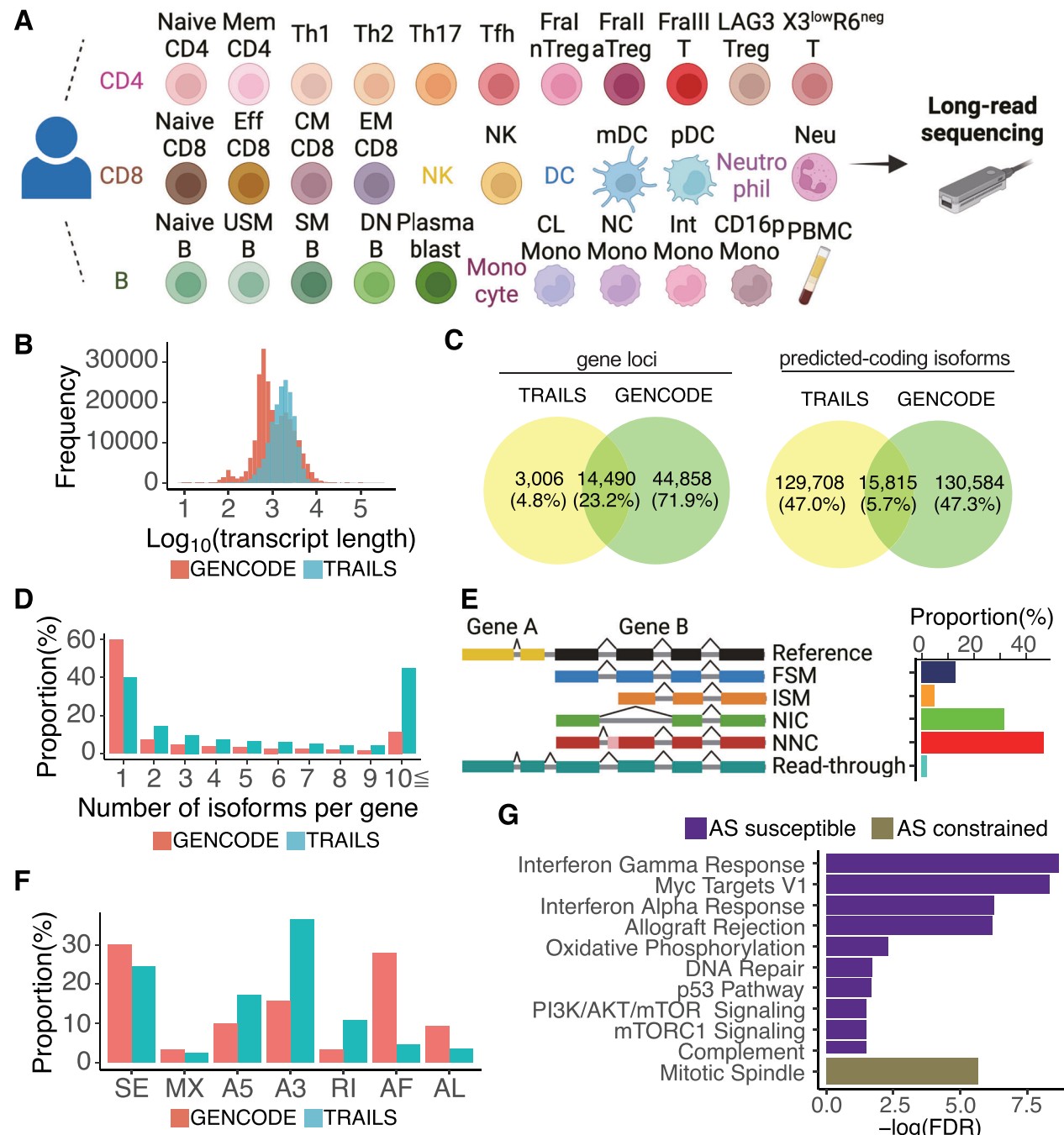

**Fig. 1 | Overview of the TRAILS. A** Summary of the cell subsets included in long-read sequencing in this study. A full description of the subset names and gating strategy is provided in Supplementary Data 1. Created with BioRender.com. **B** The distribution of transcript length. **C** Overlap of gene loci (left) and predicted-coding isoforms (right) between the TRAILS and GENCODE. **D** The proportion of the number of alternatively spliced isoforms per genomic locus. **E** The proportion of structural categories of isoforms in the TRAILS. FSM (full splice match), meaning the reference and query isoform have the same number of exons, and each internal junction agrees; ISM (incomplete splice match), meaning the query isoform has fewer 5′ exons than the reference, but each internal junction agrees; NIC (novel in catalog), meaning the query isoform does not have an FSM or ISM match, but is using a combination of known donor/acceptor sites; NNC (novel not in catalog), meaning the query isoform does not have an FSM or ISM match, and has at least one donor or acceptor site that is not annotated. **F** The proportion of splicing events of isoforms in the TRAILS. SE, skipping exon; MX, mutually exclusive exon; A5, alternative 5′ splice site; A3, alternative 3′ splice site; RI, retained intron; AF, alternative first exon; AL, alternative last exon. **G** Pathway analysis using genes that have the top 5% (purple, alternative splicing (AS) susceptible genes) and bottom 5% (gold, AS constrained genes) of the number of alternatively spliced isoforms per gene after correction by the expression level.

In addition, 1022 loci annotated as long non-coding RNAs (lncRNAs) in GENCODE were predicted as potentially coding genes by GeneMark-ST. Although lncRNAs are defined as over 200 nucleotides in length and do not code for a peptide or protein[30], previous studies have shown that a fraction of putative small ORFs within lncRNAs are translated[31]. Notably, the genomic regions of predicted ORFs of these lncRNAs were more conserved compared to those of the UTRs and introns (Fig. 2D), supporting their coding potential.

Next, we examined 529 loci encoding potentially predicted-coding genes, which have no overlap, even partial, with a gene locus

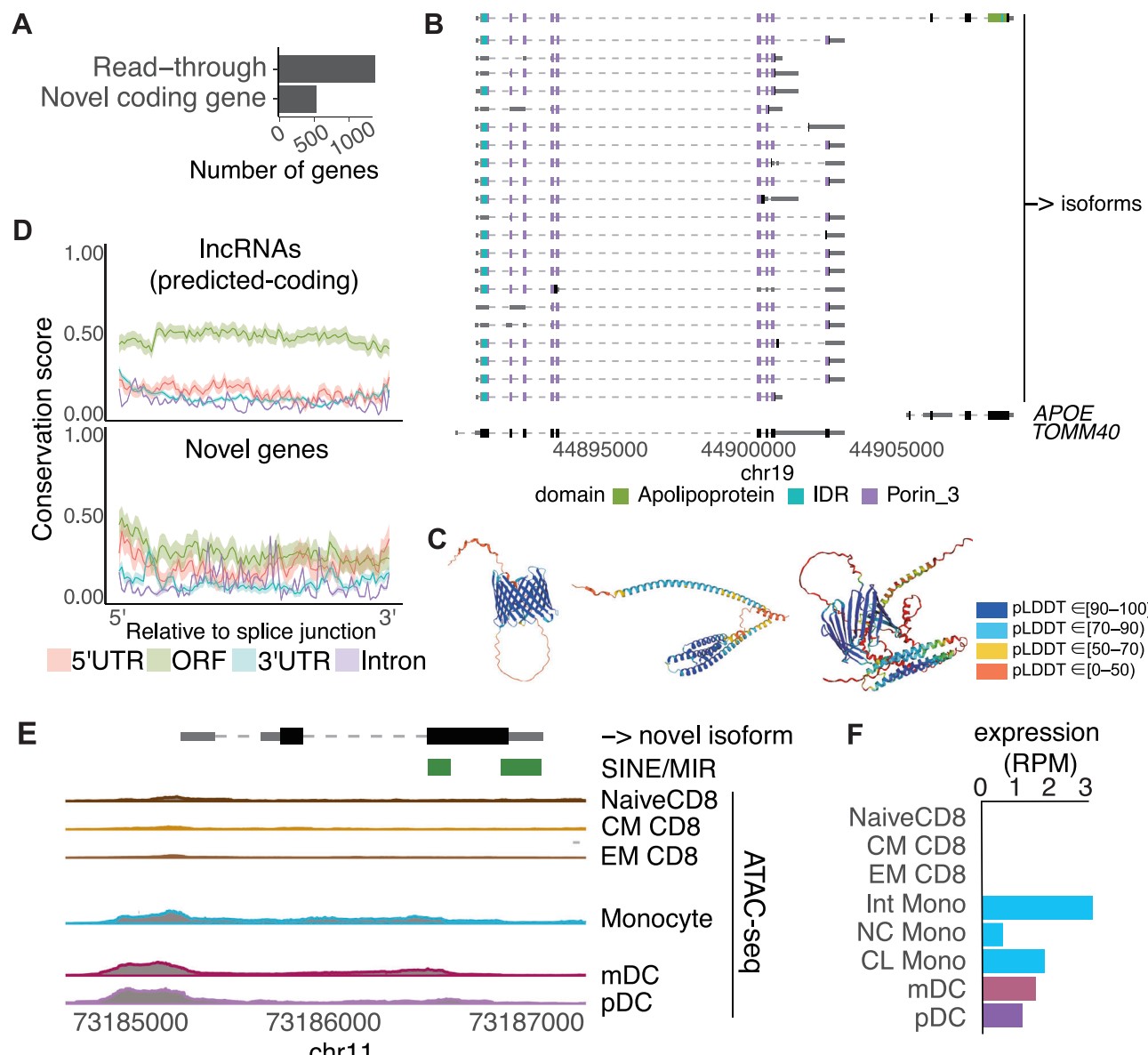

**Fig. 2 | Novel predicted-coding genes identified in TRAILS. A** The number of predicted-coding read-through transcripts and transcripts which have no overlap with a gene locus in GENCODE. **B** Example of a read-through isoform (top) transcribed from the *TOMM40* and *APOE* locus. The arrow indicates the direction of transcription in the genome. The collapsed gene structure registered in GENCODE is shown at the bottom. **C** Protein 3D structures of TOMM40 (left), APOE (center), and the read-though isoform (right) predicted using AlphaFold2. pLDDT is a per-residue estimate of its confidence on a scale from 0 to 100. Regions with pLDDT > 90, between 70 and 90, between 50 and 70, and <50 are expected to be high accuracy, well (a generally good backbone prediction), low confidence, and a reasonably strong predictor of disorder, respectively. **D** Conservation score of predicted-coding transcripts registered as lncRNAs in GENCODE (top) and predicted-coding transcripts which have no overlap with a genomic locus in GEN-CODE (bottom) according to their region. Shading indicates mean ±1.96 standard error. **E** Example of an isoform from a novel gene locus. Peaks of ATAC-seq derived from relevant non-stimulated immune cell subsets around this locus are shown. The arrow indicates the direction of transcription in the genome. **F** The expression of the novel isoform in each cell subset is shown. The expression value was normalized by reads per million (RPM).

registered in GENCODE (Fig. 2A). Notably, their predicted ORFs were highly conserved (Fig. 2D). Of these transcripts, 92% (1891 transcripts) were expressed in multiple cell types and all of them had multi-exons. We examined whether these transcripts were registered in the Comprehensive Human Expressed SequenceS (CHESS) database[32]. Most of the transcripts were found in CHESS, but the remaining 47 transcripts from mutually exclusive loci did not overlap any annotated genes in CHESS. For example, one transcript from chromosome 11 had a predicted 448 nucleotide ORF (Fig. 2E), and notably, we found an open chromatin region from the TSS to intronic regions in subsets of monocytes and dendritic cells using datasets of the assay for transposase-accessible chromatin sequencing (ATAC-seq)[32]. This

corresponded to our expression profile, in which it is only expressed in these subsets in the TRAILS (Fig. 2F). Because the ORF and 3'-UTR of this isoform contained sequences derived from short interspersed elements (SINE)/MIR, the insertion of these TEs may have made it difficult to map the reads by short-read sequencing. Further experimental verification is needed to confirm whether these potentially novel genes are truly encoding genes or just noise, such as pervasive transcription from open chromatin regions[33].

**Transposable elements inserted in isoforms**
Motivated by the finding of the potentially novel transcript above, we investigated the contribution of TEs in constituting novel genes (i.e.,

those not registered in GENCODE). TEs are major repetitive elements and make up approximately half of the human genome[34]. TEs can be subdivided into four major categories: (i) DNA transposons; (ii) long terminal repeat (LTR) retrotransposons; (iii) long interspersed elements (LINEs); and (iv) SINEs. We comprehensively searched for repetitive elements including TEs from curated libraries of Dfam[35] and Repbase[36] that are inserted in the transcripts in our database. Interestingly, repetitive elements were inserted more in those from the novel genomic loci (83.9% vs. 47.6%, chi-square test, $p < 0.001$), indicating that the TRAILS included transcripts missed by short-read sequencing due to the insertion of repetitive elements.

We also examined the contribution of TEs in the splicing diversity of known genes in the TRAILS. The median length of inserted repetitive elements including TEs was 188 nucleotides (Fig. 3A). The maximum length was 3675 nucleotides, which was a LINE/L1 inserted in the 3'-UTR of a *CCDC7* isoform (Supplementary Fig. 4). The TEs in all isoforms were summed for each class, and the most common were SINEs (Fig. 3B). Comparing the isoforms identified only in our dataset with those common to GENCODE, the former had more TEs inserted (70% vs. 52%, chi-square test, $p < 0.001$, Fig. 3C). The distribution of TEs around gene bodies was non-random, with the highest number of insertions in the 3'-UTR and the lowest numbers in the TSS (Fig. 3D). We then examined the enrichment of each TE class at each position of the isoforms by comparing their proportions with those of the entire genome. Interestingly, the proportion of TEs was non-uniform; LTRs, which are autonomous and coding TEs, were enriched in TSSs and ORFs, while SINEs were enriched in 5'-UTRs and 3'-UTRs (Fig. 3D). Among the isoforms with TEs, that of *LGALS3*, which has numerous functions in the immune system[37], was transcribed from an alternative TSS. Because LTR/ERVL-derived sequences were inserted around the TSS (Fig. 3E), this LTR has brought an alternative promoter as well as an additional CDS into this gene. A region-by-region comparison of the conservation (phastCons score >0.8)[38] of genomic loci with and without repetitive element insertions showed that the inserted regions were significantly less conserved (TSS, odds ratio (OR) = 0.42 [95% confidence interval (95%CI): 0.32−0.53]; ORFstart, OR = 0.58 [95%CI: 0.50−0.66]; ORFend, OR = 0.60 [95%CI: 0.56−0.65]; TTS, OR = 0.49 [95%CI: 0.46−0.52]; splicing junction, OR = 0.71 [95%CI: 0.70−0.73]), suggesting that repetitive elements were inserted into these regions after humans diverged from other mammals, and these contributed to the diversity of isoforms. To test this hypothesis, we compared the Kimura divergences between the TEs that overlapped with exons of GENCODE transcripts and those unique to the TRAILS (Methods), and found that the latter had significantly lower divergences in all TE categories (DNA, peaks are 25.3 and 21.7, $p < 0.001$, two-sample Kolmogorov−Smirnov test; LINE, peaks are 32.8 and 30.5, $p < 0.001$; LTR, peaks are 23.6 and 21.1, $p < 0.001$; SINE, peaks are 11.6 and 8.9, $p < 0.001$) (Fig. 3F). These results indicate that the transcripts in the TRAILS contain recently inserted TEs with fewer base substitutions.

## Isoforms expressed in a cell-type-specific manner

Each immune cell subset expresses cell-type-specific genes, such as those encoding cytokines and transcriptional factors, involved in their respective cellular functions[19,39]. Because some of these are regulated at the isoform level (e.g., a spliced isoform of *RORG* is essential in Th17 cells[23]), cluster analysis based on the isoform ratio, that is the ratio of isoform abundance over the total gene abundance, should connect subsets with identical lineages. To test this hypothesis, we first performed an unsupervised hierarchical clustering analysis based on the similarity of the isoform ratio in each related gene. As expected, we found that subsets of an identical lineage (e.g., B cell subsets) were in close proximity to each other, suggesting that the abundance of isoforms may reflect the functional characteristics of each cell type (Fig. 4A). Then, we examined the isoforms expressed in a cell-type-specific manner utilizing Shannon entropy as an index of cell

specificity and identified 2575 isoforms for 29 cell subsets (Fig. 4B; Supplementary Data 5)[40]. This cell-type specificity was recapitulated in the expression data of the relevant immune cell subsets obtained from short-read RNA-seq[19] (Supplementary Fig. 5A, B).

As cell-type-specific expression of isoforms may occur at the transcriptional level (e.g., alternative TSS usage) and at the post-transcriptional level (e.g., alternative usage of splicing sites or PAS)[41], we examined which mechanism is prominent in the cell-type-specific isoforms. We found that while the proportion of unique TSSs was lower in the cell-type-specific isoforms compared with others (4.8% vs. 5.9%, chi-square test, $p = 0.019$, Fig. 4C), these isoforms have longer 3'-UTRs (729 nucleotides vs. 579 nucleotides, Wilcoxon test, $p < 0.001$) and a higher proportion of unique sequences of the last exon (23.4% vs. 13.0%, chi-square test, $p < 0.001$), suggesting that the cell-type-specific isoforms prefer the usage of alternative splicing sites at the 3'-UTR and PAS. Interestingly, the insertion rates of repetitive elements into junction sites (24.9% vs. 21.1%, chi-square test, $p < 0.001$, Fig. 4C) and TTSs (13.9% vs. 10.5%, chi-square test, $p < 0.001$) was higher in cell-type-specific isoforms.

As RBP binding is a major mechanism of alternative splicing[42], we hypothesized that RBP contributes to post-transcriptional regulation of cell-type-specific expression. We comprehensively searched for RBP binding motifs and compared them between cell-type-specific isoforms and remaining non-specific isoforms. As a result, many kinds of RBP binding motifs were enriched in 3'-UTRs (false discovery rate (FDR) < 0.05, Fig. 4D).

We show examples of cell-type-specific isoforms in Fig. 4E. The isoform ratio of *NLRP1*−4 was the highest in neutrophils, while the isoform ratio of *NLRP1*−19, which lacks the FIIND domain essential for *NLRP1* inflammasome activity[43], was dominant in other cell types. In addition, the isoform ratio of *IL23R*, which is essential for the differentiation of Th17 cells[44], was distinct in Th17 subset in comparison with other cell types (Supplementary Fig. 5C, D). Notably, while a cell-type specific isoform (*IL23R-5*) was identified in Th17 cells, another isoform (*IL23R-2*) specific for activated-Treg cells was also identified. Because the CDSs of these isoforms were different, their differential expression may contribute to Th17 and Treg functions and their plasticity[45].

The previous study demonstrated that the alternative splicing diversity between cell types was greater than that between individuals[2]. We investigated this point by remapping short-read RNA-seq data from 15 immune cell types[19] using our database; we observed that the variance of alternative splicing between individuals was less than a quarter of that observed between cell types (Supplementary Fig. 5E).

## Regulation of translation efficiency by isoform sequences

The efficiency of protein synthesis is governed by the regulatory elements in the 5'-UTR, ORF, and 3'-UTR[5,46,47]. As a classic example, a strong Kozak sequence immediately before the first codon improves start codon recognition as a feature of highly translated mRNAs[46]. In addition, the secondary structure of mRNA may block or recruit ribosomes and other regulatory factors to enable a rapid, dynamic response to diverse cellular conditions[5]. The drawback of these studies is that their analysis is based on gene annotation inferred from short-read RNA-sequencing, which introduces the uncertainty of complex gene regions such as 3'-UTR where abundant repetitive elements are inserted (Fig. 3D). Therefore, we speculated that examining the translational efficiency of each transcript in the TRAILS would provide additional insights into the regulation of translation and bridge the knowledge gap between transcripts and proteins. For this purpose, we remapped the Ribo-seq and RNA-seq data obtained from LCL to the TRAILS, respectively, and calculated the scores of translational efficiencies at the isoform level (Methods; Fig. 5A).

We found a trend that agreed with the previously reported association between Kozak context scores and translation efficiency

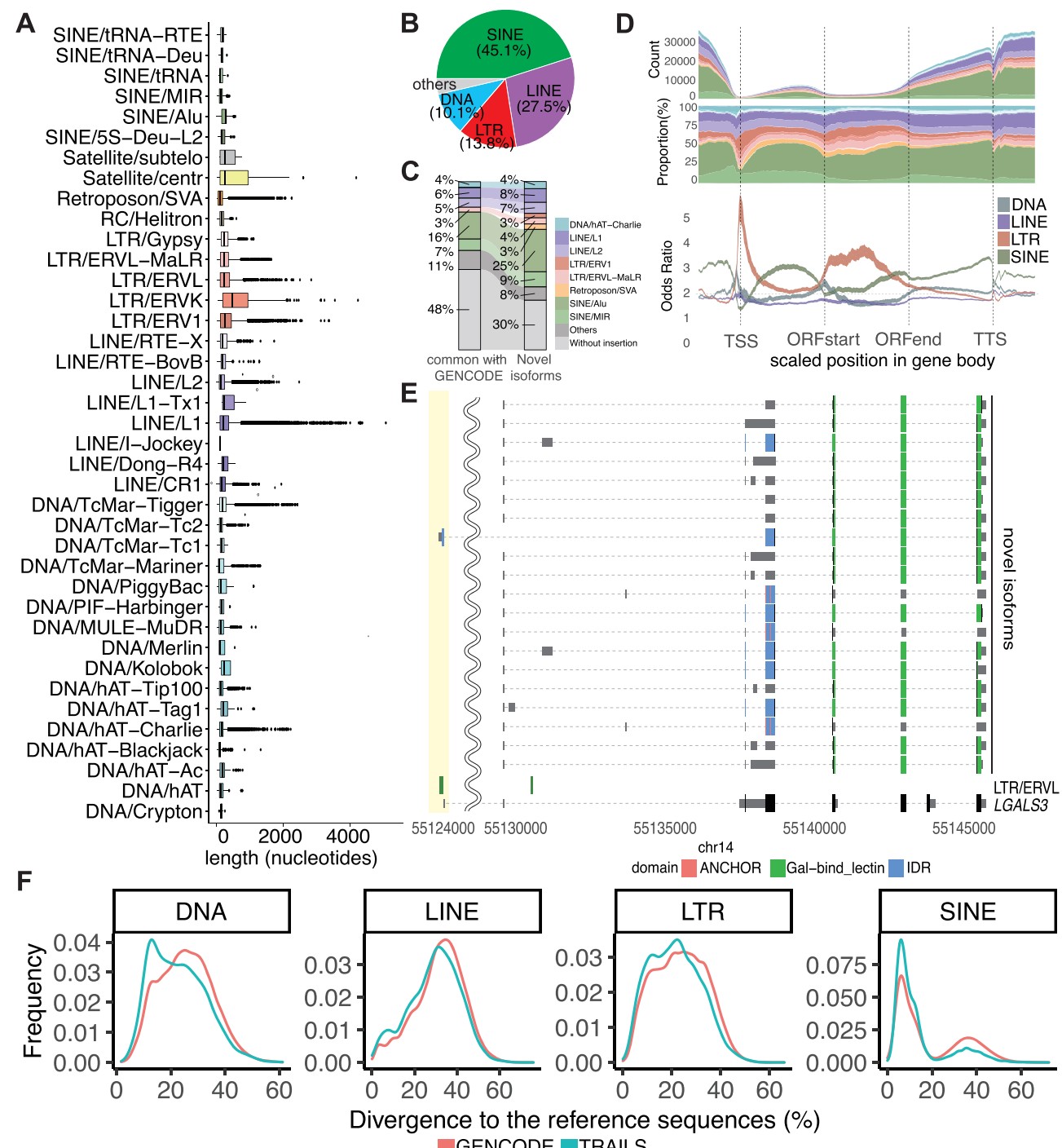

**Fig. 3 | Repetitive elements inserted in isoforms. A** The length of inserted repetitive elements in isoforms according to their class. The color of boxplot indicates their classes. Boxplots, with the center of the box as the median value, the edges of the box denote the first (Q1) and third (Q3) quartiles, minimum/maximum whisker values are calculated as Q1/Q3 -/+ 1.5 × the interquartile range (IQR), are derived from the following counts of inserterd elements: DNA/Crypton ($n = 15$), DNA/hAT ($n = 458$), DNA/hAT-Ac ($n = 149$), DNA/hAT-Blackjack ($n = 2458$), DNA/hAT-Charlie ($n = 45,368$), DNA/hAT-Tag1 ($n = 273$), DNA/hAT-Tip100 ($n = 9117$), DNA/Kolobok ($n = 86$), DNA/Merlin ($n = 5$), DNA/MULE-MuDR ($n = 464$), DNA/PIF-Harbinger ($n = 19$), DNA/PiggyBac ($n = 297$), DNA/TcMar-Mariner ($n = 2102$), DNA/TcMar-Tc1 ($n = 106$), DNA/TcMar-Tc2 ($n = 991$), DNA/TcMar-Tigger ($n = 19,222$), LINE/CR1 ($n = 5389$), LINE/Dong-R4 ($n = 145$), LINE/I-Jockey ($n = 33$), LINE/L1 ($n = 92,415$), LINE/L1-Tx1 ($n = 10$), LINE/L2 ($n = 84,476$), LINE/RTE-BovB ($n = 427$), LINE/RTE-X ($n = 2014$), LTR/ERV1 ($n = 28,183$), LTR/ERVK ($n = 2331$), LTR/ERVL ($n = 16,613$), LTR/ERVL-MaLR ($n = 31,419$), LTR/Gypsy ($n = 2467$), RC/Helitron

($n = 300$), Retroposon/SVA ($n = 27,076$), Satellite/centr ($n = 92$), Satellite/subtelo ($n = 28$), SINE/5S-Deu-L2 ($n = 193$), SINE/Alu ($n = 272,992$), SINE/MIR ($n = 95,582$), SINE/tRNA ($n = 191$), SINE/tRNA-Deu ($n = 4$), SINE/tRNA-RTE ($n = 915$). **B** The proportion of class families in the total repetitive elements inserted in the isoforms. **C** Comparison of the proportion of class families of the total repetitive elements inserted in isoforms between novel isoforms and those that are also registered in GENCODE. **D** The distribution of inserted repetitive elements in gene bodies ±1000 bp. The x-axis is a scaled position relative to the TSS. The y-axis is the counts (top), proportion (center), and the enrichment of each TE class family at the scaled position (bottom). The color represents the class of repetitive elements corresponding with Fig. 3A. **E** Example of isoforms with an inserted LTR at the alternative TSS highlighted in yellow. The collapsed gene structure registered in GENCODE is shown at the bottom. **F** The distribution of Kimura divergences to the reference sequences for four classes of TEs that were overlapped with exons unique to the TRAILS (blue) and GENCODE (red).

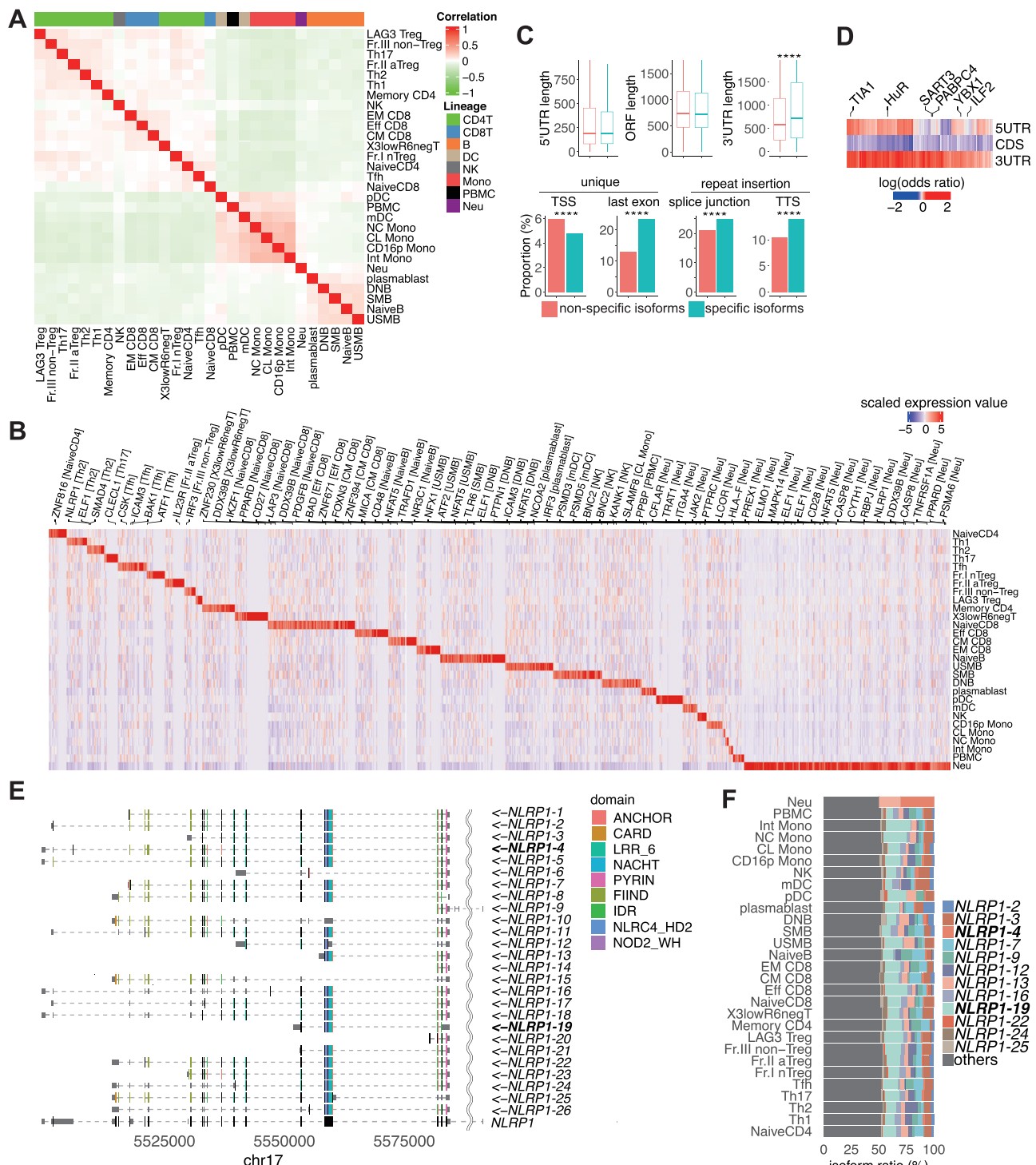

**Fig. 4 | Cell-type-specific isoforms and their characteristics. A** Unsupervised hierarchical clustering based on isoform ratios that were the top 5000 large expression variances among cell types. The colored bars indicate the cell lineages. **B** Cell-type-specific expressed isoforms. Column-wise Z scores of normalized counts are plotted. Representative gene symbols related to specific isoforms are annotated at the top. Full lists of cell-type-specific isoforms are provided in Supplementary Data 5. **C** The length of the 5′-UTR, ORF, and 3′-UTR (top) and the proportion of unique TSS, last exon, insertion of repetitive elements in splicing junctions, and TTS (bottom) compared between cell-type-specific isoforms ($n = 2575$) and others ($n = 156,794$). The significance of comparison is as follows: ****,

nominal $p < 0.0001$; ***, nominal $p < 0.001$; **, nominal $p < 0.01$; *, nominal $p < 0.05$ by two-sided Wilcoxon test. The center of each box plot is the median value, the edges of the box represent the first (Q1) and third (Q3) quartiles, and the minimum/maximum whisker values are calculated as Q1/Q3 -/ + 1.5 × the interquartile range (IQR). **D** Enrichment of each RBP motif according to gene regions compared between cell-type-specific isoforms and others. Representative significant RBPs (FDR < 0.01) are annotated at the top. **E** Example of a specifically expressed isoform transcribed from the *NLRP1* locus. The collapsed gene structure registered in GENCODE is shown at the bottom. Arrows indicate the direction of transcription in the genome. **F** Isoform ratio in *NLRP1*.

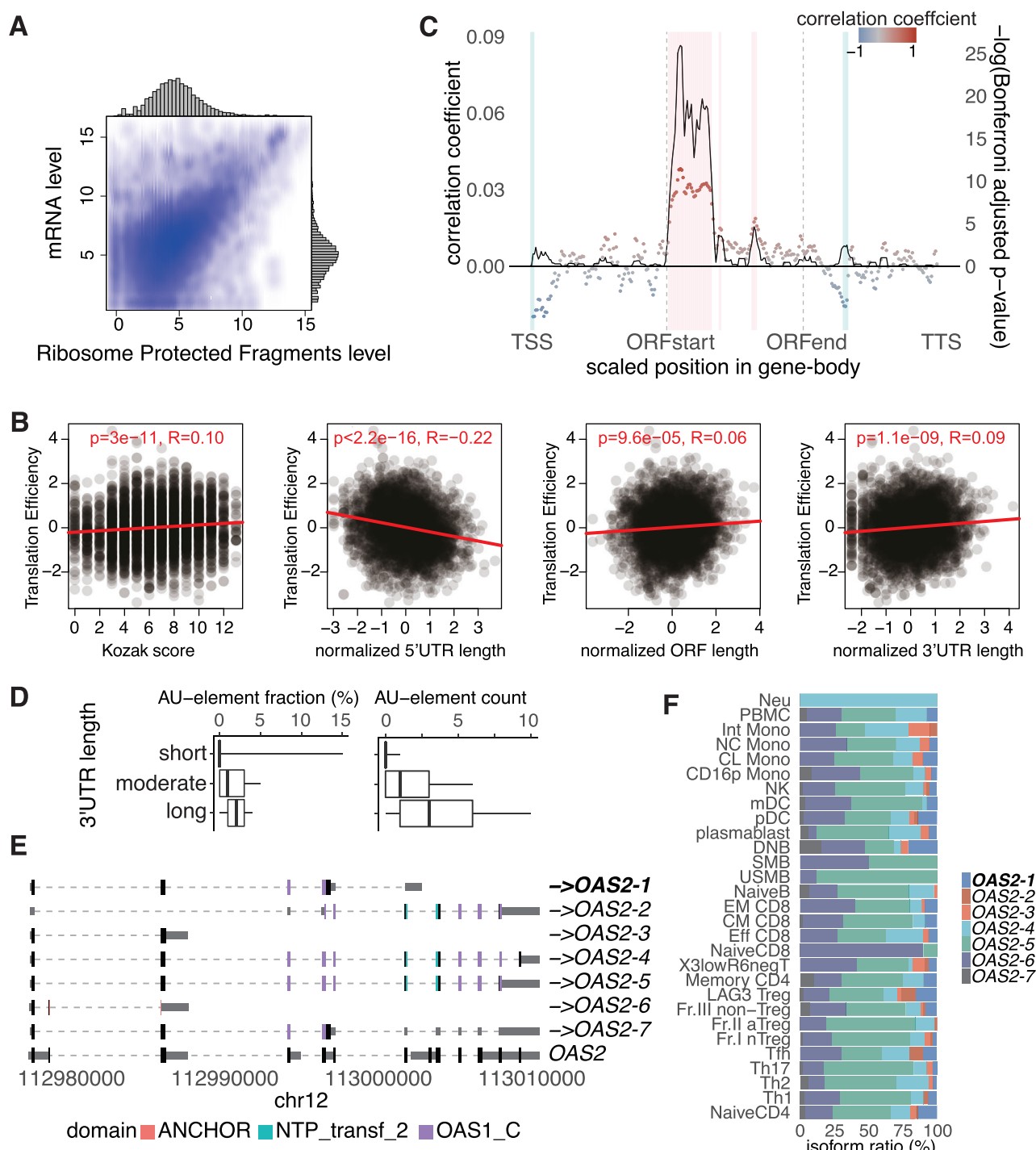

**Fig. 5 | Translational efficiency at the isoform level. A** Scatter plot of isoforms with normalized read counts of RNA-seq and Ribo-seq. Each dot represents a particular isoform. **B** Correlation plots of translational efficiency scores with Kozak context scores (left, $R = 0.10$ and $p = 3.0 \times 10^{-11}$), length of 5'-UTR (left center, $R = -0.22$ and $p < 2.2 \times 10^{-16}$), ORF (right center, $R = 0.06$ and $p = 9.6 \times 10^{-5}$), and 3'-UTR (right, $R = 0.09$ and $p = 1.1 \times 10^{-9}$). Each '$R$' represents the Spearman's rank correlation coefficient. P-values were obtained by two-sided test. **C** Correlation between translational efficiency and local folding strength of isoforms. The x-axis is a scaled position relative to the TSS. The solid line and colored points represent $-\log_{10}$ (Bonferroni adjusted p-value) and Pearson correlation coefficient, respectively. The test conducted is two-sided. The areas with colored rectangles were significantly (Bonferroni adjusted $p < 0.05$) correlated at the scaled position in the gene body. **D** Association between 3'-UTR length and AU-element fraction. 3'-UTR lengths are binned into three groups—short, moderate, and long—based on the logarithm (base 10) of their nucleotide lengths. The binning was performed using equal-interval discretization of the log-transformed lengths, creating three bins of equal log-scale range. Consequently, boxplots are derived from the following counts of isoforms: short ($n = 13,580$), moderate ($n = 126,471$), and long ($n = 14,469$). The center of the box plot is the median value, the edges of the box represent the first (Q1) and third (Q3) quartiles, and the minimum/maximum whisker values are calculated as Q1/Q3 -/ + 1.5 × the interquartile range (IQR). **E** Example of the effectively translated isoform transcribed from the *OAS2* locus. The collapsed gene structure registered in GENCODE is shown at the bottom. The effectively translated isoform (top 10%) is highlighted in bold. **F** Isoform ratio in *OAS2*.

scores[46] (Fig. 5B, Supplementary Fig. 6A). We also observed a slight association between translational efficiency and the lengths of the 5′- and 3′-UTRs was reversed; i.e., shorter 5′-UTRs, as well as longer 3′-UTRs, were associated with higher translation efficiency (Fig. 5B). To explore the underlying mechanism of association between the length of 3′-UTR length and translational efficiency, we investigated the sequence characteristic and found that transcripts with high translational efficiency had more AU-rich elements (Supplementary Fig. 6A, B). Further, transcripts with longer 3′-UTR lengths had a higher occupancy of AU-rich elements in the 3′-UTR (Fig. 5D). It is known that binding of HuR (*ELAVL1*) to AU-rich elements in 3′-UTR prevents degradation of mRNA[48]. Thus, HuR could promote translation efficiency via enhancing mRNA stability by binding to transcripts that contain AU-rich elements in the 3′-UTR. Such a regulatory mechanism of translation efficiency may be characteristic of immune cells, since *ELAVL1* is expressed ubiquitously in 29 immune cell types in TRAILS (Supplementary Fig. 6C) and most abundantly expressed in LCL across human tissues (Supplementary Fig. 6D)[7].

To further investigate cell-type specificity of the effect of 3′-UTR features, we remapped Transcript Isoforms in Polysomes sequencing (TrIP-seq) data[49] from HEK293T to our TRAILS (Methods). We observed a concordant distribution of polysomes in HEK293T compared to the translational efficiency based on Ribo-seq data from LCL (Supplementary Fig. 6E). However, when comparing isoform clusters with high and low polysomes (Supplementary Fig. 6F), we found opposite effects of 3′-UTR features (length and AU-rich elements) between HEK293T and LCL (Supplementary Fig. 6A, G).

Then, we calculated the correlation between the local folding strength of RNA and the scores of translation efficiency (Methods). We confirmed that the local folding strengths calculated for the sequences of each isoform were correlated with the Parallel Analysis of RNA Structure (PARS) score, which was calculated from two high-throughput sequencing libraries per sample and provided profiling of the secondary structure at a single nucleotide resolution[50,51] ($R = 0.67$, $p < 0.001$, Spearman's correlation test, Methods; Supplementary Fig. 6H), indicating the credibility of local folding strengths. In addition, we found a negative correlation between the local folding strength of the 5′ leader (5′ end of an isoform) and higher translation efficiency (Fig. 5C). In contrast, the local folding strength immediately after the first codon was positively correlated with the translational efficiency. These results are consistent with previous findings: stable RNA secondary structures at the 5′ leader, such as cap-proximal hairpins, block the assembly of the 43S pre-initiation complex onto the 5′-UTR, while a hairpin positioned downstream of a first codon enhances translational initiation[52,53], warranting our analysis of translation efficiency at the isoform level. As for the 3′-UTR, there was one scaled position with a negative correlation, which may reflect the complex association of RBP binding, miRNAs, and secondary structure[54].

Furthermore, we investigated whether other features unique to the isoforms were associated with translational efficiency. For that purpose, we compared the characteristics of isoforms having the highest and lowest scores of translational efficiencies (top 10% and bottom 10%, respectively). As a result, we found that having a unique TSS was associated with higher translation efficiency (7.0% vs. 2.3%, chi-square test, $p < 0.001$). In addition, those encoding a non-canonical ORF, which was defined as the first codon other than methionine using GeneMarkS-T[25], were associated with higher translation efficiency (26.1% vs. 17.6%, chi-square test, $p < 0.001$). Non-normal ORFs can be translated into proteins and have received much attention in immunity and cancers[55,56]. In the TRAILS, 28,837 isoforms were predicted to encode non-canonical ORFs. Of these, we examined 4780 isoforms for peptides by proteome analysis, and found that 22 isoforms translated peptides from the non-canonical ORFs (Supplementary Data 3). A specific example of isoforms encoding non-canonical ORF in our database is the *LGALS3*-isoform, already noted above (Fig. 3E). This isoform has a unique TSS and non-canonical ORF with serine predicted as the first codon, and its translation efficiency was in the top 10%. Another example of effectively translated isoforms was *OAS2-1* (Fig. 5E). This isoform has the shortest 5′-UTR and longest 3′-UTR among the isoforms transcribed from the *OAS2* locus. The translational efficiency of *OAS2-1* was in the top 10% and expressed abundantly in double negative (DNB) B cells (Fig. 5F).

## TRAILS usage for disease transcriptomics and genomics

To investigate whether the isoforms identified in our dataset are involved in disease pathogenesis, we compared the abundance of isoforms between case and control subjects, taking systemic lupus erythematosus (SLE) as an example. SLE is an autoimmune disease with activation of interferon signature genes known to be involved, though details of the mechanism at the isoform level are not well understood[57]. To investigate pathogenic isoforms in SLE, we remapped short-read RNA-seq datasets obtained from whole blood cells of SLE and healthy subjects to the TRAILS (SLE = 99, healthy subjects = 18)[58] (Methods). As a result, we identified 84 genes whose isoform fractions were significantly switched between SLE and healthy individuals (FDR < 0.05). Among them, IRAK1 transduces signals from TLR7 and TLR9 by phosphorylating IRF7 to promote IFNα transcription[59]. One known isoform of *IRAK1* (*ENST 00000393687.6*) contains a protein kinase domain, but the novel *IRAK1-1* lacks this domain (Fig. 6A; Supplementary Fig. 7). Although there was no difference in gene-level expression between case and control samples (Fig. 6B), the isoform fraction significantly switched, resulting in higher expression of a functional isoform (*ENST 00000393687.6*) in SLE (Fig. 6C). This implied that TLR7/9-IRAK1-IRF7 pathway activation due to upregulated expression of the functional *IRAK1* isoform (*ENST 00000393687.6*) in SLE may contribute to type 1 IFN dysregulation.

Rheumatoid arthritis (RA) is another example of autoimmune diseases characterized by chronic inflammation of the synovial[60]. We applied our database to a single immune subset, CD45RA-positive effector memory (Temra) CD8 + T cells (active RA = 8, healthy subjects = 9)[61,62]. We found that three novel isoforms in *SIGLEC10* gene, which suppresses inflammatory responses to danger (damage)-associated molecular patterns by interacting with CD24[63], were differentially expressed in Temra CD8 + T cells from active RA and healthy controls subjects (Fig. 6D). Two of them (novel isoforms 2 and 3) were predicted to be sensitive to nonsense-mediated mRNA decay (NMD), which is a surveillance pathway to degrade RNA and prevent the production of abnormal[64]. At the gene level, *SIGLEC10* was more highly expressed in RA compared to healthy individuals (Fig. 6E). However, at the isoform level, NMD-sensitive isoforms were dominant in RA (Fig. 6F). This suggests that even though the expression was increased at the gene level, the relative expression of aberrant isoforms susceptible to NMD was increased, resulting in a relatively low inflammatory suppressive function of *SIGLEC10* in RA.

Analysis of differentially expressed isoforms as presented above identified disease-relevant isoforms, but they may simply reflect the disease course (i.e., they result from the disease). If alternative splicing is regulated by genetic variants that are defined as sQTL, and meanwhile the variants present susceptibility to disease, the variants and the splicing events has the potential to be causal for the disease. Thus, integration of sQTL analysis and GWAS data can comprehensively reveal isoforms involved in disease pathogenesis. To address this, we first examined the impact of different reference annotations on junction-based sQTL. We remapped RNA-seq data of LCL samples derived from European subjects in the GEUVADIS cohort[8] with each of two annotations, GENCODE and the TRAILS, independently, and compared the number of junctions with significant sQTL (FDR < 0.05). Of the total, 14.8% were identified only in the TRAILS as a reference (Fig. 7A). To further ensure that sQTLs identified only in our database were not simply false positives, we verified that sQTL variants were

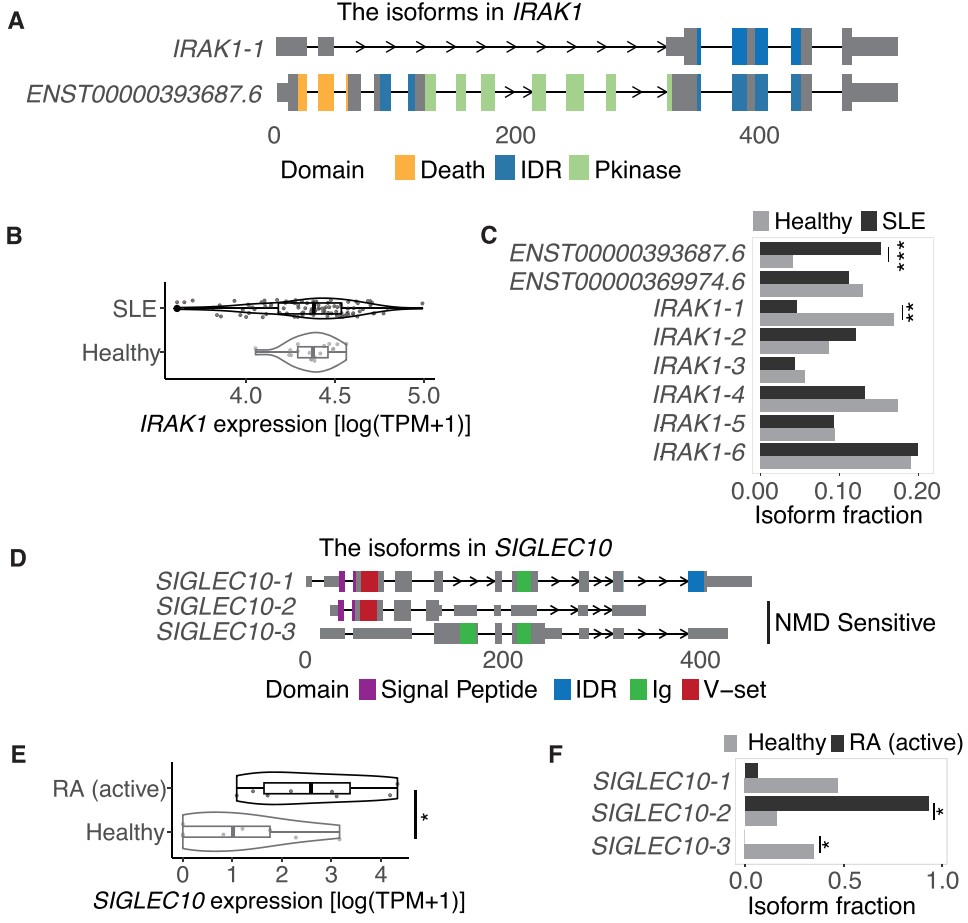

**Fig. 6 | Switched isoforms in IMDs. A** Structures of switched isoforms that are transcribed from the *IRAK1* locus between SLE and healthy controls. The x-axis shows the distance from the TSS. **B** Expression of *IRAK1* at the gene level. Boxplots, with the center of the box as the median value, the edges of the box denote the first (Q1) and third (Q3) quartiles, minimum/maximum whisker values are calculated as Q1/Q3 -/ + 1.5 × the interquartile range (IQR), are derived from the following counts of individuals: SLE (*n* = 99) and healthy controls (*n* = 18). **C** Isoform fractions (IF) in *IRAK1* (*ENST00000393687.6*, FDR = 3.6 × 10⁻⁷; *IRAK1*−1, FDR = 1.1 × 10⁻⁶). **D** The structure of switched isoforms that are transcribed from *SIGLEC10* locus between active RA and healthy controls. **E** The expression of *SIGLEC10* at the gene level. Boxplots, with the center of the box as the median value, the edges of the box denote the first (Q1) and third (Q3) quartiles, minimum/maximum whisker values are calculated as Q1/Q3 -/ + 1.5 × the interquartile range (IQR), are derived from the following counts of individuals: active RA (*n* = 8) and healthy controls (*n* = 9). **F** IF in *SIGLEC10* (*SIGLEC10*−2, FDR = 1.1 × 10⁻²; *SIGLEC10*−3, FDR = 1.1 × 10⁻²). The gene expression values in (**B**, **E**) were normalized by log-transformed Transcripts Per Kilobase Million (TPM). Statistical tests for isoform switch analysis in (**C**, **F**) were performed using the isoformSwitchTestDEXSeq function from the IsoformSwitchAnalyzeR package with default options. The test conducted is two-sided, assessing both increases and decreases in IF. The significance of the comparison is as follows: ***FDR < 0.001; **FDR < 0.01; *FDR < 0.05.

significantly enriched in the GWAS variants compared with genome-wide variants (Fig. 7B).

Next, we remapped RNA-seq datasets from various cell conditions[8,18,19,65] to refine the mapping of sQTL so that we could identify pathogenic isoforms using the TRAILS. In addition to junction-based sQTL, we performed eQTL for genetic variants that affect gene expression, and 3′aQTL[66], which associates cis-acting genetic variants with alternative polyadenylation (APA). The numbers of genes or isoform-related genes having effects on eQTL, sQTL, and 3′aQTL in one or more cell conditions were 9775, 10189, and 4087, respectively (nominally significant $p < 1 \times 10^{-5}$). Genes involved in the IFN signaling pathway were enriched in 3′aQTL-eGenes, as previously reported (Supplementary Fig. 8A). As expected from a previous study[7], the number of QTL variants increased with the sample size for each cell type (Supplementary Fig. 8B). eQTLs and 3′aQTL variants were abundant around TSS and TTS, respectively, while sQTLs were distributed throughout the gene body (Supplementary Fig. 8C). We then examined the proportion of heritability of IMDs and neurological diseases by each QTL using stratified linkage disequilibrium score regression

analyses (S-LDSCs)[67], and found that many QTLs significantly contributed to the heritability of both disease types (Fig. 7C).

Finally, we evaluated the colocalization between GWAS loci for IMDs and neurological diseases (16 GWAS data in total, Supplementary Data 6) and QTL signals using coloc (PP4 > 0.8). To take advantage of the TRAILS, which contains full-length information on isoforms, and to detect more potential pathogenic isoforms, we also performed trQTL analysis. As a result, we found that 20−60% of GWAS loci colocalized with eQTL, sQTL (junction-based sQTL and trQTL), and up to 20% of GWAS loci with 3′aQTL (Fig. 7D). Notably, we identified several eQTL signals for genes identified only in TRAILS, but not in GENCODE, that colocalized with GWAS signals (Supplementary Data 7). Among the sQTL colocalized with GWAS loci, an SNP in *MALT1* (rs11873030), which was associated with multiple sclerosis[68], had an sQTL effect for a junction read (chr18:58739121-58741865) unique to *MALT1*−1; this isoform decreased with the risk allele for the disease (Fig. 7E−G). MALT1 transduces NF-kappaB (NFκB) signaling by antigen receptor stimulation, and importantly, the isoform with the sQTL effect lacks the MALT1 C-terminal immunoglobulin-like domain (Fig. 7E).

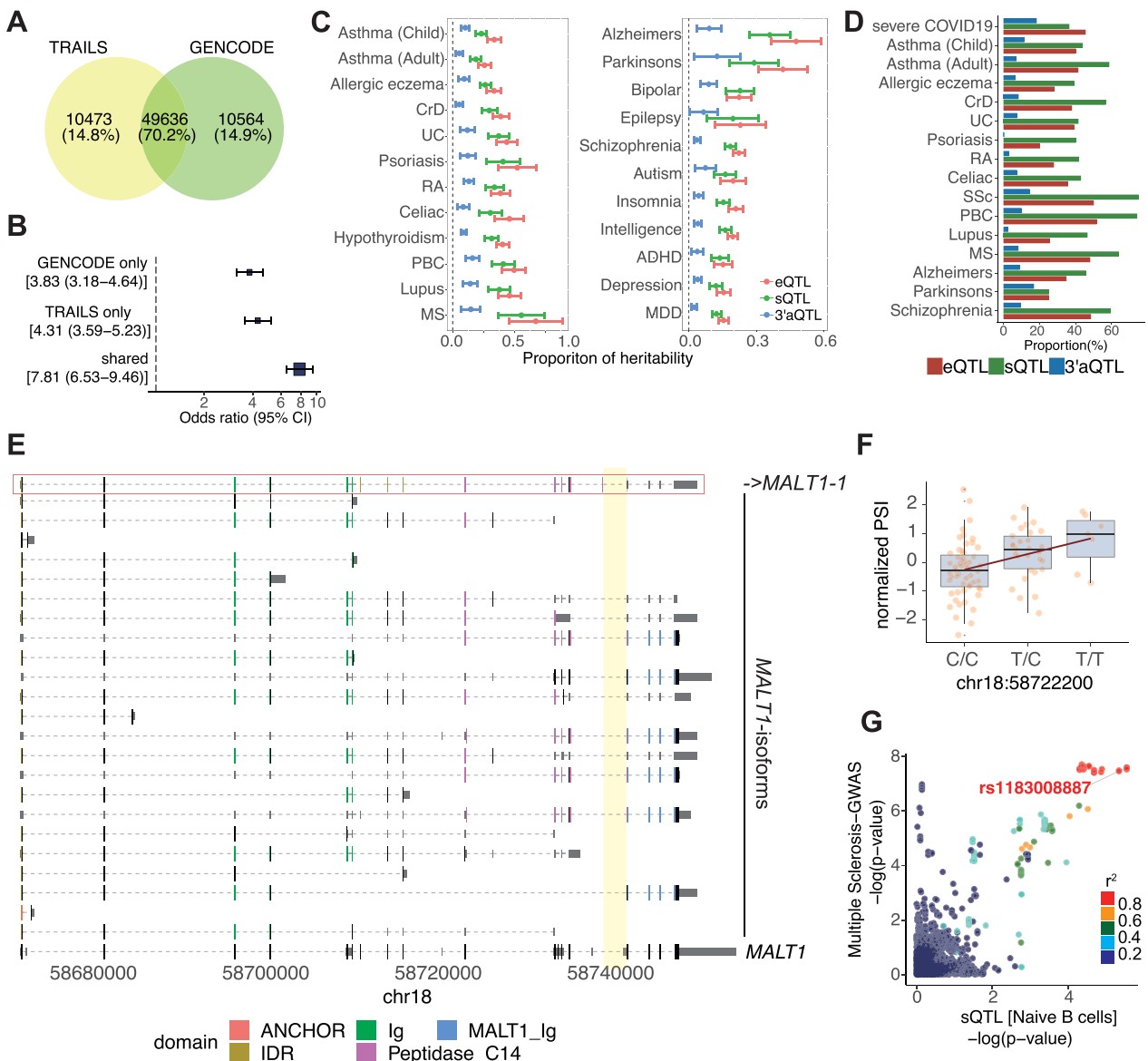

**Fig. 7 | QTL and colocalization analysis with GWAS data. A** Venn diagram for junctions with significant sQTL signals (FDR < 0.05) in comparison with the TRAILS and GENCODE as a reference. **B** The enrichment of junction-based sQTL variants (GENCODE only, 83,766 variants; TRAILS only, 86,496 variants; shared, 858,862 variants) in GWAS variants compared to randomly selected 10,000 genome-wide variants (autosomes, minor allele frequency > 0.01 in European subjects of 1000 Genomes Phase 3[107]). The list of GWAS data is available in Supplementary Data 6. Error bar indicates odds ratio ±1.96 standard error. **C** The proportion of the heritability of genomic region, considering a 500 bp upstream and downstream of each QTL variant (eQTL, 214,254 regions; sQTL, 166,676 regions; 3'aQTL, 46,040 regions), in IMDs (left) and neurological diseases (right). Error bar indicates point estimate ±1.96 standard error. CrD, Crohn's Disease; UC, ulcerative colitis; RA, rheumatoid arthritis; PBC, primary biliary cirrhosis; MS, multiple sclerosis; ADHD, attention-deficit hyperactivity disorder; MDD, major depressive disorder. **D** The

proportion of colocalized loci with each kind of QTL variants among all GWAS loci. SSc, Systemic Sclerosis. **E** Structures of isoforms transcribed from *MALT1* with a percent spliced in (PSI) value of a unique junction (chr18:58739121-58741865, highlighted in yellow) that is strongly associated with the GWAS signal of multiple sclerosis in naïve B cells. The related isoform to the unique junction (top) is framed by the red line. The collapsed gene structure registered in GENCODE is shown at the bottom. **F** QTL plot of normalized PSI of the unique junction in naïve B cells according to each genotype (*n* = 91). The center, edges, and minimum/maximum whisker values of the box are the median value, the first (Q1) and third (Q3) quartiles, and Q1/Q3 -/+ 1.5 × the interquartile range (IQR), respectively. **G** Colocalization plot of sQTL and GWAS of multiple sclerosis. Statistical test was performed using the coloc package in R to estimate the posterior probabilities of shared causal variant.

## Discussion

Our datasets, obtained by long-read RNA sequencing of 29 immune cell subsets, provide a comprehensive full-length isoform profiling of the human immune system. We found an enormous complexity of splicing events that are substantially shaped by insertions of TEs into the human genome. This complexity brought by TEs may have been underestimated, owing to the difficulty in read-mapping of TEs obtained by short-read sequencing. Our database comprehensively

demonstrated using long-read sequencing that TEs confer isoform diversity to immunological genes by introducing alternative TSS, splicing sites, and polyadenylation signals, as suggested by previous studies[69–71]. Considering their non-uniform distribution in the gene body, each class of TE would have a unique role in adding functions to the immunological genes. These additional functions obtained by TEs could have been naturally selected by the environment, because immune cells play a critical role in protecting the host from external

pathogens such as viruses or bacteria by triggering local and systemic inflammation. Indeed, a recent study demonstrated that the short isoform of the *ACE2* gene, which encodes a receptor for SARS-CoV-2, is upregulated by an interferon-inducible alternative promoter introduced by MIRb and LTR16A1 elements[72].

Our database contained many read-through transcripts, which are one of the most understudied categories of transcript isoforms, especially with regards to disease genetics. The *TOMM40_APOE* locus, where a novel read-through isoform was identified, is the most significant locus for Alzheimer's disease[73]. Notably, trQTL signals of the read-through isoform in non-stimulated monocytes were marginally colocalized with GWAS signals (PP4 = 0.74, Supplementary Fig. 8D, E). The disease risk allele of rs204468 increased the isoform ratio of the read-through isoform among the isoforms transcribed from the *TOMM40* locus (trQTL beta = 0.64, $p = 7.1 \times 10^{-6}$). Because multiple independent association signals have been detected in this locus[73], in addition to the well-established variant of *APOE4*, additional variants regulating the ratio of read-through isoform may also contribute to the pathogenesis of Alzheimer's disease.

We also demonstrated cell-type-specific alternative splicing in the immune cell subsets. This may elucidate the role of genes in each immune cell subset that regulates inflammatory signals, as exemplified by the isoform of *NLRP1*. As a major mechanism for cell-type-specific expression, we found that alternative usage of the 3′-UTR was important. The length of the 3′-UTR has immensely increased during eukaryotic evolution in comparison with the 5′-UTR[74], indicating that they may have acquired different contributions to cellular functions. Although a significant portion of the diversity of alternative splicing is attributable to cell- or tissue-specific splicing[2,75–77], it is important to consider the diversity based on biological characteristics such as age, sex, and health status. In the future, advancements in long-read sequencing technology are expected to enable better exploration and understanding of these factors. Regarding the observed cell type specificity, we primarily utilized isoform ratios rather than raw read counts to minimize biases associated with overall gene expression levels. However, it is important to acknowledge that the reliability of the isoform ratio may decrease for genes with very low read counts. In fact, for *IL23R*, which showed cell type-specific isoform expression in Th17 and Treg cells, we observed that certain T cell subsets, such as Tfh cells, did not have detectable expression of the gene in our dataset. Considering our previous research identifying associations between low-expressed isoforms and diseases[11], there is potential to uncover new insights by sequencing more reads and/or increasing the sample size, which could lead to the identification of additional cell type-specific isoforms and their relevance to disease mechanisms.

We found associations between the features of transcripts and translational efficiency. Our analysis captured the whole structure of transcripts using long-read sequencing, while previous reports have predominantly relied on annotations derived from short-read RNA-seq or transcript structures inferred from mapped short-reads using computational algorithms[41]. These traditional methods might make it difficult to infer whole transcript structures, especially in 3′-UTRs that contain many repetitive structures as we showed. We addressed this point by utilizing long-read sequencing and observed significant features relating to translational efficiency. Corresponding with the previous study[49], we observed differences in cell types as to the effect of 3′-UTR features on translational efficiency, whereas those of 5′-UTR were consistent. This suggests that different cell types have different mechanisms for controlling translation, such as binding of RNA-binding proteins like HuR to 3′-UTR. Our analysis reveals a slight transcriptome-wide trend regarding the role of each transcript feature, intriguing the necessity of experimental validation for individual transcripts in the future.

Finally, we identified novel pathogenic mechanisms via alternative splicing by performing isoform switch analysis and integrated analysis of QTLs and GWAS data. Gene-level analyses, such as differentially expressed gene analysis and eQTL analysis, alone cannot capture changes in the coding sequences and functional regions of alternatively spliced isoforms, as was the case in the *IRAK1* gene for SLE. In addition, our sQTL analysis showed that the TRAILS, used as a reference annotation, can identify functional junction-based sQTL variants that were missed when using GENCODE. Since existing alignment tools[78] preferred to align junction reads to known junctions, the use of annotations having more complete and accurate information on the splicing junctions is critical for sQTL analysis. Furthermore, our database directly provides full-length sequences for isoforms identified in trQTL analysis as well as those in sQTL analysis that have unique junction sequences. When we mapped simulated short-read RNA-seq data to our annotation, the number of transcripts from a given gene region did not strongly impact the accuracy of quantification (Supplementary Fig. 9). However, the large variability, measured by variance of transcript abundance, suggests caution is advisable when mapping short-read RNA-seq data to TRAILS, and junction-based sQTL analysis would yield more reliable results.

In summary, we made the database for isoforms expressed in 29 immune cell types using long-read sequencing technology. Analysis of existing and future short-read RNA-seq datasets combined with the TRAILS will facilitate the discovery of unknown pathogeneses of diseases and new therapeutic targets.

## Methods

### Sample collections

We sorted PBMCs into 29 immune cell subsets from a healthy volunteer (42-year-old female) using a 14-color cell sorter, BD FACSAria Fusion (BD Biosciences), with purity >99% using a MoFlo XDP instrument (Beckman Coulter) (Supplementary Data 1, 2). Erythrocytes were lysed with potassium ammonium chloride buffer, and non-specific binding was blocked with Fc-gamma receptor antibodies. Sorted cells were lysed and stored at −80 °C. Total RNA was extracted using the MagMAX total RNA kit (Ambion, Life Technologies). Total RNA preparation (100 µg) was added to 100 µL nuclease-free water and poly-A selected using NEXTflex Poly(A) Beads (BIOO Scientific) according to the manufacturer's instructions and stored at −80 °C. We intended to collect 5000 cells with at least 1000 cells per subset (5000 cells were collected for >80% of samples). We followed previously reported immune cell definitions provided by the Human Immunology Project[79] for the flow cytometry staining panel with slight modification due to the availability of labeled antibodies. In addition, CXCR3$^{low}$CCR6$^{-}$ (X3$^{low}$R6$^{neg}$) T cells and LAG3$^{+}$ Treg cells were sorted following previous studies, respectively[80,81]. Neutrophils were collected with EasySep Direct Human Neutrophil Isolation Kits (STEMCELL Technologies) or MACSxpress Neutrophil Isolation Kits human (Miltenyi Biotec) with an aim of $2 \times 10^6$ cells, lysed, and stored at −80 °C, followed by RNA isolation with a RNeasy Mini Kit (QIAGEN). We additionally collected PBMC from a 40-year-old male to validate the presence of isoforms in TRAILS. This study was approved by the Ethics Committees of the Medical Research Institute, Tokyo Medical and Dental University and RIKEN Center for Integrative Medical Sciences. Written informed consent was obtained from each volunteer. The design and conduct of this study fully complied with all relevant regulations regarding the use of human study participants and adhered to the ethical principles outlined in the Declaration of Helsinki.

### Long-read RNA sequencing and processing

We prepared cDNA libraries with a SMART-seq v4 Ultra Low Input RNA Kit (Takara Bio) and SQK-LSK109 (Oxford Nanopore Technologies). We used SMARTScribe Reverse Transcriptase for cDNA synthesis and SeqAmp DNA Polymerase for PCR amplification (20 cycles) of cDNA. A hundred fmol of cDNAs were sequenced by MinION Flow Cell (R9.4.1, FLO-MIN116; Oxford Nanopore Technologies) for 48 h. Basecalling was

performed using Guppy with the SUP (super high accuracy) model. We aligned reads to the GRCh38 genome reference using minimap2[82] with default parameters and reference to the splice junctions in the GEN-CODE annotation. The median number of raw reads in 29 subsets was 5,721,968 (Supplementary Fig. 1A). We mapped raw reads with quality >Q7 and used SAMtools[83] to filter out reads with a mapping quality (MAPQ) less than 10, resulting in 4,566,622 reads as the median of 29 subsets (Supplementary Fig. 1B). Then, we used the flair pipeline[84] to identify the full-length of isoforms and filtered them using the following criteria: (1) transcripts whose 5′ end was located within 100 bp from the TSS annotated by refTSS[85] and/or TSSclassifier ("relaxed" or "strict"), which are based on the FANTOM CAGE (Cap Analysis of Gene Expression) peak[86], were extracted, (2) transcripts that have at least three full-length supporting reads (80% coverage and spanning 25 bp of the first and last exons) in total with the "--stringent" option (Supplementary Fig. 1C). For splicing junction correction in the process of the flair pipeline, we used realigned reads with the GENCODE annotation obtained from 30 short-read RNA-seq datasets of immune cells[8,18,19,65,87].

Then, we used SQANTI3 with default parameters to filter out transcripts that are considered artifacts by intra-priming and reverse-transcription switching[88] and to classify structural categories.

For validation of TRAILS, we sequenced independent PBMC sample from a 40-years-old male using PromethION P2 Solo (R10.4.1, V14 chemistry). The reason we selected PBMC is that PBMC contains all cell types in our TRAILS. We mapped raw reads with quality score >15 and used SAMtools to filter out reads with MAPQ > 10, resulting in 26,551,929 reads (average read length is 1760 base). We then generated the isoform annotation containing 100,607 isoforms (male PBMC annotation) using the same FLAIR pipeline as our original TRAILS.

Transcripts were compared to Workman et al. flair-called transcripts[24] using gffcompare[89] with the "--strict-match" option, which only allows a limited variation of the outer coordinates of the terminal exons by at most 100 bases.

## Genes susceptible to alternative splicing

To investigate the characteristics of genes that are susceptible to alternative splicing, we performed linear regression with the objective variable as the number of alternatively spliced isoforms and the explanatory variable as the read counts of genes in long-read sequencing. Then, we extracted genes with the top 5% highest and lowest 5% residuals by the resid function in the stats R package and performed pathway analysis using Enrichr[89,90] independently.

## ATAC-seq

ATAC-seq data of immune cells[32] were obtained from GEO under accession GSE118189. As in the original paper, we processed raw reads as follows: we trimmed transposase adapters with Trim_Galore with a minimum length of 20 in paired-end mode. We aligned trimmed reads using Bowtie2[91] with default parameters. The Bowtie2 index was constructed with the default parameters for the GRCh38 reference genome. We filtered out reads that mapped to chrM and used SAMtools[83] to filter out reads with MAPQ < 10. Additionally, duplicate reads were discarded using Picard Chromatin accessibility peaks were identified with MACS3[92] under default parameters and '--nomodel --nolambda --keep-dup all --call-summits'. The count of absolute peaks per cell type refers to the number of peak regions reported in the 'narrowPeak' file (peaks with multiple summits are only counted once). The peak count estimates were adjusted by sample read depth.

## In silico annotation for isoforms

For each isoform, we annotated whether it encodes a protein[25], causes nonsense-mediated decay (NMD)[88], contains repetitive elements[93], is a transmembrane protein[94], contains a domain motif[95], contains a signal peptide[96], or has intrinsically disordered regions (IDRs)[97] in silico based on the nucleotide sequence obtained by long-read sequencing. To annotate inserted repetitive elements, we used the isoform sequence through the gene body and the reference genome (GRCh38) sequence for upstream and downstream (±1 kb) of the gene body. Protein 3D structures are predicted using AlphaFold2[98].

## Kimura divergences

The Kimura two-parameter (K2P) model is the measure of nucleotide substitutions that have occurred during the evolutionary process[99]. To calculate the genetic distance between the TRAILS and GENCODE transcripts, we extracted the TEs that overlapped with exons of GEN-CODE transcripts (645,325 TEs) and those unique to the TRAILS (34,150 TEs). Kimura divergences were then calculated using RepeatMasker[93], and differences were tested using the two-sample Kolmogorov-Smirnov test.

## Proteome analysis by nanoLC/MS/MS

We referred to ongoing proteome analysis data by liquid chromatography (LC)/mass spectrometry (MS)/MS-based global proteomics and protein terminomics (Nishida and Ishihama et al., manuscript in preparation) to validate whether predicted ORFs encoded in isoforms in the TRAILS were translated. For sample preparation, $1 \times 10^7$ of THP-1 cells (ATCC; American Type Culture Collection) with/without 10 ng/ml of Phorbol 12-myristate 13-acetate (PMA) treatment for 72 h and LCL cells (NA12878, Coriell Institute) with/without 50 ng/ml IFN-α2 for 6 h were lysed with phase transfer surfactant buffer[100,101] followed by digestion with Lys-C/trypsin. These four samples were divided into four fractions each (16 samples in total) and labeled with 16-plexed TMTpro reagents to prepare a single TMT set. For protein termi-nomics, the TMT set was used to isolate protein terminal peptides using a strong cation exchange (SCX) chromatography system consisting of an Agilent 1260 Infinity II Bio-Inert LC with a BioIEX SCX column (250 mm × 4.6 mm, 5 μm, nonporous) (Santa Clara, CA), as described previously[101]. The isolated peptides were fractionated into 24 vials by reversed-phase HPLC at high pH conditions using a Nexera X2 system (Shimadzu, Japan, Kyoto) with a L-column 3 (2.1 mm × 150 mm, 3 μm, 110 Å). We also conducted protein C-terminomics using the CHAMP protocol[102]. In brief, the cell lysates were digested by V8 pro-tease. After dividing each sample into four fractions, the protein C-terminal peptides were isolated using $CeO_2$ chromatography. The 16 multiplexed samples were mixed to prepare a single TMT set and fractionated into 24 vials by reversed-phase HPLC as described above. For global proteomics, the tryptic digests of four different samples were divided into four fractions each (16 samples in total) and labeled with 16-plexed TMTpro reagents to prepare a single TMT set and fractionated into 24 vials by reversed-phase HPLC as described above. All samples were desalted by SDB-StageTips[102,103].

NanoLC/MS/MS measurement was performed on an Orbitrap Exploris 480 mass spectrometer (Thermo Fisher Scientific, Waltham, MA) and an Ultimate 3000 LC system with a self-pulled needle column (250 mm, 100 μm ID) packed with Reprosil-Pur 120 C18-AQ 1.9 μm (Dr. Maisch, Ammerbuch, Germany). The flow rate was 400 nL/min. The LC mobile phases consisted of solvent A (0.5% acetic acid) and solvent B (0.5% acetic acid and 80% acetonitrile). The gradient was set as follows: 5–10% B in 2.5 min, 10–19% B in 57.8 min, 19–29% B in 21 min, 29–40% B in 8.7 min, and 40–99% B in 0.1 min, followed by 99% B for 5 min. The electrospray voltage was set to 2.4 kV in the positive mode. For the survey scan, the mass range was from 375 to 1600 $m/z$ with a resolution of 60,000, 100% normalized AGC target, and auto maximum injection time. For the MS/MS scan, the first mass was set to 110 $m/z$ with a resolution of 45,000, 0.7 $m/z$ of isolation window, 100% normalized AGC target, and auto max-imum injection time. Fragmentation was performed by higher-energy collisional dissociation with a normalized collision energy of 30. The dynamic exclusion time was set to 60 s.

## Proteome data analysis

The MS raw files were searched to identify peptides by MaxQuant. To identify the novel isoforms, we customized the database. For LCL, we quantified isoforms by remapping the RNA-seq dataset of the GEU-VADIS (Genetic European Variation in Disease) project[8] (n = 463, EMBL-EBI, E-GEUV-1). For THP-1, we quantified isoforms by remapping the RNA-seq dataset derived from naïve THP-1 cells (n = 3, deposited in the GEO under the GSE157052). Considering sample size, we filtered out isoforms with minimum TPM for all samples = 0 and <2 for LCL and THP-1, respectively. In addition, to predict novel proteins, we deleted isoforms whose entire predicted ORFs were included in GENCODE. As a result, predicted ORFs encoded by 16,190 isoforms were retrieved as novel isoform candidates. Then, we constructed a non-redundant protein database by combining them with the Swiss-Prot database of human proteins including isoforms (42,360 entries, 2022_06) for the database search in this study.

For tryptic peptides, methionine oxidation and protein N-terminal acetylation were selected as variable modifications, and cysteine carbamidomethylation and peptide N-terminal and lysine TMTpro labels as fixed modifications. For V8 protease-digested peptides, methionine oxidation was selected as a variable modification and cysteine carbamidomethylation and peptide N-terminal and lysine TMTpro labels as fixed modifications. A maximum of two missed cleavages were allowed. False Discovery Rate (FDR) filtering by target-decoy method was set to 1% for both peptide-spectral match (PSM) and protein levels. Manual inspection was performed on the remaining MS/MS spectra and an Andromeda score >80 was set as an additional acceptance criterion. This corresponds to an FDR < 0.075% at the PSM level. Furthermore, the identified peptide sequences were checked against known protein sequences in Swiss-Prot and matches were excluded.

## Short-read RNA sequencing and processing

We utilized the datasets (single nucleotide polymorphism (SNP) array and RNA-seq data) of previous expression quantitative trait locus (eQTL) studies obtained from four Europeans cohorts: the EvoImmunoPop project[18] (European Genome-phenome Archive [EGA], EGAS00001001895), the DICE (database of immune cell expression, expression quantitative trait loci, and epigenomics) project[19] (the database of Genotypes and Phenotypes (dbGaP), phs001703.v1.p1), the Immune Variation (ImmVar) study[65], (dbGaP, phs000815.v1.p1), and the GEUVADIS project[8] (EMBL-EBI, E-GEUV-1).

We additionally performed genotype imputation using SNP array data. Pre-imputation quality control (QC) of the genotyping data was performed using PLINK[104] with the following parameters (--mind 0.02 --king-cutoff 0.0884 --geno 0.01 --maf 0.01 --hwe 1e-5). Post-QC variants were prephased using SHAPEIT[105], and imputation was performed using MiniMac3[106] and 1000 Genomes Phase 3 (release 5) as the reference panel[107]. Post-imputation QC was performed using PLINK with the following parameters (--minimac3-r2-filter 0.3). Genotyped and imputed SNPs or indels with minor allele frequency (MAF) ≥ 0.01 were used for subsequent QTL analysis with related expression datasets.

For RNA-seq, 3′ ends with low-quality bases (Phred quality score <20), and adaptor sequences were trimmed using Trim_Galore from sequenced reads. We realigned the trimmed reads on the GRCh38 genome using STAR[78] in two-pass mode with the de novo transcript annotations derived from long-read RNA-seq of 29 immune cell types.

Expression was quantified using StringTie2[108] and kallisto[109] independently using generated bam files from STAR and trimmed reads, respectively. For gene-level quantification, we combined all isoforms of a gene into a single transcript as described elsewhere[110]. Raw read counts were normalized with the Transcripts Per Kilobase Million (TPM) method[111].

## Cell-type-specific isoforms

After filtering out isoforms with low expression levels (reads per million, RPM > 2) and isoform ratios in each related gene (isoform ratio >0.2), cell-type-specific isoforms were identified based on their Shannon entropies using the ROKU[112] function in the TCC package[113].

## RBP binding analysis

To investigate the association between RBPs and specifically expressed isoforms, we searched RBP binding motifs in the sequence of the 3′-UTR of each isoform using RBPmap[114]. Then, the numbers of predicted binding sites for each RBP by region (5′-UTR, ORF, and 3′-UTR) were aggregated and compared between specifically expressed isoforms and others.

## Translational efficiency

We utilized RNA-seq and Ribo-seq datasets obtained from 52 common Yoruba individuals among the RNA-seq dataset derived from the GEUVADIS project[8] (EMBL-EBI, E-GEUV-1) and the Ribo-seq dataset deposited in GEO under GSE61742[115], respectively. To calculate translational efficiency at the isoform level, trimmed reads were aligned to the de novo transcriptome sequences generated from long-read sequencing for 29 immune cell subsets using STAR[78] as with tools developed for the same purpose[116,117], designed for transcript-level quantification with short-read data. We used the same STAR parameters as these tools to optimize for transcript quantification. As reported in the original paper[116,117], the method was tested using synthetic Ribo-seq reads with known profiles, demonstrating strong performance across various sequencing error rates and a strong Pearson correlation between footprint assignments and actual ribosome profiles. Then, we applied generated bam files to the *coverageDepth* and *translationalEfficiency* functions with corrections using the maximum translational efficiency value in the 90 most highly ribosome-occupied nucleotides window within the feature in *ribosomeProfilingQC* R package[118,119]. We calculated the translational efficiency only of isoforms that satisfied coverage depth >1 of both Ribo-seq and RNA-seq (86,967 isoforms) to avoid the potential over-estimating of translational efficiency due to low coverage.

To investigate the differences in cell types for translational efficiency, we remapped TrIP-seq data obtained from HEK293T (deposited in GEO under GSE69352[49]) on the GRCh38 genome using STAR[78] in two-pass mode with the de novo transcript annotations derived from long-read sequencing for 29 immune cell types. Transcript level abundances were calculated using StringTie2[108] and normalized by TPM method. Given the differences in cell types sequenced for our database and TrIP-seq data, we selected isoforms expressed in HEK293T as mRNA (TPM > 1 in all three samples of short-read RNA-seq data [SRR9019712, SRR9019713, and SRR9019714] deposited in GEO under GSE130781[120]). We then clustered isoforms and categorized as high polysomes and low polysomes by hierarchical clustering using polysome profiling as done in the previous study[49].

## Secondary structure

To estimate the presence of local RNA secondary structures, a window length of 25 nucleotides was moved at the step size of one nucleotide, starting from TSS to TTS, and the Gibbs free energy ($\Delta G$) was calculated as the predicted local folding strength for each window using the RNAfold program[121]. Then, we tested the correlation translational efficiency of a particular isoform and $\Delta G$ values at each scaled position relative to the TSS (0 = TSS, 1, 2, ..., 100 = first nucleotide of the ORF, 101, ..., 200 = last nucleotide of the ORF, 201, ..., 300 = TTS), and correlation coefficients were averaged over isoforms at each position.

To validate predicted local folding strength, we tested correlations with the Parallel Analysis of RNA structure (PARS) score[50,51]. To calculate PARS scores, we downloaded RNA fragments generated from

LCLs of a family trio with treatment by RNase V1 or S1 nuclease deposited in GEO under GSE50676[50] and mapped on the GRCh38 genome using STAR[78] in two-pass mode with the de novo transcript annotations derived from long-read sequencing for 29 immune cell types. Then, we quantified the number of double-stranded reads (V1) and single-stranded reads (S1) that were initiated on each base on the RNA. The read counts of double and single stranded reads for each sequencing sample were normalized by sequencing depth. For a particular isoform with $N$ bases in total, the PARS score of its $i$th base was defined by the following formula where V1 and S1 are normalized V1 and S1 scores, respectively. A small number (5) was added to reduce the potential over-estimating of structural signals of bases with low coverage:

$$PARS_{i=1...N} = \log_2(V1_i + 5) - \log_2(S1_i + 5) \qquad (1)$$

After removing the scaled position with all PARS scores being zero among three samples, we tested the correlation between the averaged PARS score and predicted local folding strength at each scaled position. We capped the PARS score to ± 7. We calculated PARS scores of only isoforms whose median TPM were the highest in each gene to minimize noise from multi-mapping. We quantified isoform expression values using LCL samples in the GEUVADIS cohort[8]. In total, PARS scores of 5800 isoforms were calculated.

### Isoform-switch analysis
We downloaded short-read RNA-seq datasets obtained from clinical patients and healthy controls from the GEO through accession numbers GSE72509 for SLE and GSE89408 and GSE118829 for RA. From raw sequenced reads, 3′ ends with low-quality bases (Phred quality score <20) and adaptor sequences were trimmed using Trim_Galore. An expression matrix was generated using kallisto[109] with default parameters and the de novo transcript annotations derived from long-read sequencing for 29 immune cell types. Then, we used IsoformSwitchAnalyzeR in the R package to detect genes that have changed splicing patterns[122]. Briefly, in IsoformSwitchAnalyzeR, isoform usage is assessed using isoform fraction (IF) values, which represent the proportion of a gene's total expression attributed to a particular isoform. The IF is computed by dividing the expression of an individual isoform by the sum of the expression of all isoforms for that gene (isoform_exp / gene_exp). To quantify the change in isoform usage between conditions, the difference in isoform fraction (dIF) is calculated by subtracting the IF of one condition from the other (IF2 - IF1). These dIF values serve as a measure of effect size, analogous to fold changes in gene or isoform expression analysis. To identify isoforms with statistically significant changes in usage, a dIF threshold of 0.1 and a FDR cutoff of 0.05 were applied. Statistical test was performed using the isoformSwitchTestDEXSeq function[123,124].

### QTL analysis
Independent QTL analysis was performed for each condition. In common with all QTL analyses, the following process was performed: (1) normalization of the expression matrix was performed with quantile normalization, rank-transformed normalization, and PEER normalization using 15 hidden factors for all QTL analyses[125], and (2) the variants with MAF ≥ 0.01 within a 1-megabase (Mb) window around each transcript using MatrixEQTL of the R package[126] with the top 10 genetic principal components as covariates.

The pipeline for sQTL analysis was the same as in gene-level eQTL analysis, except for the preparation of the expression matrix. A comparison of two representative quantification methods, (1) alignment-based transcript quantification and (2) alignment-free transcript quantification, showed a high correlation in expression values at the gene level, but a moderate correlation at the isoform level between the two methods (Supplementary Fig. 10). Therefore, we utilized the

independent isoform ratio in related genes derived from StringTie2[108] and kallisto[109], which are alignment-based transcript quantifications and alignment-free transcript quantifications for trQTL analysis, respectively. In addition, to capture differences in coding-sequence more sensitively, we used the isoform ratio quantified by kallisto after clustering the isoforms with completely identical coding-sequences by VSEARCH[127]. We also conducted junction-based sQTL analysis using LeafCutter[10].

With regard to 3′aQTL, we used dynamic analyses of APA from the RNA-seq (DaPars) algorithm[128,129] to identify APA events. The multisample DaPars v.2 regression framework calculates the percentage of the distal poly(A) site usage index (PDUI) value for each gene in each condition. Subsequently, we analyzed the association between variants within 1-Mb from the 3′-UTR region and quantile- and rank-normalized PDUI values with covariates, the same as with eQTL and sQTL analyses. To investigate the enriched function of genes with 3′aQTL effect, pathway analysis was performed using Enrichr[90].

### Stratified linkage disequilibrium score regression analysis (S-LDSC)
We extracted eQTL, sQTL, and 3′aQTL variants (SNPs or indels with nominal $p < 1×10^{-5}$) and performed S-LDSC[67] adjusting for functional annotation (baseline model v1.2 provided by the developers). Formatted GWAS summary statistics for S-LDSC by developers were downloaded from https://alkesgroup.broadinstitute.org/sumstats_formatted/.

### Colocalization analysis of QTL and GWAS
To evaluate the colocalization of QTL and GWAS signals, we applied a Bayesian framework using coloc of the R package[130]. We tested for a 500,000 bp window centered on the GWAS lead variant and considered PP-H4 (posterior probability of shared causal variant) >0.8 as a significant colocalization. Formatted GWAS summary statistics for S-LDSC by developers were downloaded from https://alkesgroup.broadinstitute.org/sumstats_formatted/ and severe COVID-19 from the COVID-19 HGI release5 (https://storage.googleapis.com/covid19-hg-public/20201215/results/20210107/COVID19_HGI_A2_ALL_eur_leave_23andme_20210107.b37.txt.gz). We used LocusZoom to visualize the colocalization[131].

### Mapping a simulated short-read RNA-seq to TRAILS
To examine the performance of mapping short-read RNA-seq to TRAILS, we generated a simulated pair-end short-read RNA sequencing dataset with assumption that all isoforms in our dataset have equal expression level (×5 coverage per transcript) using polyester R package (https://github.com/alyssafrazee/polyester). We tested isoforms which are 13 or less related isoforms transcribed from the same loci because isoforms with more related isoforms are rare and reduce the reliability of the analysis. We then grouped the isoforms according to the number of isoforms expressed from the same gene region and compared the expression values. We mapped to GRCh38 reference genome with TRAILS as reference annotation and other default options of STAR two-pass mode.

### Statistical test
The statistical tests performed are indicated in the figure legends or Methods.

### Reporting summary
Further information on research design is available in the Nature Portfolio Reporting Summary linked to this article.

## Data availability
Isoform expression data have been deposited in the DNA Data Bank of Japan (DDBJ) via the National Bioscience Database Center (NBDC) Human Database under accession code DRA016285. The MS raw data

and analysis files have been deposited in the ProteomeXchange Consortium via the jPOST partner repository with the data set identifier PXD040962.

## Code availability

We used publicly available software for the analyses. The source code and generated annotations are available at https://github.com/juninamo/TRAILS. Results in this study can also be browsed at our website at http://gfdweb.tmd.ac.jp:3838/.

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

## Acknowledgements

Computations were partially performed on the NIG supercomputer at the ROIS National Institute of Genetics. This study was supported by the Japan Society for the Promotion of Science (JSPS) Grant-in-Aids for JSPS Fellows (grant numbers: 21J00596 and 21J15131), Grant-in-Aids for Scientific Research (B) (grant numbers: 18H02849, 21H02459, and 22H02597), Grant-in-Aid for Scientific Research (S) (grant numbers: 18043381), and Grant-in-Aid for Challenging Research (grant number: 21K19501) from the MEXT of Japan, and grants from Nanken-Kyoten, TMDU and Medical Research Center Initiative for High Depth Omics. We thank K. Kobayashi for her technical assistance.

## Author contributions

J.I. conducted bioinformatics analysis with the help of M.T.U., K.Yamaguchi., and Y.Kochi, A.S. managed and contributed to sample collection,

cell sorting, and long-read RNA sequencing. H.N. and Y.I. contributed to proteome analysis. H.H. and K.I. performed quality control of the sample for long-read RNA-sequencing. K. Yamamoto, and Y.Kochi designed and managed the project. K.S., Y. Kaneko and T.T. provided the short-read RNA sequencing datasets for isoform switch analysis. J.I. and Y.Kochi wrote the manuscript with contributions from all authors to the final version of the manuscript.

## Competing interests

The authors declare no competing interests.
