## [Peer Review File · Nature Communications]

Long-read sequencing for 29 immune cell subsets reveals disease-linked isoformsREVIEWER COMMENTS

Reviewer #1 (Remarks to the Author):

Inamo and colleagues describe Immune Isoform Atlas, a catalog of full-length transcript isoforms in human immune cells based on nanopore long-read RNA-seq of 29 immune cell subsets. While long-read RNA-seq is an increasingly powerful and popular technology for transcriptome analysis, the present study has significant limitations with regard to its sample source, dataset, data analysis, and utility. As a result, the presented atlas will be of limited interest to the research community.

1. A major limitation of this atlas is that the entire dataset was derived from a single donor (a 42-year-old female). Given that age and sex differences are widespread in immune cell transcriptomes, a dataset based on a single donor is not representative. This is a serious problem that limits the value of this atlas as a resource.
2. The authors performed nanopore cDNA sequencing on the MinION platform, generating approximately 5.7M raw reads and 4.6M usable reads per immune cell subset. This sequencing coverage is quite modest for the purpose of isoform discovery, again limiting the value of this dataset and the resulting atlas.
3. The authors used FLAIR to analyze their data and discover known and novel transcript isoforms. Although FLAIR is a well-known tool for analyzing nanopore long-read RNA-seq data, its capability for discovering novel transcript isoforms in the absence of sample-matched short-read RNA-seq data is limited. Multiple recently published tools, such as IsoQuant (PMID: 36593406) and ESPRESSO (PMID: 36662851), report significantly better performance for isoform discovery and quantification compared to FLAIR and other existing tools.
4. The overlap of isoforms in this Immune Isoform Atlas with isoforms in other databases is strikingly low, e.g. 4.9% with GENCODE and 7.8% with isoforms found by long-read RNA-seq in the GM12878 cell line. How much of this discrepancy reflects true biological differences, versus potential technical artifacts including false positives in isoform discovery?
5. The authors report many read-through transcripts in the atlas, but do not provide experimental validation. Are these read-through transcripts real or could they reflect template-switching artifacts during nanopore cDNA sequencing? Do these read-through transcripts still show up in nanopore direct RNA-seq data of the same cell types?
6. Several applications of the Immune Isoform Atlas presented in this manuscript involve mapping short-read RNA-seq data from distinct sample sources to isoforms cataloged in this atlas. However, the unique value of this Immune Isoform Atlas is unclear in these settings. Would an isoform atlas built from long-read RNA-seq data of diverse human tissues (e.g. GTEx, see PMID: 35922509), which would also contain many novel isoforms, be equally or potentially more useful?
7. The authors report that shorter 5'-UTR and longer 3'-UTR are associated with higher translational efficiency (Figure 5B). This finding seems to contradict previous results based on polysome sequencing (PMID: 26735365).
8. The proteomics dataset analyzed in this work is from another manuscript in preparation. The authors need to make the raw MS proteomics dataset accessible as part of the present manuscript, as opposed to a separate manuscript whose publication timeline is unclear. As related to this issue, the authors also need to clarify the availability of raw and processed nanopore RNA-seq data in this work.

Reviewer #2 (Remarks to the Author):

Inamo et al. report the long reads sequencing of 29 immune cell types. They use Nanopore sequencing and provide an extensive analysis by integration with other existing datasets. They integrate ATAC-seq, Ribo-seq, RNA-seq, QTL, and proteomics data. They predict a large number of novel isoforms and show, the cell-type specific expression and the pathological relevance of some of them.

While this is a nice integrative study of the transcriptional landscape of the immune cell types, I am worried about the robustness of some of the presented analysis and their conclusions. Specifically, I am concerned with the reliability of the novel isoform prediction calls from the noisy Nanopore technology, the accuracy of RNA-seq-based quantification of genes with a high number of very similar isoforms, and some aspects of the translational efficiency analysis.

The study reports a very high number of novel isoforms, most of them NNC, but also a high number of novel genes (14%). Our work on the LRGASP project benchmarking long-reads methods indicates that analysis pipelines tend to report a very high number of novel transcripts that are not recapitulated by other analysis pipelines. This was also true for FLAIR, which had low accuracy for novel isoform detection. Also, we found that novel transcripts are more likely not to be experimentally validated than GENCODE-annotated transcripts and that Nanopore data is way noisier than Pacbio data. Additionally, it was also found that many novel isoforms are detected in a sample-specific manner and those novel isoforms that were experimentally validated were usually found by several analysis pipelines. Therefore, despite the presented analysis pipeline includes many checkpoints, it is not actually benchmarked, and the accuracy of the predicted transcript models is unknown. Since the claim for novel transcripts in this paper is strong, I think that more validation of the novel calls is required, ideally by amplification of isoform-specific amplicons. If not possible, I would suggest that data is analyzed by other algorithms (Bambu, IsoQuant, or StringTie2) and that only those novel transcripts detected by multiple methods are used to define the Immune Isoform Atlas.

If I understood it correctly, out of 16,190 predicted coding transcript isoforms, only 279 could be identified using proteomics. It is not clear if these were unique peptides or were also compatible with known proteins. According to methods only novel coding sequences predicted in this study and SwissProt isoforms were included as search space, but this does not preclude that the identified peptide could also be compatible with a known protein. Moreover, it is unclear if the FRD control (1%) imposed is over the peptides or the hits. If this is over the peptides, how many peptides were searched?

In line 143, it is unclear if the 1,022 loci predicted as non-coding in GENCODE are loci identified in this study or just GENCODE predictions. Also, figure 2D shows that, while GENCODE lncRNAs seem to have higher conservation scores at predicted ORFs, this is not the case for other novel genes. This aspect is not highlighted in the discussion and should be commented on.

Another claim of this section is that most of the novel genes were validated from CHES, however, no specific data is shown and also there is no control for specificity. Therefore, it is difficult to understand to which extent these new genes really represent novel bona fide genes or are just transcriptional noise. For example, how many sequences present in CHES are orphans? What is the probability that we are seeing here pervasive transcription? (b.t.w. pervasive transcription occurs at open chromatin region, therefore the ATAC-seq results could be reflecting just this). Are any of these novel genes found in more than one cell type? Are these novel genes spliced? Do they have gene signatures defined by Havana to call genes?

I feel that in order to call these novel expressed regions as genes, more consistent evidence should be found regarding reproducibility, gene architecture, etc. If this is not possible, I would call these regions expressed loci, but not novel genes.

In figure 4. Was the quantification obtained from long reads? Is this based on the long reads count or based on short reads? How is the TPM in supplementary table 5 calculated?

Extended data Figure 3A claims that results were equivalent to Figure 4, but this is not evident when looking at the figure. Please explain.

The analysis of translational efficiency was done by mapping existing RIBO-seq data to the long-read transcript models. How was mapping specificity controlled? Since this paper claims an average of 10 isoforms per gene, with most of them alternative donor and acceptor sites, it might be difficult to differentiate isoforms from any short-reads dataset. If reads are not specific for isoforms, conclusions of the properties of isoforms with high translational efficiency might be biased. The same holds for the IRAK1 gene, where more than 10 isoforms are present. How reliable is short-reads-based quantification of isoforms in a locus with so many isoforms?

Also in Figure 5. I do not think that from the correlation plots shown in figure 5B, it can be concluded that there is a correlation between translational efficiency and Kozak score, ORF length, and 3'UTR length as R2 values are very low. Although the p.value is significant, this is just the reflection of the high number of observations used in the analysis, but this is not enough to claim a relationship. The magnitude of the R-value must be considered and in these cases, the correlations are close to zero.

Extended data Figure 5 is inverted, and should be corrected.

Reviewer #3 (Remarks to the Author):

In their paper, Immune Isoform Atlas: Landscape of alternative splicing in human immune cells, Inamo and colleagues use long-read sequencing to generate a full-length isoform annotation of 29 human immune cell subsets, which they name the Immune Isoform Atlas. The majority (88%) of the identified isoforms had no sequences registered in GENCODE (version 38). They characterized different properties of the unannotated isoforms and showed that repetitive elements significantly explained their diversity. In addition, they used the Immune Isoform Atlas as a reference annotation to reanalyze existing gene expression data and perform alternative splicing, QTL mapping, and colocalization analysis of QTLs and GWAS data. They identify a number of sQTL variants that were missed when using GENCODE. Overall, the manuscript is well written, the statistical analyses are sound, and the results support most of the conclusions and claims. However, additional evidence is needed to support the usefulness of the atlas for disease transcriptomics and genomics. I list a few major and minor comments below

Major

1. How do you envision this atlas being used? Would it complement or substitute GENCODE, e.g., for disease transcriptomics and QTL mapping studies? The majority of isoforms in GENCODE are not identified in the Immune Isoform Atlas and many sQTLs are only identified when using GENCODE annotations. Should both annotations be used?
2. The paper would benefit from a few biological replicates. For example, in line 135 you mention that: "Of note, 1,365 (40%) loci encoded so-called read-through isoforms ... Although read-through isoforms are known to be transcribed in particular situations in cells, such as in malignancy and infection many read-through isoforms ... were also transcribed in cells under normal physiological conditions.". Could this be a consequence of profiling a single sample whose immune response might be triggered by an infection?
3. Line 266-270 and Figure 5B: these correlations are very low and seem to only be significant because of the large number of isoforms tested. Please report what correlation metric was used and how was the non-independence of isoforms accounted for (which would result in falsely significant p-values).
4. The SLE data set used for differential splicing analysis is too small and likely underpowered. The results do not make a strong case for the usefulness of the atlas for differential splicing analyses versus standard annotations. The manuscript would benefit from analyzing a larger data set for SLE

(e.g., <https://pubmed.ncbi.nlm.nih.gov/35389781/>) or other immune-related traits.

Minor

1. Line 48: add a citation for "Indeed, in the GTEx project, 23% of the GWAS loci were co-localized with sQTL." I assume it is from this one <https://pubmed.ncbi.nlm.nih.gov/33499903/>?
2. Line 128 – 132: Only a small number (278 of 16,190) of potentially novel coding isoforms seems to be confirmed with the proteomic analyses. Is this expected? How does the validation rate look for known protein-coding isoforms?
3. Line 189 and Figure 3D: Please add p-values for the statement: "Interestingly, the proportion of TEs was non-uniform; LTRs, which are autonomous and coding TEs, were enriched in TSSs and ORFs, while SINEs were enriched in 5'-UTRs and 3'-UTRs".
4. Line 241 and Figure 4D: "We comprehensively searched for RBP binding motifs and compared them between cell-type-specific isoforms and others." What are "other" isoforms? Is the enrichment significant after multiple testing adjustments?
5. Line 317 – 319 and 936-938:
 - o How many genes were tested for changes in splicing patterns?
 - o How many of the significant genes (84) were only discovered using the immune atlas and not the standard GENCODE annotation? It is currently unclear how useful the atlas is for identifying AS versus using GENCODE.
 - o Why are you using a .05 cutoff for dIF?
 - o You introduce RA data in the Isoform-switch analysis method section but do not report any results.

Dear Editors and Reviewers

Thank you very much for reviewing our manuscript and offering valuable advice.

We have addressed your comments with point-by-point responses and revised the manuscript accordingly. We appreciate that the comments greatly helped us to improve our manuscript with details of results and description.

Responses to the Comments by Reviewer #1:

1. A major limitation of this atlas is that the entire dataset was derived from a single donor (a 42-year-old female). Given that age and sex differences are widespread in immune cell transcriptomes, a dataset based on a single donor is not representative. This is a serious problem that limits the value of this atlas as a resource.

Reply:

Thank you for commenting on this important point. We agree that a single donor might not be a representative for the human transcriptomes. Among their diversity, we think the sex difference would be a serious problem because many immunological diseases display sex differences in epidemiology and pathology. To address this, we additionally collected PBMC from a 40-year-old male and applied long-read RNA sequencing using the latest ONT platform (PromethION R10.4.1, V14 chemistry). This dataset allowed us to compare the isoforms in PBMC between female and male. Meanwhile, it also enabled us to validate the presence of isoforms in Immune Isoform Atlas, owing to its higher quality of sequences, higher read depth, and comprehensive coverage of all the immune subsets by PBMC.

We mapped the raw reads with quality > Q15 and yielded 26,551,929 reads (MAPQ >10). The same FLAIR pipeline as used in this study generated 100,607 isoform annotations (male PBMC annotation). Next, we extracted isoforms in PBMC from the Immune Isoform Atlas (female PBMC annotation, 7,527 isoforms) and compared them with the male PBMC annotation using gffcompare¹. Notably, 85.0% (n= 6,399) of isoforms in female PBMC annotation were also discovered in the male PBMC annotation. Because we observed significantly higher expression levels of the shared isoforms compared to those non-shared (10.5 vs 3.9 of median Reads Per Million (RPM), P<0.001), we think we could capture even more isoforms by increasing the read-depth more.

Further, we validated 30.6% (n=48,757) of total isoforms in our Immune Isoform Atlas by male PBMC annotation. It should be mentioned that we sequenced into fine-grained 29 cell types so that we could detect cell-type specific transcripts (Fig 4). These transcripts may be diluted in PBMC and could not be detected in the male PBMC sample. Although we have validated the presence of remaining transcripts in our Atlas (69.4%) by short-read junctions, the validation status of isoforms also found in male PBMC is higher. We therefore added a label to these isoforms indicating that they have been validated ("isoform_info.txt.gz" in https://github.com/juninamo/isoform_atlas/tree/main).

These results suggest that differences in cell-types, rather than sex and/or individual, is a major driver for diversity of alternative splicing in immune cells. This is consistent with the previous study demonstrating that a large proportion of alternative splicing is tissue-specific²⁻⁵, and that heterogeneity from inter-individual is only <30%³. To further validate this idea in our dataset, we remapped short-read RNA sequencing obtained from 15 immune cell types from European individuals (1,140 samples from 76 individuals) to our Immune Isoform Atlas⁶. Next, we calculated percent-spliced-in (PSI), which is an objective metric for differential splicing⁷, for each junction (n=139,110) and sample. We then calculated the proportion of variance derived from both cell types and individuals out of the total variance of PSI. As a result, the variance of cell types was 6-fold higher than that of individuals (median 7.4% vs 1.2%, p<0.001 by Wilcoxon test). If we sequence 29 immune cell types from multiple individuals with a sample size comparable to the recent large-scale consortium for long-read sequencing using human samples (~100 samples for the ONT platform^{8,9}), it would be only ~three individuals, making it difficult to distinguish true diversity within individuals from artificial noise. This tradeoff with respect to sample size (*individuals* or *cell types*) could be overcome if the cost of analysis using long-read sequencing decreases and becomes more accessible. For now, we believe it is valuable to examine the heterogeneity of alternative splicing within immune cell types, even if those are from the same individual, given that immune cells exhibit cell-type-specific functions even when derived from the same individual.

In all, we focused on diversity of regulation of splicing within immune cell types in the current study. However, as pointed out by the reviewer, inter-individual variation (as well as changes along with aging) should not be overlooked and need to be addressed in the future. Therefore, we mentioned it as a limitation in the Result and Discussion. In addition, we deposited the annotation using male PBMC sample in our Github repository (https://github.com/juninamo/isoform_atlas/blob/main/male_PBMC.gtf.gz).

#####

Extended Data Fig 5E. Proportion of variance of percent-spliced-in (PSI), isoform expression, and isoform ratio explained by differences in cell types and individuals. The expression value was normalized with the Transcripts Per Kilobase Million (TPM).

#####

Line 94; *As a validation, we additionally sequenced the transcripts of PBMC from a 40-year-old healthy male using the latest PromethION platform (Methods). We validated 85.0% (n= 6,399) of isoforms expressed in PBMC used in the Immune Isoform Atlas. We also validated 30.6% (n=48,757) of total isoforms in all cell types. The difference in proportions between PBMCs and all cell types is likely to reflect the dilution of transcripts in PBMC that are expressed in a cell-type-specific manner.*

Line 281; *The previous study demonstrated that the alternative splicing diversity between cell types was greater than that between individuals². We investigated this point by remapping short-read RNA-seq data from 15 immune cell types¹⁹ using our Atlas; we observed that the variance of alternative splicing between individuals was less than a quarter of that observed between cell types (Extended data Figure 5E).*

Line 484; *Although a significant portion of the diversity of alternative splicing is attributable to cell- or tissue-specific splicing^{2,74-76}, it is important to consider the diversity based on biological characteristics such as age, sex, and health status. In the future, advancements in long-read sequencing technology are expected to enable better exploration and understanding of these factors.*

2. The authors performed nanopore cDNA sequencing on the MinION platform, generating approximately 5.7M raw reads and 4.6M usable reads per immune cell subset. This sequencing coverage is quite modest for the purpose of isoform discovery, again limiting the value of this dataset and the resulting atlas.

Reply:

We thank for this essential comment. The long-read sequencing technology tends to have fewer reads compared to short-read sequencing since the former has longer length per read. The median raw reads of GTEx long-read data per sample is approximately 6.3M pre-alignment and 4.9M post-alignment, which is similar to our data⁸. This number of reads in our dataset, when converted to reads with 75bp short-read sequencing, corresponds to approximately 106M reads ($\approx 4,566,622$ [number of mapped reads] \times 1,752 bps [median read length] / 75 bps). This number far exceeds the number of reads typically required for short-read sequencing of human samples¹⁰.

To investigate the power to detect isoforms in our dataset, we conducted sensitivity analysis by calculating total number of isoforms by down-sampling reads for each cell type. Since the power is expected to depend on expression levels for each transcript as discussed in the comment 1, we separately analyzed highly and lowly expressed transcripts by cell types. Consequently, we found that the curve was convex for transcripts with high expression, suggesting that the number of reads in our dataset was near to be sufficient. In contrast, the saturation curves of transcripts with low expression levels still show an upward trend, suggesting that additional transcripts can be found by increasing the number of reads. This is consistent with the result that the male PBMC samples sequenced using PromethION (as presented for the comment 1)

yielded more reads and identified more transcripts. We added this sensitivity analysis in Extended Data Fig. 2 and described it in Results.

#####

Extended Data Fig 2. Power analysis by downsampling reads. X-axis represents the proportion of downsampling reads. Y-axis represents the proportion of the number of identified isoforms when compared to using all reads. The line type represents high-expressed (Reads Per Million ≥ 2 in long-read RNA sequencing data [right] and top 25%ile of minimum junction coverage obtained by remapping short-read RNA sequencing data^{6,11–14} [left], solid) and low-expressed (dashed) isoforms in each cell type, respectively.

#####

Line 106; *To assess the capability of our dataset in discovering isoforms, we conducted a sensitivity analysis by downsampling reads. Our results revealed that our dataset exhibited sufficient power in detecting transcripts with high expression (Extended Data Figure 2). However, we observed that increasing the number of reads would enable the identification of additional transcripts with low expression.*

3. The authors used FLAIR to analyze their data and discover known and novel transcript isoforms. Although FLAIR is a well-known tool for analyzing nanopore long-read RNA-seq data, its capability for discovering novel transcript isoforms in the absence of sample-matched short-read RNA-seq data is limited. Multiple recently published tools, such as IsoQuant (PMID: 36593406) and ESPRESSO (PMID: 36662851), report significantly better performance for isoform discovery and quantification compared to FLAIR and other existing tools.

Reply:

Since IsoQuant was released after the submission of this paper, we initially generated annotations using FLAIR, which has been used in recent papers and successfully discovered biologically significant transcripts in the immune system and cancers^{8,15-17}. However, as the reviewer pointed out, it is important to know how much difference in the outputs between different tools, so we additionally created the annotation using IsoQuant and compared it to our Immune Isoform Atlas. **Our comparison differed from the original paper of IsoQuant in that we utilized short-read RNAseq data obtained from the corresponding cell lineage^{6,11-14} for junction correction as implemented in the FLAIR pipeline, which substantially improved the filtering of the noise¹⁸.** Further, we confirmed the transcription start sites for all isoforms by independent CAGE-seq datasets by incorporating them in the FLAIR pipeline, which allowed us to discover high confident transcripts.

We observed that 43,872, 115,497, and 444,667 isoforms were common in both the FLAIR pipeline (Immune Isoform Atlas) and the IsoQuant pipeline, only detected by FLAIR and IsoQuant, respectively. Importantly, we found that IsoQuant is conservative to detect isoforms with intron-retention (IR) compared to FLAIR (2.7% vs 27.0%, $p < 0.001$; figure below). We validated this point using independent long-read RNA sequencing data from lymphoblastoid cell lines (LCL) utilizing isoSeq3 (PacBio) (Figure below; Ueda M et al, manuscript in preparation). IsoSeq (PacBio) yields more accurate sequences ($>QD50$) and thus more confident isoforms than nanopore technologies. Our comparison with IsoSeq3 (+PacBio) data validated IsoQuant (+Nanopore) tended to filter out IR isoforms as a noise (figure in next page). Because IR is involved in various biological contexts and pathomechanism¹⁹⁻²¹, this point may be critical when investigating pathogenic alternative splicing. Importantly, the majority of isoforms ($>70\%$) in the IsoQuant-specific isoforms are mono-exon, suggesting that most of them could be non-coding RNAs. We observed the same trend when comparing our Atlas and another annotation made by ESPRESSO (IR isoform ratio; 27.1 vs 13.0%). We found that the performance of ESPRESSO to detect IR isoforms was unchanged (13.6%) if junctions were corrected (--SJ_bed) using the same short-read RNA-seq datasets for our Atlas, suggesting that FLAIR's unique junction correction algorithm allowed us to identify IR isoforms with high sensitivity. Notably, ESPRESSO identified more than two-fold isoforms compared with FLAIR, suggesting FLAIR may be more conservative for isoform annotation.

Based on these findings, we decided to focus on coding RNAs (including those with IR) and therefore remained to use the FLAIR pipeline. However, for reference of readers, we deposited the annotation by the IsoQuant pipeline (https://github.com/juninamo/isoform_atlas/tree/main/IsoQuant.transcript_models.gtf.gz) and ESPRESSO pipeline (https://github.com/juninamo/isoform_atlas/blob/main/Espresso.gtf.gz) in our Github repository to allow them to use more confident isoforms identified by multiple pipelines.

#####

Venn diagram showing overlap of isoforms identified by the FLAIR pipeline and IsoQuant pipeline.

The long-read annotations obtained from independent long-read sequencing data using lymphoblastoid cell lines of STAT2 locus are shown, with red boxes highlighting the areas where intron retention events were filtered out by IsoQuant. Yellow colored annotations on the bottom are from GENCODE.

Another point raised by this reviewer is the usage of sample-matched short-read RNA-seq data for junction correction. We obtained short-read RNA-seq data of 29 immune cell types from the same individual which have been sequenced in another ongoing project (Suzuki A et al.). We added the data to those originally used in the construction of Immune Isoform Atlas (short-read RNA-seq datasets of 30 immune cell states from 5 cohorts)^{6,11-14} to correct splice junction. As a result, we observed that most of the isoforms (96.6%, n=156,970) were already found in the

original Immune Isoform Atlas. In contrast, when we used only the sample-matched short-read RNA-seq dataset for junction correction, 17% of total isoforms in our atlas were missed. One reason for this might be the strict filtering criterion that the splicing junctions should be validated by 3 or more reads in short-read RNA-seq datasets. This criterion may miss relatively low-expressed transcripts, correspondingly with a low number of junction reads, when using only sample-matched short-read RNA-seq datasets. Based on these results, we considered it reasonable to use the original Immune Isoform Atlas with the FLAIR pipeline validated by the large dataset from multiple individuals for downstream analysis.

4. The overlap of isoforms in this Immune Isoform Atlas with isoforms in other databases is strikingly low, e.g. 4.9% with GENCODE and 7.8% with isoforms found by long-read RNA-seq in the GM12878 cell line. How much of this discrepancy reflects true biological differences, versus potential technical artifacts including false positives in isoform discovery?

Reply:

We agree that the overlap of isoforms in our Immune Isoform Atlas and other databases is low. However, it should be noted that overlap of isoforms in the long-read data by GTEx was similarly low to that from GM12878 cell line (13.1%)⁸.

One reason for this is that cell type specificity is one of the major drivers for alternative splicing, as discussed in reply to comment 1. Our long-read sequencing data was collected from 29 primary immune cell types, while GM12878 cell line was lymphoblastoid cells, which are infected with EB-virus, so that it has different cell status from our 29 primary immune cell types. Further, when comparing with annotation from GM12878 cell line, we applied conservative method by specifying the "--strict-match" option using gffcompare, which only allows a limited variation of the outer coordinates of the terminal exons by at most 100 bases.

We consider that the novel transcripts included only in our Immune Isoform Atlas compared to the annotation from GENCODE version 38 and LCL reflects biological differences and rather than noise, because larger proportion of isoforms were validated by the independent male PBMC sample as discussed in the comment 1 above. In addition, the sQTL analysis demonstrated that isoforms included only in the Immune Isoform Atlas enriched in IMDs-GWAS variants equally or more than those included in GENCODE version 38 (**Figure 7B**). If our Immune Isoform Atlas contained more false isoforms (technical artifacts) than existing annotation, sQTL signals would have a lower association with disease risk variants. Further, we observed that Immune Isoform Atlas contained more reliable isoforms annotated by GENCODE version 38 than less reliable isoforms, supporting the reliability of our atlas (**Extended Data Figure 1D**).

5. The authors report many read-through transcripts in the atlas, but do not provide experimental validation. Are these read-through transcripts real or could they reflect template-switching artifacts during nanopore cDNA sequencing? Do these read-through transcripts still show up in nanopore direct RNA-seq data of the same cell types?

Reply:

The isoforms of Immune Isoform Atlas were validated by junction reads obtained from short-read RNA-seq. This means that the presence of junctions unique to the read-through transcripts has been validated, and that transcripts due to artifacts such as template-switches have been theoretically removed. Further, our group validated the existence of read-through transcript from the *IFNAR2-IL10RB* locus and examined the role of its protein product, by biological experiments, in the severity of COVID19²² (*Immunity* under review).

However, as the reviewer pointed out, it is also important to confirm expression of read-through isoforms using direct RNA, so we investigated their expression using publicly available nanopore direct RNA sequencing data obtained from the LCL (GM12878)¹⁷. First, **we confirmed the existence of representative read-through isoforms we mentioned in the main text, such as *TOMM40-APOE* and *IFNAR2-IL10RB* (Extended data Figure 3)**. Then, we quantified all read-through isoforms of our Atlas in this direct RNA-seq data using “flair – quantify”. We applied the strictest mode by specifying both “*check splice site*” and “*stringent*” options. Among the read-through loci identified in 29 cell types (n=1,365) in our Immune Isoform Atlas, we validated that 437 loci are truly expressed in LCL. Further, among the read-through transcripts identified in female PBMC (n=392), **74.2% (n=291) were validated by the male PBMC** (the class code was “=” by gffcompare). Together, these data confirmed the presence of read-through transcripts in our Atlas.

#####

Extended data Figure 3. Validation of read-through isoforms using direct RNA sequencing¹⁷. Validated read-through isoforms transcribed from *IFNAR2-IL10RB* region (left) and *TOMM40-APOE* region (right). The read-through isoforms are highlighted by red.

#####

Line 157; *Using an independent PBMC sample from a male individual (Methods), we verified the expression of 74.2% (291 out of a total of 392) read-through transcripts expressed in female PBMC used for our atlas, including the transcript of TOMM40_APOE. To rule out the possibility of artifacts in the sequencing, we further investigated their expressions using direct RNA sequencing data obtained from the LCL (GM12878)²⁴. Among the read-through loci identified in 29 cell types (n=1,365) in our Immune Isoform Atlas, we validated that 437 loci were truly*

expressed in LCL, including those from TOMM40_APOE and IFNAR2_IL10RB (Extended data Figure 3).

6. Several applications of the Immune Isoform Atlas presented in this manuscript involve mapping short-read RNA-seq data from distinct sample sources to isoforms cataloged in this atlas. However, the unique value of this Immune Isoform Atlas is unclear in these settings. Would an isoform atlas built from long-read RNA-seq data of diverse human tissues (e.g. GTEx, see PMID: 35922509), which would also contain many novel isoforms, be equally or potentially more useful?

Reply:

Thank you for this important point. Existing annotations generated using long-read RNA sequencing, such as GTEx, are for tissue levels (e.g., whole blood, lymph nodes, etc.), which are substantially influenced by the heterogeneous distribution of cell types. This heterogeneity would substantially influence the fine mapping of eQTL/sQTL as discussed in the previous study²³. We consider it is essential to obtain the annotation using fine-grained immune cell types given their diverse phenotypes. As our Immune Isoform Atlas captured many (n=2,575) transcripts specific to each immune cell type with unique functional annotation such as domains, our dataset as annotation can be used for various types of analyses using short-read RNA-seq data by remapping them and following downstream analysis (e.g., differentially expressed transcripts, isoform switch analysis, and QTL), which would provide unknown roles of immune cells in the pathogenesis of IMDs.

7. The authors report that shorter 5'-UTR and longer 3'-UTR are associated with higher translational efficiency (Figure 5B). This finding seems to contradict previous results based on polysome sequencing (PMID: 26735365).

Reply:

Thank you for letting us know this important paper (hereafter, "Floor's paper"). The data presented in the paper are really suggestive for us to interpret our findings. We took the contradictions that you raised seriously and considered possible reasons.

First, the length of the 5'-UTR influences the navigation of ribosomes to the start codon through a secondary structure, while the length of the 3'-UTR affects mRNA stability and localization^{24,25}. Previous studies used existing annotations such as GENCODE or transcript structures "inferred" by computational algorithms such as Cufflinks. The latter methods would have difficulties in inferring 3'-UTRs that contain many repetitive structures (**Figure 3D**). In contrast, the strength of our analysis compared with previous reports is that we could capture the whole structure of transcripts by using long-read sequencing. As for the translation efficiency, we observed concordant associations with the Floor's paper in the features of 5'-UTR and ORF, which have less repetitive structures (**Extended data Figure 6A**). Further, as another supporting finding of our analysis, the observation for Kozak context score was also consistent with another previous report²⁶.

In addition to the effect of 3'-UTR length on translation, that of AU-rich element in the 3'-UTR also showed an inverse correlation with the Floor's paper (**Extended data Figure 6A**). As transcripts with longer 3'-UTR lengths had a higher occupation rate of AU-rich elements in the 3'-UTR (**Figure 5D**), their association with translational efficiency may be attributable to these AU-rich elements; indeed, RNA-binding proteins such as HuR have shown to provide mRNA stability by binding to the AU-element in 3'-UTR²⁷. In support of this hypothesis, our cell type-specific transcripts had longer 3'-UTR lengths and showed a significant enrichment of HuR-binding motifs (**Figure 4C-D**). Further, we confirmed *HuR* is ubiquitously expressed in immune cell types and is most abundantly expressed in lymphoblastoid cells (**Extended data Figure 6C,D**; note that the lower expression in whole blood is due to lack of expression in neutrophils). Therefore, the positive association between 3'-UTR length and translation efficiency observed in our dataset might be largely influenced by the HuR binding to isoforms. This cell-type specific regulation of 3'-UTR was also mentioned in the Floor's paper; the effect of 3'-UTR on the translation was more variable among different cell types than 5'-UTR (Figure 6 and Discussion in the Floor's paper).

To further investigate cell-type specificity of the effect of 3'-UTR features, we remapped TriP-seq data obtained from HEK293T in the Floor's paper to our Atlas (given the difference in cell types, we selected isoforms expressed in HEK293T with TPM > 1). We first investigated meta-transcript distribution of polysomes by isoforms' translational efficiency as we ranked in Figure 5. We confirmed the top 10% isoforms had significantly more polysomes than those of bottom 10% (**Extended data Figure 6E**). We then clustered isoforms by polysome profiling and compared transcript features between isoforms with high polysomes and low polysomes as the same as in the Floor's paper (**Extended data Figure 6F**). Notably, we observed a concordant direction of 3'-UTR features (length and AU-rich elements) with the results in the Floor's paper but a discordant direction with those obtained in our original analysis (**Extended data Figure 6A and 6G**). This may indicate differences in the translation machinery between the cell types examined (LCL vs HEK293T), such as the expression levels of RNA-binding proteins like HuR.

In summary, we think both the underestimation of 3'-UTR sequences due to repetitive elements as well as the cell-type difference may be two major reasons for the contradictions. We considered these discussions to be biologically important. However, as we did not perform any experimental validation, we only added these points in the Result and Discussion section.

#####

Figure 5D, Association between 3'-UTR length and AU-element fraction. 3'-UTR length is binned into three groups (short, moderate, and long).

Extended data Figure 6. Effect of transcript features on translational efficiency. A, Meta-transcript distributions of correlation coefficients for transcript features evaluated for translational efficiency in lymphoblastoid cell lines (LCL). Positive coefficients represent positive association for translational efficiency and error bars are bootstrapped 95% confidence intervals. All correlation tests are conducted by Spearman's correlation test. **B,** Correlation between 3'-UTR GC contents and AU-element fraction. **C,** Expression of HuR (*ELAVL1*) across 29 immune cell types in Immune Isoform Atlas. **D,** Expression of HuR (*ELAVL1*) across human tissue in GTEx. **E,** Meta-transcript distributions of polysome profiles in HEK293T for isoforms (isoforms in the top 10%ile are more efficiently translated) by remapping TriP-seq data obtained. P-value was obtained by comparing two different distributions by Kolmogorov-Smirnov test. **F,** Meta-transcript

polysome distributions for four isoform clusters by hierarchical clustering using isoform polysome profiles. G, Meta-transcript distributions of odds ratio comparing isoforms with high polysomes (cluster 3 and 4 in panel F) and those with low polysomes (cluster 2) in HEK293T for transcript features. H, Correlation between Parallel Analysis of RNA structure (PARS) scores and predicted local folding strength of isoforms.

#####

Line 288;

*The efficiency of protein synthesis is governed by the regulatory elements in the 5'-UTR, ORF, and 3'-UTR^{5,45,46}. As a classic example, a strong Kozak sequence immediately before the first codon improves start codon recognition as a feature of highly translated mRNAs⁴⁵. In addition, the secondary structure of mRNA may block or recruit ribosomes and other regulatory factors to enable a rapid, dynamic response to diverse cellular conditions⁵. **The drawback of these studies is that their analysis is based on gene annotation inferred from short-read RNA-sequencing, which introduces uncertainty regarding complex gene regions such as 3'-UTR where abundant repetitive elements are inserted (Figure 3D).** Therefore, we speculated that examining the translational efficiency of each transcript in the Immune Isoform Atlas would provide additional insights into the regulation of translation and bridge the knowledge gap between transcripts and proteins. For this purpose, we remapped the Ribo-seq and RNA-seq data **obtained from LCL** to the Immune Isoform Atlas, respectively, and calculated the scores of translational efficiency at the isoform level (**Methods; Figure 5A**).*

*As expected, isoforms with a strong Kozak context score had higher translation efficiency scores (**Figure 5B, Extended data Figure 6A**). Interestingly, the association between translational efficiency and the lengths of the 5'- and 3'-UTRs was reversed; i.e., shorter 5'-UTRs, as well as longer 3'-UTRs, were associated with higher translation efficiency (**Figure 5B**). **To explore the underlying mechanism of association between the length of 3'-UTR length and translational efficiency, we investigated the sequence characteristic and found that transcripts with high translational efficiency had more AU-rich elements (Extended data Figure 6A).** Further, transcripts with longer 3'-UTR lengths have a higher occupancy of AU-rich elements in the 3'-UTR (**Figure 5D**). It is known that binding of HuR (ELAVL1) to AU-rich elements in 3'-UTR prevents degradation of mRNA⁴⁷. Thus, HuR could promote translation efficiency via enhancing mRNA stability by binding to transcripts that contain AU-rich elements in the 3'-UTR. Such a regulatory mechanism of translation efficiency may be characteristic of immune cells, since HuR is expressed ubiquitously in 29 immune cell types in Immune Isoform Atlas (**Extended data Figure 6C**) and most abundantly expressed in LCL across human tissues (**Extended data Figure 6D**)⁷.*

*To further investigate cell-type specificity of the effect of 3'-UTR features, we remapped Transcript Isoforms in Polysomes sequencing (TriP-seq) data⁴⁸ to our Immune Isoform Atlas (**Methods**). We confirmed a concordant distribution of polysomes in HEK293T compared to the translational efficiency based on Ribo-seq data from LCL (**Extended data Figure 6E**). However, when comparing isoform clusters with high and low polysomes (**Extended data Figure 6F**), we observed opposite effects of 3'-UTR features (length and AU-rich elements) between HEK293T and LCL (**Extended data Figure 6A and G**).*

Line 489; *We found significant associations between the features of transcripts and translational efficiency. The strength of our analysis compared with previous reports is capturing the whole structure of transcripts using long-read sequencing. Previous reports have predominantly relied on annotations derived from short-read RNA-seq or transcript structures inferred from mapped short-reads using computational algorithms⁴⁰. These traditional methods make it difficult to infer whole transcript structures, especially in 3'-UTRs that contain many repetitive structures as we showed. We addressed this point by utilizing long-read sequencing and observed significant features relating to translational efficiency. Corresponding with the previous study⁴⁸, we observed differences in cell types as to the effect of 3'-UTR features on translational efficiency, whereas those of 5'-UTR were consistent. This suggests that different cell types have different mechanisms for controlling translation, such as binding of RNA-binding proteins like HuR to 3'-UTR. It should be noted that our analysis reveals a transcriptome-wide trend regarding the role of each transcript feature. However, experimental validation for individual transcripts will be necessary in the future.*

8. The proteomics dataset analyzed in this work is from another manuscript in preparation. The authors need to make the raw MS proteomics dataset accessible as part of the present manuscript, as opposed to a separate manuscript whose publication timeline is unclear. As related to this issue, the authors also need to clarify the availability of raw and processed nanopore RNA-seq data in this work.

Reply:

Our nanopore RNA-seq dataset, which was in process at the time of submission, is now available for publication because the procedures for its publication have been completed (ID: DRA016285 [BioProject: PRJDB15836]). The MS dataset will also be made public in this paper (ID: PXD040962, URL: <https://repository.jpostdb.org/preview/24710522664423f12ecda7>, Pass: 1430)

Responses to the Comments by Reviewer #2:

The study reports a very high number of novel isoforms, most of them NNC, but also a high number of novel genes (14%). Our work on the LRGASP project benchmarking long-reads methods indicates that analysis pipelines tend to report a very high number of novel transcripts that are not recapitulated by other analysis pipelines. This was also true for FLAIR, which had low accuracy for novel isoform detection. Also, we found that novel transcripts are more likely not to be experimentally validated than GENCODE-annotated transcripts and that Nanopore data is way noisier than Pacbio data. Additionally, it was also found that many novel isoforms are detected in a sample-specific manner and those novel isoforms that were experimentally validated were usually found by several analysis pipelines. Therefore, despite the presented analysis pipeline includes many checkpoints, it is not actually benchmarked, and the accuracy of the predicted transcript models is unknown. Since the claim for novel transcripts in this paper

is strong, I think that more validation of the novel calls is required, ideally by amplification of isoform-specific amplicons. If not possible, I would suggest that data is analyzed by other algorithms (Bambu, IsoQuant, or StringTie2) and that only those novel transcripts detected by multiple methods are used to define the Immune Isoform Atlas.

Reply:

Thank you for this helpful comment. As this point was also raised by the reviewer 1, we compared the FLAIR pipeline to the latest analytical tools IsoQuant and ESPRESSO (these tools have been shown to outperform older tools such as Bambu and StringTie2). In addition, we confirmed the presence of transcripts in our Atlas by using additional dataset; we obtained PBMC from a 40-year-old male and applied long-read RNA sequencing using the latest ONT platform (PromethION R10.4.1, V14 chemistry). This dataset allowed us to validate the presence of isoforms in Immune Isoform Atlas, owing to its higher quality of sequences, higher read depth, and comprehensive coverage of all the immune-subsets by PBMC.

First, we found that IsoQuant is conservative to detect isoforms with intron-retention (IR) compared to FLAIR (2.7% vs 27.0%, $p < 0.001$). We validated this point using independent long-read RNA sequencing data from lymphoblastoid cell lines (LCL) utilizing PacBio and isoSeq3, where ~20% of IR isoforms were detected with $>QD50$ (Ueda M et al, manuscript in preparation). These indicated that IsoQuant tended to filter out IR isoforms as a noise, and this can be critical given that IR is involved in various biological contexts and pathomechanism¹⁹⁻²¹. In addition, the majority of isoforms ($>70\%$) in the IsoQuant-specific isoforms are mono-exon, suggesting that most of them could be non-coding RNAs.

We evaluated another tool, ESPRESSO, and found that its performance to detect IR isoforms was similarly low (13.6%) even if junctions were corrected using the same short-read RNA-seq datasets for our Atlas. Notably, ESPRESSO identified more than two-fold isoforms compared with FLAIR, suggesting FLAIR may be more conservative for isoform annotation. Based on these findings, we decided to focus on coding RNAs (including those with IR) and therefore remained to use the FLAIR pipeline. However, for reference of readers, we deposited the annotation by the IsoQuant pipeline (https://github.com/juninamo/isoform_atlas/tree/main/IsoQuant.transcript_models.gtf.gz) and ESPRESSO pipeline (https://github.com/juninamo/isoform_atlas/blob/main/Espresso.gtf.gz) in our Github repository to allow them to use more confident isoforms identified by multiple pipelines.

Next, we mapped 26,551,929 reads from male PBMC with quality $> Q15$ and MAPQ > 10 obtained by PromethION (R10.4.1, V14 chemistry). The same FLAIR pipeline as used in this study generated 100,607 isoform annotations (male PBMC annotation). We extracted isoforms in PBMC from the Immune Isoform Atlas (female PBMC annotation, 7,527 isoforms) and compared them with the male PBMC annotation using gffcompare¹. Notably, 85.0% of isoforms in female PBMC annotation were also discovered in the male PBMC annotation. We believe we were able to identify transcripts more reliably by incorporating independent CAGE-seq datasets into the FLAIR pipeline, which allowed us to discover high-confidence transcripts. Further, we validated 30.6% ($n=48,757$ including 34,838 novel transcripts) of total isoforms in our Immune Isoform Atlas by male PBMC annotation (we added a label to these isoforms indicating that they have been validated; "isoform_info.txt.gz" in

https://github.com/juninamo/isoform_atlas/tree/main). It should be mentioned that we sequenced into fine-grained 29 cell types so that we could detect cell-type specific transcripts as shown in Figure 4. These transcripts may be diluted in PBMC and could not be detected in the male PBMC sample.

Finally, our claim for novel transcripts might be emphasized on the read-through transcripts (including *TOMM40-APOE* in the abstract), and we also validated expression of read-throughs. Because these transcripts might be artifacts of our cDNA-seq (e.g. template switching), we confirmed their existence using nanopore direct RNA-seq data obtained from the LCL (GM12878)¹⁷. Among the read-through loci (n=1,365) in our Immune Isoform Atlas, we validated that 437 loci including *TOMM40-APOE* locus are truly expressed in LCL. Further, our group experimentally examined the role of read-through transcript from *IFNAR2-IL10RB* locus and clarified its molecular function in the severity of COVID19²² (*Immunity*, under review). We believe these results support our pipeline for isoform discovery.

#####

Venn diagram showing overlap of isoforms identified by the FLAIR pipeline and IsoQuant pipeline.

Nanopore + FLAIR
 Nanopore + IsoQuant
 PacBio + Isoseq

The long-read annotations obtained from independent long-read sequencing data using lymphoblastoid cell lines of STAT2 locus are shown, with red boxes highlighting the areas where intron retention events were filtered out by IsoQuant. Yellow colored annotations on the bottom are from GENCODE.

Extended data Figure 3. Validation of read-through isoforms using direct RNA sequencing¹⁷. Validated read-through isoforms transcribed from *IFNAR2-IL10RB* region (left) and *TOMM40-APOE* region (right). The read-through isoforms are highlighted by red.

#####

If I understood it correctly, out of 16,190 predicted coding transcript isoforms, only 279 could be identified using proteomics. It is not clear if these were unique peptides or were also compatible with known proteins. According to methods only novel coding sequences predicted in this study and SwissProt isoforms were included as search space, but this does not preclude that the identified peptide could also be compatible with a known protein. Moreover, it is unclear if the FRD control (1%) imposed is over the peptides or the hits. If this is over the peptides, how many peptides were searched?

Reply:

The predicted coding transcripts identified by proteomics are 276 unique isoforms. There is no redundancy with known proteins. This is because after searching the database, we used three filters as described below: the first filter applies the FDR1% threshold by target-decoy search method. This was done at the spectral level (PSM level) and at the protein level. For the remaining identified spectra, we applied the second filter, a threshold of 80 Andromeda score. This was set based on visual confirmation of the MS/MS spectra. This resulted in an FDR (PSM level) of 0.075%. As a third filter, we checked whether the identified peptide sequences were included in the Swiss-Prot DB, which contains isoforms, along with the protease cleavage specificity, and excluded peptide sequences included in the Swiss-Prot DB. As a result, 279 unique isoforms survived.

In line 143, it is unclear if the 1,022 loci predicted as non-coding in GENCODE are loci identified in this study or just GENCODE predictions. Also, figure 2D shows that, while GENCODE lncRNAs seem to have higher conservation scores at predicted ORFs, this is not the case for other novel genes. This aspect is not highlighted in the discussion and should be commented on. Another claim of this section is that most of the novel genes were validated from CHES, however, no specific data is shown and also there is no control for specificity. Therefore, it is difficult to understand to which extent these new genes really represent novel bona fide genes or are just transcriptional noise. For example, how many sequences present in CHES are orphans? What is the probability that we are seeing here pervasive transcription? (b.t.w. pervasive transcription occurs at open chromatin region, therefore the ATAC-seq results could be reflecting just this). Are any of these novel genes found in more than one cell type? Are these novel genes spliced? Do they have gene signatures defined by Havana to call genes? I feel that in order to call these novel expressed regions as genes, more consistent evidence should be found regarding reproducibility, gene architecture, etc. If this is not possible, I would call these regions expressed loci, but not novel genes.

Reply:

Sorry for the confusion. The coding potential of 1,022 loci was predicted in our Immune Isoform Atlas, but they are annotated (predicted) as noncoding (lncRNAs) in GENCODE.

The high conservation score of the ORF regions within the novel genes in the Immune Isoform Atlas supports the possibility of truly coding transcripts. As the reviewer pointed out, if these transcripts were expressed in more than one cell type, we would have more confidence in their presence. Therefore, we looked at the number of cell types expressing them. Consequently, of total 2,062 transcripts from novel gene loci, we confirmed that 92% (1,891 transcripts) were expressed in multiple cell types in our long-read sequencing dataset. Another point raised is regarding the presence of splicing; all of transcripts from novel gene loci were 2 or more exons (this is because our FLAIR pipeline needed validation of splicing junction reads), leaving the coding potential of these transcripts. However, as commented by the reviewer, due to the lower conservation score compared to the aforementioned lncRNAs, we cannot dismiss the possibility that these transcripts may be non-functional without experimental validation. As a limitation, this

point was added in Results and we changed to call these “novel genes” as “potentially novel genes” in our manuscript.

Line 166; *In addition, 1,022 loci annotated as long non-coding RNAs (lncRNAs) in GENCODE were predicted as **potentially coding genes** by GeneMark-ST (Figure 2A). Although lncRNAs are defined as over 200 nucleotides in length and do not code for a peptide or protein²⁹, previous studies have shown that a fraction of putative small ORFs within lncRNAs are translated³⁰. Notably, the genomic regions of predicted ORFs of these lncRNAs were more conserved compared to those of the UTRs and introns (Figure 2D), supporting their coding potential.*

*Next, we examined 529 loci encoding novel predicted coding genes, which did not overlap any annotated genomic loci in GENCODE (Figure 2A). Notably, their predicted ORFs were highly conserved (Figure 2D). **Of these transcripts, 92% (1,891 transcripts) were expressed in multiple cell types and all of them had multi-exons.** We examined whether these transcripts were registered in the Comprehensive Human Expressed SequenceS (CHESS) database³¹. Most of the transcripts were found in CHESS, but the remaining 47 transcripts from mutually exclusive loci did not overlap any annotated genes in CHESS. For example, one potential novel transcript from chromosome 11 had a predicted 448 nucleotide ORF (Figure 2E), and notably, we found an open chromatin region from the TSS to intronic regions in subsets of monocytes and dendritic cells using datasets of the assay for transposase-accessible chromatin sequencing (ATAC-seq)³¹. This corresponded to our expression profile, in which it is only expressed in these subsets in the Immune Isoform Atlas (Figure 2F). Because the ORF and 3'-UTR of this isoform contained sequences derived from short interspersed elements (SINE)/MIR, the insertion of these TEs may have made it difficult to map the reads by short-read sequencing. **Further experimental verification is needed to confirm whether these potentially novel genes are truly encoding genes or just noise, such as pervasive transcription from open chromatin regions**³².*

In figure 4. Was the quantification obtained from long reads? Is this based on the long reads count or based on short reads? How is the TPM in supplementary table 5 calculated? Extended data Figure 3A claims that results were equivalent to Figure 4, but this is not evident when looking at the figure. Please explain.

Reply:

Expression in Figure 4 was quantified with long-read RNA sequencing data using FLAIR's quantify mode (*flair.py quantify*) with `--tpm` in option (source code; in https://github.com/juninamo/isoform_atlas/blob/main/code/flair.sh). TPM here is “transcripts per million”, so different from “Transcripts Per Kilobase Million” which are also normalized by length and generally used for short-read RNA sequencing data as discussed in their Github (<https://github.com/BrooksLabUCSC/flair/issues/163>).

In Extended data Figure 3A in the original submission, we used an independent short-read RNA sequencing dataset to validate whether cell-type specific isoforms expressed dominantly in

corresponding cell types and recapitulated concordant expression in the relevant immune cell subsets. To explain these clearly, we changed legends of Extended data Figure 3A as following;

Extended data Figure 5 (Extended data Figure 3A before revision). **The concordant expression of cell-type-specific isoforms in relevant cell types of short-read RNA-seq datasets.**

(A) Expression of cell-type-specific isoforms (**Figure 4B**) in cell subsets obtained from short-read RNA-seq datasets *as validation*¹⁹. *Columns indicate isoforms, and the colored bar at the top indicates the cell type that is specifically expressed in the long-read dataset. Rows indicate cell types in the short-read RNA-seq dataset. Column-wise Z scores of normalized counts are plotted. Cell types are arranged by hierarchical clustering.*

The analysis of translational efficiency was done by mapping existing RIBO-seq data to the long-read transcript models. How was mapping specificity controlled? Since this paper claims an average of 10 isoforms per gene, with most of them alternative donor and acceptor sites, it might be difficult to differentiate isoforms from any short-reads dataset. If reads are not specific for isoforms, conclusions of the properties of isoforms with high translational efficiency might be biased. The same holds for the IRAK1 gene, where more than 10 isoforms are present. How reliable is short-reads-based quantification of isoforms in a locus with so many isoforms? Also in Figure 5. I do not think that from the correlation plots shown in figure 5B, it can be concluded that there is a correlation between translational efficiency and Kozak score, ORF length, and 3'UTR length as R² values are very low. Although the p-value is significant, this is just the reflection of the high number of observations used in the analysis, but this is not enough to claim a relationship. The magnitude of the R-value must be considered and in these cases, the correlations are close to zero.

Reply:

Thank you for this important comment. First, regarding the multi-mapping issue for multiple isoforms, we took the same approach used in the previous papers^{28,29}; we used STAR with the following option to minimize ambiguous reads as possible when mapping ribo-seq reads or RNA-seq to Immune Isoform Atlas annotations (source code:

https://github.com/juninamo/isoform_atlas/blob/main/code/translational_efficiency.sh).

```
--outFilterMultimapNmax 8 \  
--alignSJoverhangMin 8 \  
--alignSJDBoverhangMin 1 \  
--sjdbScore 1 \  
--outFilterMismatchNmax 4 \  
--alignIntronMin 20 \  
--alignIntronMax 1000000 \  
--alignMatesGapMax 1000000
```

To examine the relationship between the number of isoforms per gene and the accuracy of quantification at transcript level, we generated a simulated short-read RNA sequencing dataset with assumption that all isoforms in our Atlas have equal expression level (~5 coverage per

transcript, equal FPKM) using polyester R package (<https://github.com/alyssafrazee/polyester>). We tested isoforms which are 13 or less related isoforms transcribed from the same loci because isoforms with more related isoforms are rare and reduce the reliability of the analysis. We then grouped the isoforms according to the number of isoforms expressed from the same gene region and compared the expression values. We tested 2 different mapping strategies, 1) mapping to genome as a standard way (STAR two-pass mode with default options), and 2) mapping to transcriptome as described above. Consequently, we observed that transcriptome-mapping strategy quantified more accurately (narrow and similar confidence interval of FPKM across groups of number of related-isoforms) than genome-mapping strategy, in the case of both paired-end and single-end RNA-seq.

#####

Performance of mapping to transcriptome (blue) and genome (red) using a simulated short-read RNA sequencing dataset (equal FPKM for all isoforms) using *polyester* R package. We tested cases of both paired-end and single-end by specifying *paired=TRUE* and *FALSE*, respectively, in *simulate_experiment* function.

#####

As the reviewer pointed out, it is still a challenging field to link translatoe with transcriptome as discussed in recent review paper³⁰. Translation efficiency is controlled by several factors, including uORF, secondary structure, miRNA, and codon optimality. While the correlation of the length of the 5'-UTR or 3'-UTR was not strong given their R2 values, it is still statistically

significant, suggesting their contribution to translation efficiency. In support of our results, the correlation between transcript features associated with the 5'-UTR and ORFs and translation efficiency is consistent with previous reports using other technique (TrIP-seq) or ribosome profiling³¹ (**Extended data Figure 6**). However, we observed a lack of agreement in the features associated with the 3'-UTR, which has been explained in our response to Reviewer 1's comment 7. We therefore think interpretation of our findings should be done more carefully, and we added this point in Discussion.

Line 504; *It should be noted that our analysis reveals a transcriptome-wide trend regarding the role of each transcript feature. However, experimental validation for individual transcripts will be necessary in the future.*

Extended data Figure 5 is inverted, and should be corrected.

Reply:

Thank you for detecting our mistake. We modified the figure.

Responses to the Comments by Reviewer #3:

In their paper, Immune Isoform Atlas: Landscape of alternative splicing in human immune cells, Inamo and colleagues use long-read sequencing to generate a full-length isoform annotation of 29 human immune cell subsets, which they name the Immune Isoform Atlas. The majority (88%) of the identified isoforms had no sequences registered in GENCODE (version 38). They characterized different properties of the unannotated isoforms and showed that repetitive elements significantly explained their diversity. In addition, they used the Immune Isoform Atlas as a reference annotation to reanalyze existing gene expression data and perform alternative splicing, QTL mapping, and colocalization analysis of QTLs and GWAS data. They identify a number of sQTL variants that were missed when using GENCODE. Overall, the manuscript is well written, the statistical analyses are sound, and the results support most of the conclusions and claims. However, additional evidence is needed to support the usefulness of the atlas for disease transcriptomics and genomics. I list a few major and minor comments below

Major

1. How do you envision this atlas being used? Would it complement or substitute GENCODE, e.g., for disease transcriptomics and QTL mapping studies? The majority of isoforms in GENCODE are not identified in the Immune Isoform Atlas and many sQTLs are only identified when using GENCODE annotations. Should both annotations be used?

Reply:

Thank you for this essential comment. We think our Immune Isoform Atlas is useful for accurately quantifying the full transcript length of samples from immune cell types (samples

using short-read RNA sequencing can be mapped to our annotation). In particular, we believe it is even more useful for samples obtained from individuals with immune cell-associated traits, such as autoimmune diseases, as shown in isoform-switch analysis of SLE and additional analysis of rheumatoid arthritis (please, see the results below). The reason why many sQTL were detected only when using GENCODE may be that GENCODE annotations were constructed by data using multiple tissues, including those not covered by our immune-cell samples. Another reason might be the complexity of Immune Isoform Atlas compared to GENCODE; the higher number of transcript isoforms in each gene might reduce the sensitivity of sQTL analysis for some isoforms as discussed in our previous study³². Therefore, as the reviewer suggested, we think both annotations should be used depending on the sample types or sample sizes.

2. The paper would benefit from a few biological replicates. For example, in line 135 you mention that: “Of note, 1,365 (40%) loci encoded so-called read-through isoforms ... Although read-through isoforms are known to be transcribed in particular situations in cells, such as in malignancy and infection many read-through isoforms ... were also transcribed in cells under normal physiological conditions.”. Could this be a consequence of profiling a single sample whose immune response might be triggered by an infection?

Reply:

The sample used in this study was confirmed to be free of underlying health status such as infection or malignancy at the time of enrollment. Further, we confirmed this control individual was not an outlier among the other healthy controls based on the genome-wide global expression pattern by mapping short-read RNA sequencing data, which were obtained by another ongoing project (Suzuki A et al).

PCA analysis using genome-wide gene expression profiling by mapping short-read RNA sequencing obtained from 74 healthy controls. Red dot represents the sample that was sequenced in the current study.

#####

Regarding biological replicate, we confirmed the presence of transcripts in our Atlas by using additional dataset; we obtained PBMC from a 40-year-old male and applied long-read RNA sequencing using the latest ONT platform (PromethION R10.4.1, V14 chemistry). This dataset allowed us to validate the presence of isoforms in Immune Isoform Atlas, owing to its higher quality of sequences, higher read depth, and comprehensive coverage of all the immune-subsets by PBMC. We mapped 26,551,929 reads from male PBMC with quality > Q15 and MAPQ >10 obtained by PromethION (R10.4.1, V14 chemistry). The same FLAIR pipeline as used in this study generated 100,607 isoform annotations (male PBMC annotation). We extracted isoforms in PBMC from the Immune Isoform Atlas (female PBMC annotation, 7,527 isoforms) and compared them with the male PBMC annotation using gffcompare¹. Notably, 85.0% of isoforms in female PBMC annotation were also discovered in the male PBMC annotation. Further, we validated 30.6% (n=48,757) of total isoforms in our Immune Isoform Atlas by male PBMC annotation (we added a label to these isoforms indicating that they have been validated; “isoform_info.txt.gz” in https://github.com/juninamo/isoform_atlas/tree/main). It should be mentioned that we sequenced into fine-grained 29 cell types so that we could detect cell-type specific transcripts as shown in Figure 4. These transcripts may be diluted in PBMC and could not be detected in the male PBMC sample.

Finally, we also validated expression of read-through transcripts. Because these transcripts might be artifacts of our cDNA-seq (e.g. template switching), we confirmed their existence using nanopore direct RNA-seq data obtained from the LCL (GM12878)¹⁷. Among the read-through loci (n=1,351) in our Immune Isoform Atlas, we validated that 437 loci including *TOMM40-APOE* locus are truly expressed in LCL. Further, our group experimentally examined the role of read-through transcript from *IFNAR2-IL10RB* locus and clarified its molecular function in the severity of COVID19²² (*Immunity*, under review).

#####

Extended data Figure 3. Validation of read-through isoforms using direct RNA sequencing¹⁷. Validated read-through isoforms transcribed from IFNAR2-IL10RB region (left) and TOMM40-APOE region (right). The read-through isoforms are highlighted by red.

#####

3. Line 266-270 and Figure 5B: these correlations are very low and seem to only be significant because of the large number of isoforms tested. Please report what correlation metric was used and how was the non-independence of isoforms accounted for (which would result in falsely significant p-values).

Reply:

Thank you for pointing out important points. This correlation coefficient was obtained by Spearman's correlation test. Since it was not mentioned, we added to figure legends. Translation efficiency is controlled by several factors, including uORF, secondary structure, miRNA, and codon optimality. While the R^2 values imply moderate association between the length of the 5'-UTR or 3'-UTR and translational efficiency (in this sense, we agree with the reviewer's comment that the correlations are low), the p-values indicate significant associations, indicating their contribution to translation efficiency. In support of our results, the correlation between transcript features associated with the 5'-UTR and ORFs and translation efficiency is consistent with previous reports using other technique for ribosome profiling³¹ (**Extended data Figure 6**) (the lack of agreement with features associated with the 3'-UTR is explained in our response to Reviewer 1).

If our understanding is correct about "non-independence of isoforms", the reviewer concerns multi-mapping of reads from ribo-seq to transcriptome. To minimize the effect of multi-mapping ribo-seq to isoforms transcribed from the same gene, we used STAR with the option to control multimapping (--outFilterMultimapNmax 8, which is more strict than default 10) as responded to the comment from Reviewer 2. Further, given that different isoforms express from the same gene exclusively, we calculated the translational efficiency only of isoforms that satisfied coverage depth > 1 of both ribo-seq and RNA-seq, which also avoid the potential over-estimating of translational efficiency due to low coverage. To examine the relationship between the number of isoforms per gene and the accuracy of quantification at transcript level, we generated a simulated short-read RNA sequencing dataset with assumption that all isoforms in our Atlas have equal expression level (~5 coverage per transcript, equal FPKM) using polyester R package (<https://github.com/alyssafrabee/polyester>). We tested isoforms which are 13 or less related isoforms transcribed from the same loci because isoforms with more related isoforms are rare and reduce the reliability of the analysis. We then grouped the isoforms according to the number of isoforms expressed from the same gene region and compared the expression values. We tested 2 different mapping strategies, 1) mapping to genome as a standard way (STAR two-pass mode with default options), and 2) mapping to transcriptome as described above. Consequently, we observed that transcriptome-mapping strategy quantified more accurately (narrow and similar confidence interval of FPKM across groups of number of related-isoforms) than genome-mapping strategy, in the case of both pair-end and single-end RNA-seq.

However, as the reviewer commented, the correlation coefficient was not strong, and we might only observe a transcriptome-wide trend that cannot necessarily be applied to individual isoforms. Therefore, we added this point in Discussion.

#####

Performance of mapping to transcriptome (blue) and genome (red) using a simulated short-read RNA sequencing dataset (equal FPKM for all isoforms) using polyester R package. We tested cases of both pair-end and single-end by specifying paired=TRUE and FALSE, respectively, in simulate_experiment function.

#####

Line 504; *It should be noted that our analysis reveals a transcriptome-wide trend regarding the role of each transcript feature. However, experimental validation for individual transcripts will be necessary in the future.*

4. The SLE data set used for differential splicing analysis is too small and likely underpowered. The results do not make a strong case for the usefulness of the atlas for differential splicing analyses versus standard annotations. The manuscript would benefit from analyzing a larger data set for SLE (e.g., <https://pubmed.ncbi.nlm.nih.gov/35389781/>) or other immune-related traits.

Reply:

Thank you for your helpful comment. Since the SLE paper the reviewer suggested used single-cell RNA sequencing data that did not cover the full-length transcripts, we could not add it to our isoform-switch analysis. For sample size, the current algorithm of isoform-switch analysis

requires a huge memory, especially when using the annotation containing a lot of isoforms like our atlas and short-read RNA-seq data with large sample size. Thus, to gain additional evidence for involvement of novel isoforms in the pathogenesis of immune-related diseases (IMDs) via isoform switch, we performed an additional analysis using RA dataset and found novel isoforms in *SIGLEC10* gene were differentially expressed in Temra CD8+ T cells from active RA and healthy controls subjects. We added this result to Result and Figure 6. Further, we conducted isoform-switch analysis for other IMDs and the results are available on the web app (<http://gfdweb.tmd.ac.jp:3838/>).

#####

Figure 6. Switched isoforms in IMDs.

(D) The structure of isoforms, which are transcribed from *SIGLEC10* locus and switched between RA (active) and healthy controls.

(E) The expression of *SIGLEC10* at the gene level.

(F) The isoform fractions in *SIGLEC10*.

#####

Line 386; *Rheumatoid arthritis (RA) is another example of autoimmune diseases characterized by chronic inflammation of the synovial*⁵⁸. We applied our atlas to a single immune subset, CD45RA-positive effector memory (Temra) CD8+ T cells (active RA = 9, healthy subjects = 9)^{59,60}. We found that three novel isoforms in *SIGLEC10* gene, which suppresses inflammatory responses to danger (damage)-associated molecular patterns by interacting with CD24⁶¹, were differentially expressed in Temra CD8+ T cells from active RA and healthy controls subjects (Figure 6D). Two of them (novel isoforms 2 and 3) were predicted to be sensitive to nonsense-mediated mRNA decay (NMD), which is a surveillance pathway to degrade RNA and prevent the production of abnormal⁶². At the gene level, *SIGLEC10* was more highly expressed in RA compared to healthy individuals (Figure 6E). However, at the isoform level, NMD-sensitive isoforms were dominant in RA (Figure 6F). This suggests that even though the expression was increased at the gene level, the relative expression of aberrant isoforms susceptible to NMD

was increased, resulting in a relatively low inflammatory suppressive function of SIGLEC10 in RA.

Minor

1. Line 48: add a citation for “Indeed, in the GTEx project, 23% of the GWAS loci were co-localized with sQTL.” I assume it is from this one <https://pubmed.ncbi.nlm.nih.gov/33499903/>

Reply:

Sorry for our unclear explanation. For this point, we cited this article (Science 36, 1318-1330 (2020)). We added this citation in the corresponding sentence.

2. Line 128 – 132: Only a small number (278 of 16,190) of potentially novel coding isoforms seems to be confirmed with the proteomic analyses. Is this expected? How does the validation rate look for known protein-coding isoforms?

Reply:

As the reviewer commented, the validation rate by proteomics analysis is not high, and this is concordant as in the previous paper of GTEx (7%, 2,575 /33,251), even when they analyzed only genes with high expression at mRNA levels⁸. Our long-read RNA sequencing data were obtained from primary cells and those of proteomics were from cell-lines (THP-1 cells and LCL), which may additionally explain the low concordance.

3. Line 189 and Figure 3D: Please add p-values for the statement: “Interestingly, the proportion of TEs was non-uniform; LTRs, which are autonomous and coding TEs, were enriched in TSSs and ORFs, while SINEs were enriched in 5'-UTRs and 3'-UTRs”.

Reply:

Thank you for your helpful comment. When investigating enrichment of each TE class at relative position across the gene-body in comparison with across the genome, we considered it inappropriate to evaluate them by either selecting arbitrarily a certain point or averaging of those regions given that UTR and ORF are contiguous regions. Therefore, we calculated the odds ratio and 95% confidence interval of enrichment at binned position across the gene-body by conditional maximum likelihood estimation using `oddsratio.fisher()` function from `epitools` R package. We then plotted the odds ratio (solid line, bottom of Figure 3D) and 95% confidence interval (shading, bottom of Figure 3D) to visualize TE enrichment through metagene level. Thus, we did not provide a single p-value for each gene region like TSS, UTR, and ORF. As shown in the line plot in bottom of Figure 3D, we can clearly see the enrichment of LTR around TSS and ORF (95% confidence interval of odds ratio is far from 1), while SINEs were enriched within 5'-UTRs and 3'-UTRs.

4. Line 241 and Figure 4D: “We comprehensively searched for RBP binding motifs and compared them between cell-type-specific isoforms and others.” What are “other” isoforms? Is the enrichment significant after multiple testing adjustments?

Reply:

Sorry for the confusion and thank you for the important point. “other” here means all isoforms that were not all cell-type-specific. To be clear, we changed “others” to “remaining non-specific isoforms” in the corresponding sentence in the manuscript.

For RBP analysis, of the total RBPs (396) listed in the original heatmap (Figure 4D), 316 RBPs are significant after correcting multiple testing (adjusted p-value < 0.05). We updated this heatmap in Figure 4D using only those with adjusted p-value < 0.05 and added this point in the corresponding sentence in the manuscript.

5. Line 317 – 319 and 936-938:

o How many genes were tested for changes in splicing patterns?

Reply:

We tested all 17,496 genes identified in our Immune Isoform Atlas.

o How many of the significant genes (84) were only discovered using the immune atlas and not the standard GENCODE annotation? It is currently unclear how useful the atlas is for identifying AS versus using GENCODE.

Reply:

Thank you for this important point. All significant 84 genes in isoform switch analysis using SLE dataset were known genes in the standard GENCODE annotation. However, 68 of 84 genes comprised unannotated isoforms (n=121) in GENCODE, which displayed significant upregulation or downregulation along with disease status. In addition, the GENCODE annotations were constructed based on the findings from short-read RNA-seq datasets, and they do not always accurately capture the full length of each transcript, including functional domains. This would bias the quantification of transcript abundance as well as the biological interpretation of analysis results; our Immune Isoform allows more accurate analysis of isoform changes. For example, we found a disease-associated isoform in *IRAK1* lacking Pkinase domain, which was unannotated in GENCODE. Further, using additional RA dataset, we identified three novel isoforms in *SIGLEC10* locus as describe above. Two of them were predicted to be sensitive to nonsense-mediated mRNA decay (NMD). Importantly, we found NMD-sensitive isoforms were dominant in RA (**Figure 6F**). This suggests that the aberrant isoform susceptible to NMD was increased in RA, resulting in an impaired inflammatory suppressive response via *SIGLEC10*. These results support the strength of our Atlas compared to GENCODE, especially in studying isoform changes in immune cells.

o Why are you using a .05 cutoff for dIF?

Reply:

We noticed that we used .1 as cutoff for dIF as the same as the default setting. We corrected the text.

o You introduce RA data in the Isoform-switch analysis method section but do not report any results.

Reply:

This was our mistake. We first intended to present the analysis using this RA dataset in this paper as well, but due to limited space, we had to omit it. However, as the reviewer pointed out, the analysis of SLE alone was insufficient. Therefore, we restored the RA dataset as described above.

Reference

1. Pertea, G. & Pertea, M. GFF Utilities: GffRead and GffCompare. *F1000Res.* **9**, (2020).
2. Castle, J. C. *et al.* Expression of 24,426 human alternative splicing events and predicted cis regulation in 48 tissues and cell lines. *Nat. Genet.* **40**, 1416–1425 (2008).
3. Wang, E. T. *et al.* Alternative isoform regulation in human tissue transcriptomes. *Nature* **456**, 470–476 (2008).
4. Yeo, G., Holste, D., Kreiman, G. & Burge, C. B. Variation in alternative splicing across human tissues. *Genome Biol.* **5**, R74 (2004).
5. Tapial, J. *et al.* An atlas of alternative splicing profiles and functional associations reveals new regulatory programs and genes that simultaneously express multiple major isoforms. *Genome Res.* **27**, 1759–1768 (2017).
6. Schmiedel, B. J. *et al.* Impact of Genetic Polymorphisms on Human Immune Cell Gene Expression. *Cell* **175**, 1701–1715.e16 (2018).
7. Li, Y. I. *et al.* Annotation-free quantification of RNA splicing using LeafCutter. *Nat. Genet.* **50**, 151–158 (2018).
8. Glinos, D. A. *et al.* Transcriptome variation in human tissues revealed by long-read sequencing. *Nature* **608**, 353–359 (2022).
9. Pardo-Palacios, F. *et al.* Systematic assessment of long-read RNA-seq methods for

- transcript identification and quantification. (2021) doi:10.21203/rs.3.rs-777702/v1.
10. Conesa, A. *et al.* A survey of best practices for RNA-seq data analysis. *Genome Biol.* **17**, 1–19 (2016).
 11. Lappalainen, T. *et al.* Transcriptome and genome sequencing uncovers functional variation in humans. *Nature* **501**, 506–511 (2013).
 12. Quach, H. *et al.* Genetic Adaptation and Neandertal Admixture Shaped the Immune System of Human Populations. *Cell* **167**, 643–656.e17 (2016).
 13. Lee, M. N. *et al.* Common genetic variants modulate pathogen-sensing responses in human dendritic cells. *Science* **343**, 1246980 (2014).
 14. Ishigaki, K. *et al.* Polygenic burdens on cell-specific pathways underlie the risk of rheumatoid arthritis. *Nat. Genet.* **49**, 1120–1125 (2017).
 15. Hu, W. *et al.* Systematic characterization of cancer transcriptome at transcript resolution. *Nat. Commun.* **13**, 6803 (2022).
 16. Ishigaki, K. *et al.* Multi-ancestry genome-wide association analyses identify novel genetic mechanisms in rheumatoid arthritis. *Nat. Genet.* **54**, 1640–1651 (2022).
 17. Workman, R. E. *et al.* Nanopore native RNA sequencing of a human poly(A) transcriptome. *Nat. Methods* **16**, 1297–1305 (2019).
 18. Tang, A. D. *et al.* Full-length transcript characterization of SF3B1 mutation in chronic lymphocytic leukemia reveals downregulation of retained introns. *Nat. Commun.* **11**, 1438 (2020).
 19. Smart, A. C. *et al.* Intron retention is a source of neoepitopes in cancer. *Nat. Biotechnol.* **36**, 1056–1058 (2018).
 20. Shiraishi, Y. *et al.* Systematic identification of intron retention associated variants from massive publicly available transcriptome sequencing data. *Nat. Commun.* **13**, 5357 (2022).
 21. Song, R. *et al.* Dynamic intron retention modulates gene expression in the monocytic differentiation pathway. *Immunology* **165**, 274–286 (2022).

22. Mitsui, Y. *et al.* CiDRE+ M2c macrophages hijacked by SARS-CoV-2 cause COVID-19 severity. *bioRxiv* 2022.09.30.510331 (2022) doi:10.1101/2022.09.30.510331.
23. Donovan, M. K. R., D'Antonio-Chronowska, A., D'Antonio, M. & Frazer, K. A. Cellular deconvolution of GTEx tissues powers discovery of disease and cell-type associated regulatory variants. *Nat. Commun.* **11**, 955 (2020).
24. Lim, Y. *et al.* Multiplexed functional genomic analysis of 5' untranslated region mutations across the spectrum of prostate cancer. *Nat. Commun.* **12**, 4217 (2021).
25. Chatterjee, S. & Pal, J. K. Role of 5'- and 3'-untranslated regions of mRNAs in human diseases. *Biol. Cell* **101**, 251–262 (2009).
26. Jia, L. *et al.* Decoding mRNA translatability and stability from the 5' UTR. *Nat. Struct. Mol. Biol.* **27**, 814–821 (2020).
27. Rothamel, K. *et al.* ELAVL1 primarily couples mRNA stability with the 3' UTRs of interferon-stimulated genes. *Cell Rep.* **35**, 109178 (2021).
28. Wang, H., McManus, J. & Kingsford, C. Isoform-level ribosome occupancy estimation guided by transcript abundance with Ribomap. *Bioinformatics* **32**, 1880–1882 (2016).
29. Reixachs-Solé, M., Ruiz-Orera, J., Albà, M. M. & Eyra, E. Ribosome profiling at isoform level reveals evolutionary conserved impacts of differential splicing on the proteome. *Nat. Commun.* **11**, 1–12 (2020).
30. Reixachs-Solé, M. & Eyra, E. Uncovering the impacts of alternative splicing on the proteome with current omics techniques. *Wiley Interdiscip. Rev. RNA* **13**, e1707 (2022).
31. Floor, S. N. & Doudna, J. A. Tunable protein synthesis by transcript isoforms in human cells. *Elife* **5**, (2016).
32. Yamaguchi, K. *et al.* Splicing QTL analysis focusing on coding sequences reveals mechanisms for disease susceptibility loci. *Nat. Commun.* **13**, 4659 (2022).

Reviewers' comments:

Reviewer #1 (Remarks to the Author):

I appreciate the authors' efforts to address the reviewers' comments, including adding data from a second donor. However, I am still concerned about the limited sample size (2 donors), and the value of the presented atlas to the research community.

Reviewer #2 (Remarks to the Author):

I thank the authors for their extensive review of their paper and for addressing the points of concern raised by all three reviewers. However, I still have some concerns about the way the results are presented.

One aspect of my review was the high number of novel transcripts of the NNC type and the little overlap with GENCODE annotation. Additionally, Reviewer 1 noted that the data comes from only one individual. The authors addressed this criticism by sequencing a PBMC sample from another subject and comparing it to the cell-type data. They concluded that most of the transcripts in PBMC were also found in the cell-type data, but only 30% of the cell-type data was recapitulated in the PBMC samples. They argue that this reflects the fact that specific cell types are diluted in the PBMC. While this might be true, the fact remains that the PBMC still fails to replicate most of the cell-type-specific transcripts. Additionally, the comparison to the IsoQuant analysis reveals very little overlap between the two transcript reconstruction methods. This discrepancy possibly reflects differences in the developers' understanding of what constitutes a true transcript, particularly evident in the differences in calling intron-retention transcripts. In summary, most of the transcripts presented here are novel transcripts that have been observed in only one biological sample and predicted by only one analysis tool. I consider this weak evidence to support calling these sequences a solid Atlas of immune cell-type transcripts. Having said that, the many transcript models reported here might still describe RNA molecules present in the sequenced sample and uncover a transcriptome diversity that is worth reporting. For instance, the LRGASP project validated many novel, single-sample transcripts that were detected by one or a few reads using PCR. The question is whether this fraction of the data should be included in a cell-type specific Atlas, which aims to be representative of the cell type. I suggest that only those transcript models found consistently (through replication or with a high number of associated long reads) should be considered for the Atlas, while other models found at low abundance and in only one sample should be regarded as part of a repository of sequences found in the designated cell types. This approach will ensure the robustness and comprehensiveness of the provided data.

I am still not convinced of the accuracy of transcript quantification by mapping short-read RNA-seq data in a dataset with an average of 10 transcripts per gene and high similarity among them, as shown in the different examples in the main and extended data figures. The additional analysis provided in the rebuttal does not truly indicate what constitutes good performance nor reflects the reality of transcripts having different expression levels. Therefore, it does not seem to strongly validate the ability of STAR to provide reliable quantification of such a complex transcriptome. Finally, I would encourage not to claim the statistical significance of the correlation between translation efficiency and Kozak score, ORF length, and 3' UTR length transcript features, as this is irrelevant given the number of data points and the small correlation value. Indicating that a slight trend is observed that agrees with the literature is acceptable.

Reviewer #3 (Remarks to the Author):

The authors have addressed all my concerns.

Point-to-point responses

Reviewer #1 (Remarks to the Author):

>>>

I appreciate the authors' efforts to address the reviewers' comments, including adding data from a second donor. However, I am still concerned about the limited sample size (2 donors), and the value of the presented atlas to the research community.

>>>

We appreciate for this comment. Because Reviewer 2 also expressed concerns about the robustness of our atlas and recommended the creation of a more reliable transcriptome model, we have made the decision to generate a new transcript atlas that includes only transcripts consistently identified in both the original dataset (29 immune cell types) and the validation dataset (PBMC). We will make this curated atlas publicly available (as linked below). Meanwhile, for readers who want more exploratory data, the current transcriptome model will still be provided (we changed its name to "Transcriptomic Resource of Immune Cells using Long-read Sequencing (TRAILS)" to emphasize the exploratory nature of the data). We think the data presented in Fig 2 and onwards that used TRAILS should not be substantially influenced by inter-individual variation of transcriptomes, because the inter-individual variation was much less than inter-cell-type variation as shown in our previous rebuttal. We also modified the title of our manuscript to "Alternative splicing in human immune cell subsets and disease genetics" to underscore the profound insights our findings offer into disease genetics, rather than placing undue emphasis on the data's atlas-like qualities. Meanwhile, we clearly indicated in the abstract that our data of 29 immune subsets was generate from an individual. Like the T2T Genome Project (*Science* 2022), there are many studies with scientific advances that analyzed a single individual, and we hope our analysis can serve as a touchstone for projects aiming to analyze inter-individual differences. We still believe that our findings would contribute to scientific advance in genome biology as well as disease genetics.

https://github.com/juninamo/TRAILS/Validated_TRAILS_by_PBMC.gtf.gz

<https://github.com/juninamo/TRAILS/TRAILS.gtf.gz>

Reviewer #2

>>>

I thank the authors for their extensive review of their paper and for addressing the points of concern raised by all three reviewers. However, I still have some concerns about the way the results are presented.

One aspect of my review was the high number of novel transcripts of the NNC type and the little overlap with GENCODE annotation. Additionally, Reviewer 1 noted that the data comes from only one individual. The authors addressed this criticism by sequencing a PBMC sample from another subject and comparing it to the cell-type data. They concluded that most of the transcripts in PBMC were also found in the cell-type data, but only 30% of the cell-type data was recapitulated in the PBMC samples. They argue that this reflects the fact that specific cell types are diluted in the PBMC. While this might be true, the fact remains that the PBMC still fails to replicate most of the cell-type-specific transcripts. Additionally, the comparison to the IsoQuant analysis reveals very little overlap between the two transcript reconstruction methods. This discrepancy possibly reflects differences in the developers' understanding of what constitutes a true transcript, particularly evident in the differences in calling intron-retention transcripts. In summary, most of the transcripts presented here are novel transcripts that have been observed in only one biological sample and predicted by only one analysis tool. I consider this weak evidence to support calling these sequences a solid Atlas of immune cell-type transcripts. Having said that, the many transcript models reported here might still describe RNA molecules present in the sequenced sample and uncover a transcriptome diversity that is worth reporting. For instance, the LRGASP project validated many novel, single-sample transcripts that were detected by one or a few reads using PCR. The question is whether this fraction of the data should be included in a cell-type specific Atlas, which aims to be representative of the cell type. I suggest that only those transcript models found consistently (through replication or with a high number of associated long reads) should be considered for the Atlas, while other models found at low abundance and in only one sample should be regarded as part of a repository of sequences found in the designated cell types. This approach will ensure the robustness and comprehensiveness of the provided data.

>>>

We thank for careful review of our revised paper and positive and critical comments.

1. Providing a robust transcript atlas in addition to the exploratory atlas

In response to the feedback from Reviewer 2, we have reconsidered the significance of dataset robustness. We acknowledge the concern raised about the absence of verification through alternative technical approaches. Therefore, we decided to generate a new transcript atlas that

exclusively comprises transcripts consistently identified in both the original dataset (29 immune cell types) and the validation dataset (PBMC). We will make this curated atlas publicly available as linked below. Meanwhile, for readers who want more exploratory data, the current transcriptome model will still be provided (Transcriptomic Resource of Immune Cells using Long-read Sequencing" (TRAILS)). The latter name change was made to respect the opinion of Reviewer 2 (and hopefully that of Reviewer 1) and to avoid the name using "atlas". Retaining this exploratory resource enables us to conduct subsequent analyses that have yielded significant findings, potentially laying the groundwork for further research endeavors.

https://github.com/juninamo/TRAILS/Validated_TRAILS_by_PBMC.gtf.gz

<https://github.com/juninamo/TRAILS/TRAILS.gtf.gz>

Line 107, Page 4;

We validated 85.0% (n= 6,399) of isoforms expressed in PBMC used in the TRAILS. We also validated 30.6% (n=48,757) of total isoforms in all cell types; of the isoforms validated by this independent data set, 29.6% were full splice match (FSM), 33.4% had a novel combination of known splice sites NIC, and 28.2% had novel splicing site NNC in comparison with GENCODE version 38 (hereafter GENCODE). We made them publicly available, considering both cases where users want more exploratory data (TRAILS) and where they want validated data (**Data and code availability**). From here, downstream analysis was performed using TRAILS to focus on cell type diversity in the landscape of transcriptome.

>>>

I am still not convinced of the accuracy of transcript quantification by mapping short-read RNA-seq data in a dataset with an average of 10 transcripts per gene and high similarity among them, as shown in the different examples in the main and extended data figures. The additional analysis provided in the rebuttal does not truly indicate what constitutes good performance nor reflects the reality of transcripts having different expression levels. Therefore, it does not seem to strongly validate the ability of STAR to provide reliable quantification of such a complex transcriptome.

>>>

2. Mapping accuracy of short-read RNA-seq data

We acknowledged the concerns regarding the accuracy of quantifying transcripts through the mapping of short-read RNA-seq data. As shown in our simulation data and as Reviewer 2 rightly pointed out, our validation isn't very robust. When we mapped simulated short-read RNA-seq data to

our database, we found little impact on the accuracy of quantification based on the number of transcripts from a given gene region. However, as Reviewer 2 commented, the significant variability, measured by the variance of FPKM, suggests caution is advisable when mapping short-read RNA-seq data. The current technologies, even qPCR, have substantial coefficient of variations (CV for qPCR is usually estimated to be 20~25%) that preclude precise quantification of a complex transcriptome. Therefore, we described in the main text the uncertainty of quantification using short-read RNA-seq datasets.

Line 602, Page 13;

When we mapped simulated short-read RNA-seq data to our annotation, the number of transcripts from a given gene region did not have a strong impact on the accuracy of quantification (**Extended data Figure 10**). However, the large variability, measured by variance of transcript abundance, suggests caution is advisable when mapping short-read RNA-seq data to TRAILS, and junction-based sQTL analysis would yield more reliable results.

Extended data Figure 10. Performance of mapping to TRAILS using a simulated paired-end short-read RNA sequencing dataset (equal FPKM for all isoforms).

>>>

Finally, I would encourage not to claim the statistical significance of the correlation between translation efficiency and Kozak score, ORF length, and 3'UTR length transcript features, as this is irrelevant given the number of data points and the small correlation value. Indicating that a slight trend is observed that agrees with the literature is acceptable.

>>>

3. Reconsidering our claims on translation efficiency

We agree with Reviewer 2's encouragement not to claim the statistical significance of the correlation between translation efficiency and transcript features. We are considering revising the section to simply mention that a subtle trend was observed, consistent with prior studies, without asserting statistical significance.

Line 345, Page 8;

We found a trend that agreed with the previously reported association between Kozak context scores and translation efficiency scores⁴⁵ (**Figure 5B, Extended data Figure 6A**). We also observed a slight association between translational efficiency and the lengths of the 5'- and 3'-UTRs was reversed.

Reviewer #3

>>>

The authors have addressed all my concerns.

>>>

We thank for the careful and constructive comments.

Reviewers' comments:

Reviewer #2 (Remarks to the Author):

I thank the authors for addressing my concerns and their willingness to reconsider some of the claims in the manuscript.

Reviewer #4 (Remarks to the Author):

Alternative splicing in human immune cells and disease genetics
by Inamo et al.

Inamo et al. present an in-depth analysis of the transcriptomes of 29 immune cell types using long-read sequencing. They report a large diversity of transcript isoforms (termed TRAILS), many of which are not present in GENCODE annotation. Interestingly, a large fraction of novel transcripts appears to arise from read-through transcripts, including a TOMM40_APOE fusion transcript that the authors validate by multiple orthogonal approaches. The novel transcripts also show more transposable element insertions, in line with the notion that these are expected to be captured better with long-read sequencing. Comparing between the 29 cell types, the authors find strong differences in the isoform patterns, suggesting a certain degree of cell type specificity. Finally, the authors use the TRAILS as a reference to map short-read RNA-seq data, including Ribo-seq data to analyse translation as well as patient data to investigate isoform switching and splicing quantitative trait loci (sQTLs) in human diseases. Overall, the manuscript is well written and accompanied by carefully designed figures.

Major points:

Identification of novel isoforms:

The authors report that a large number of the detected isoforms do not overlap with GENCODE annotation and are hence considered as novel. However, it appears that this comparison does only take into account protein-coding genes in GENCODE which may lead to an overestimation of novel isoforms. Indeed, a substantial fraction of non-GENCODE-overlapping genes from Figure 1C turns out to be GENCODE-annotated lncRNAs and pseudogenes in Figure 2A. I do not understand the rationale of taking these genes out from the GENCODE annotation in the first place. Also, it is unclear how overlap between TRAILS and GENCODE was defined with regard to complete and partial matches. E.g., according to Figure 2A, a major fraction of the 3448 "coding genes" from TRAILS that do not overlap with GENCODE annotation do correspond to read-through events, i.e., they do overlap with coding genes.

The authors use GeneMarkS-T to predict coding potential and propose that more than 90% of all detected isoforms are coding (145,523 out of 159,369). Moreover, it seems that in Figure 2A, all of the 3448 non-GENCODE-overlapping genes (Figure 1C) are predicted to be coding. The fraction of coding transcripts appears to be overly high. How does this compare to previous reports? How robust are these predictions when using alternative tools?

In this regard, it also seems that the section of novel isoforms mixes their novelty/annotation with their coding potential. E.g., as mentioned above, for assessing novelty, not just protein-coding but all genes should be taken into account. Similarly, the presence of ATAC-seq peaks (Figure 2E) provides evidence for a cell type-specific expression, but does not support the proposed coding potential of the given isoform. The authors should try to more clearly separate these two aspects.

Cell type specificity:

The authors report that many isoforms are expressed in a cell type-specific manner. While this observation is conceivable, I still wonder to which extent the presented analyses are influenced by differences in overall expression level rather than isoform expression. E.g., in Figure 4A, the authors show that isoform abundance allows to cluster the cell types based on lineages. Can a similar pattern be produced if only using genes with a minimum number of reads across cell types are used? Similarly, in the example of NLRP1 (Figure 4E,F), could it be that NLRP1 shows a considerably lower expression in neutrophils? Similarly, some of the isoform distributions for IL23R (Extended data Figure 5C-D) look almost discrete. Could it be that expression is hardly detected in some of the cell types?

Remapping of short-read RNA-seq data to TRAILS:

Similar to Reviewer #2, I am concerned about the accuracy of remapping short-read RNA-seq data to the TRAILS. The method section does not provide sufficient information on this part to evaluate how this was achieved. As an example, the authors state that "To calculate translational efficiency at the isoform level, trimmed reads were aligned to the de novo transcriptome sequences generated from long-read sequencing for 29 immune cell subsets using STAR as with tools developed for the same purpose" (ref 117, 118). Does this mean that the authors used the tool ORQAS (ref 117) for their analyses?

Minor comments:

Phylogenetic conservation: Why is conservation so much higher in predicted ORFs of lncRNAs compared to novel genes? Given that this analysis is focussed on exons, could it be that the distribution reflects conservation of splice sites rather than coding potential? Are there differences in exon lengths between the sets compared?

The term TRAILS appears in the first chapter of results without proper definition.

"At the gene level, we found at least one transcript isoform in the TRAILS for 51% of the genomic loci where coding transcripts are annotated in GENCODE"

I think that the % value is incorrect here as it refers to the Jaccard index, while the sentence is from the perspective of GENCODE loci. According the Venn diagram, this should be $7991/(7991+11926) = 59.9\%$.

"Comparing the percentage of transcriptional support level categories defined by GENCODE for the isoforms shared between GENCODE and the TRAILS, the most reliable category (all splice junctions are supported by at least one non-suspect mRNA) was the highest"

Check sentence

Figure 5A: Using point densities instead of transparency, which is quickly saturated, would allow to evaluate the consistency of the correlation across the data range.

Why would stabilization of mRNAs increase translational efficiency (which is normalized for transcript abundance)?

Dear Reviewers,

Reviewer #2

>>>

I thank the authors for addressing my concerns and their willingness to reconsider some of the claims in the manuscript.

>>>

We thank you for the careful and constructive comments.

Reviewer #4 (Remarks to the Author):

>>>

Inamo et al. present an in-depth analysis of the transcriptomes of 29 immune cell types using long-read sequencing. They report a large diversity of transcript isoforms (termed TRAILS), many of which are not present in GENCODE annotation. Interestingly, a large fraction of novel transcripts appears to arise from read-through transcripts, including a TOMM40_APOE fusion transcript that the authors validate by multiple orthogonal approaches. The novel transcripts also show more transposable element insertions, in line with the notion that these are expected to be captured better with long-read sequencing. Comparing between the 29 cell types, the authors find strong differences in the isoform patterns, suggesting a certain degree of cell type specificity. Finally, the authors use the TRAILS as a reference to map short-read RNA-seq data, including Ribo-seq data to analyse translation as well as patient data to investigate isoform switching and splicing quantitative trait loci (sQTLs) in human diseases. Overall, the manuscript is well written and accompanied by carefully designed figures.

>>>

Thank you for your positive remarks and appreciation of the significance of our study.

>>>

Major points:

Identification of novel isoforms:

The authors report that a large number of the detected isoforms do not overlap with GENCODE annotation and are hence considered as novel. However, it appears that this comparison does only take into account protein-coding genes in GENCODE which may lead to an overestimation of novel isoforms. Indeed, a substantial fraction of non-GENCODE-overlapping genes from Figure 1C turns out to be GENCODE-annotated lncRNAs and pseudogenes in Figure 2A. I do not understand the rationale of taking these genes out from the GENCODE annotation in the first place.

>>>

>

We are grateful for your insights and the opportunity to clarify aspects of our research.

Upon considering your valuable feedback, we compared 1) all GENCODE genes and all TRAILS genes (locus level comparison, regardless of coding or noncoding), and 2) all GENCODE predicted-coding isoforms and all TRAILS predicted-coding isoforms (isoform level comparison). Regarding the isoform level comparison, to minimize overestimation, we also predicted coding potential of all isoforms registered in GENCODE using GeneMarkST as same as TRAILS. Consequently, we revised the manuscript and Figure 1C accordingly. This entails refraining from labeling genes identified as

lncRNA/pseudogenes by GENCODE as 'novel'. Instead, the focus will now be on read-through transcripts and on potentially novel genes that do not overlap with gene regions already registered in GENCODE.

Line 115, Page 4;

At the gene loci level, we found 3,006 genomic loci in the TRAILS where transcripts are not annotated in GENCODE (**Figure 1C**). Further, we predicted coding potential of all isoforms in both TRAILS and GENCODE using GeneMark-ST²⁵ and compared the number of them. As a result, we found 129,708 isoforms with sequences not registered in GENCODE (**Figure 1C**).

>>>

Also, it is unclear how overlap between TRAILS and GENCODE was defined with regard to complete and partial matches. E.g., according to Figure 2A, a major fraction of the 3448 "coding genes" from TRAILS that do not overlap with GENCODE annotation do correspond to read-through events, i.e., they do overlap with coding genes.

>>>

>

Regarding the comparison of TRAILS and GENCODE, we employed the SQANTI3 pipeline, a widely used method in defining the overlap between de novo transcriptome annotation (TRAILS) and existing annotation (GENCODE). In SQANTI3, transcripts in de novo annotation are compared based on their splice sites and gene region as described in their tutorial. The Venn diagram in Figure 1C was described by complete match. As for the read-through transcripts, they are annotated as novel because they have new splice sites composed of two known genes. For the novel predicted coding genes highlighted in Figure 2, there is no overlap, even partial, with a gene locus registered in GENCODE.

The reason we focused on read-through transcripts as novel genes is to emphasize their significance in disease etiology, often overlooked in other research. Indeed, in a recently published paper, we demonstrated that the *TOMM40-APOE* transcript, intriguingly identified in our study, is located exclusively in mitochondria and has cell toxic ability⁹. Additionally, our recent publication in *Immunity* unveiled a read-through transcript from the *IFNAR2-IL10RB* region, implicating its role in severe COVID-19 cases⁸.

>>>

The authors use GeneMarkS-T to predict coding potential and propose that more than 90% of all detected isoforms are coding (145,523 out of 159,369). Moreover, it seems that in Figure 2A, all of the 3448 non-GENCODE-overlapping genes (Figure 1C) are predicted to be coding. The fraction of coding transcripts appears to be overly high. How does this compare to previous reports? How robust are these predictions when using alternative tools?

>>>

>

Thank you for your insightful queries regarding the coding potential of the detected isoforms. The robustness of GeneMarkS-T is well-documented in its original publication ⁶. This tool has demonstrated approximately 90% accuracy with a well-calibrated false positive rate (<15%), outperforming other alternative tools like TransDecoder. Furthermore, its accuracy and usefulness is supported by other studies including human samples using long-read sequencing technology ⁷. Consequently, we chose this tool for our analysis.

Regarding the high proportion of predicted-coding in our TRAILS, the original paper on SQUANTI¹⁰, utilizing GeneMark S-T, analyzed full-length cDNA sequenced via PacBio from mouse neural cells and oligodendrocyte cells. As a result, the majority of transcripts were predicted as coding. Specifically, by different classification based on splice junctions, FSM, ISM, NIC, and NNC contain predicted CDSs, with respective proportions of 97%, 90%, 87.8%, and 92.8%, which were comparable to our findings.

The reason why all of the 3,448 non-GENCODE-overlapping genes are predicted to be coding is that we filtered in only transcripts as predicted-coding in this analysis beforehand. It should be noted that the number has been changed since we compared all GENCODE genes including lncRNAs/pseudogenes in the revised manuscript as described above.

>>>

In this regard, it also seems that the section of novel isoforms mixes their novelty/annotation with their coding potential. E.g., as mentioned above, for assessing novelty, not just protein-coding but all genes should be taken into account. Similarly, the presence of ATAC-seq peaks (Figure 2E) provides evidence for a cell type-specific expression, but does not support the proposed coding potential of the given isoform. The authors should try to more clearly separate these two aspects.

>>>

>

Thank you for this helpful comment on an important issue. We totally agree that we should clearly separate the two aspects of transcripts (novelty and coding potential). As we responded to your first comment above, we compared 1) all GENCODE genes and all TRAILS genes (locus level comparison, regardless of coding or noncoding), and 2) all GENCODE predicted-coding isoforms and all TRAILS predicted-coding isoforms (isoform level comparison), which we believe would help to understand the two aspects of the transcripts. Consequently, we revised Fig 1C and the manuscript accordingly.

>>>

Cell type specificity:

The authors report that many isoforms are expressed in a cell type-specific manner. While this observation is conceivable, I still wonder to which extent the presented analyses are influenced by differences in overall expression level rather than isoform expression. E.g., in Figure 4A, the authors

show that isoform abundance allows to cluster the cell types based on lineages. Can a similar pattern be produced if only using genes with a minimum number of reads across cell types are used? Similarly, in the example of *NLRP1* (Figure 4E,F), could it be that *NLRP1* shows a considerably lower expression in neutrophils? Similarly, some of the isoform distributions for *IL23R* (Extended data Figure 5C-D) look almost discrete. Could it be that expression is hardly detected in some of the cell types?

>>>

>

We appreciate your critical observation on the influence of overall expression levels versus isoform expression.

In Figure 4A, the isoform abundance for cell type clustering is calculated using the expression ratios, not read counts, which we consider theoretically eliminates bias due to overall gene expression levels.

In terms of cell type-specific isoforms such as *NLRP1* (Figure 4B,E,F), we have included in our methods a threshold for expression ratios (>0.2) and a corrected read count number (Reads Per Million, RPM > 2), ensuring that we eliminate bias from transcripts with lower overall gene expression.

In terms of *IL23R*, as you pointed out, some cell types, such as naive B cells, did not express specific isoform (*IL23R*-5 isoform) for Th17 cells.

Isoform expression (Reads Per Million) by cell types in *IL23R* locus.

>>>

Remapping of short-read RNA-seq data to TRAILS:

Similar to Reviewer #2, I am concerned about the accuracy of remapping short-read RNA-seq data to the TRAILS. The method section does not provide sufficient information on this part to evaluate how this was achieved. As an example, the authors state that "To calculate translational efficiency at the isoform level, trimmed reads were aligned to the de novo transcriptome sequences generated from long-read sequencing for 29 immune cell subsets using STAR as with tools developed for the same purpose" (ref 117, 118). Does this mean that the authors used the tool ORQAS (ref 117) for their analyses?

>>>

>

Regarding the remapping of short-read RNA-seq data to TRAILS, we acknowledge the importance of accuracy in identifying alternative splicing events. We have optimized the STAR parameters for transcript quantification with referring to Ribomap¹¹ and ORQAS¹², tools designed for transcript-level quantification with short-read data, as following;

```
--outFilterMultimapNmax 8 \  
--alignSJoverhangMin 8 \  
--alignSJDBoverhangMin 1 \  
--sjdbScore 1 \  
--outFilterMismatchNmax 4 \  
--alignIntronMin 20 \  
--alignIntronMax 1000000 \  
--alignMatesGapMax 1000000
```

We did not use Ribomap or ORQAS pipelines directly because these tools were developed for different purposes. As reported in the original paper of Ribomap, the method was tested using synthetic ribo-seq reads with known profiles, demonstrating strong performance across various sequencing error rates and a strong Pearson correlation between footprint assignments and actual ribosome profiles. We demonstrated that the variance of the quantification by the number of isoforms within particular loci were minimized using these parameters in the simulated dataset as shown in Extended data Fig 10. To clarify this point, the following statement was added to the revised manuscript.

Line 1068, Page 45;

To calculate translational efficiency at the isoform level, trimmed reads were aligned to the *de novo* transcriptome sequences generated from long-read sequencing for 29 immune cell subsets using STAR⁷⁸ as with tools developed for the same purpose^{117,118}, **designed for transcript-level quantification with short-read data. We used the same STAR parameters as these tools to optimize for transcript quantification. As reported in the original paper^{117,118}, the method was tested using synthetic ribo-seq reads with known profiles, demonstrating strong performance across various sequencing error rates and a strong Pearson correlation between footprint assignments and actual ribosome profiles.**

>>>

Minor comments:

Phylogenetic conservation: Why is conservation so much higher in predicted ORFs of lncRNAs compared to novel genes? Given that this analysis is focussed on exons, could it be that the distribution reflects conservation of splice sites rather than coding potential? Are there differences in exon lengths between the sets compared?

>>>

>

We appreciate your important suggestion on the conservation score by different types of novel genes. Given the higher conservation scores of lncRNAs compared to novel genes, we initially expected that lncRNAs would have longer exons, but in fact lncRNAs had shorter results when it came to ORF. We believe the reason for this is that some of the GENCODE lncRNAs already contain functional transcripts³, and these give them a higher conservation score than the as yet unproven novel genes group identified in our TRAILS.

region	median of exon length (nt)		p-value
	lncRNAs	novel genes	
5'UTR	171.5	122.5	0.21
ORF	115	170	0.004
3'UTR	397	311.5	0.19

Table of exon length by regions.

The term TRAILS appears in the first chapter of results without proper definition.

>Thank you for pointing this out. We modified our manuscript.

>>>

"At the gene level, we found at least one transcript isoform in the TRAILS for 51% of the genomic loci where coding transcripts are annotated in GENCODE"

I think that the % value is incorrect here as it refers to the Jaccard index, while the sentence is from the perspective of GENCODE loci. According the Venn diagram, this should be $7991/(7991+11926) = 59.9\%$.

>>>

>Thank you for pointing this out. We modified our manuscript.

>>>

"Comparing the percentage of transcriptional support level categories defined by GENCODE for the isoforms shared between GENCODE and the TRAILS, the most reliable category (all splice junctions are supported by at least one non-suspect mRNA) was the highest"

Check sentence

>>>

>Thank you for pointing this out. To make it clear, we modified our manuscript.

Line 121, Page 4;

A comparison of the percentage of transcriptional support level categories for isoforms common to GENCODE and TRAILS showed that the highest percentage was in the most reliable category (all splice junctions are supported by at least one non-suspect mRNA) defined by GENCODE (**Extended data Figure 1E**).

>>>

Figure 5A: Using point densities instead of transparency, which is quickly saturated, would allow to evaluate the consistency of the correlation across the data range.

>>>

>

Thank you for pointing this out. We modified the figure.

Figure 5A. Scatter plot of isoforms with normalized read counts of RNA-seq and Ribo-seq. Each dot represents a particular isoform.

>>>

Why would stabilization of mRNAs increase translational efficiency (which is normalized for transcript abundance)?

>>>

>

It's important to note that translational efficiency is influenced by a complex interplay of factors including mRNA stability, the availability of translation machinery and factors, the sequence elements within the mRNA that affect translation, and the overall cellular context. Therefore, mRNA stabilization is just one of many factors that can affect translational efficiency. For specific mechanisms and examples, it is best to consult the articles mentioned and other specialized literature on the topic ¹⁴. For example,

1. **Increased Availability for Translation:** More stable mRNAs have a longer half-life in the cell, which means they are available for translation over a more extended period. This increased window of opportunity allows for more rounds of translation before the mRNA is degraded.
2. **Efficient Ribosome Recruitment:** Stabilized mRNAs may be more likely to have a structure that is conducive to ribosome binding. Ribosomes are the cellular machinery responsible for protein synthesis, and their efficient recruitment is critical for high translational efficiency.

References

1. Sun, B. B. *et al.* Genetic associations of protein-coding variants in human disease. *Nature* **603**, 95–102 (2022).
2. Takata, A. *et al.* Comprehensive analysis of coding variants highlights genetic complexity in developmental and epileptic encephalopathy. *Nat. Commun.* **10**, 2506 (2019).
3. Mattick, J. S. *et al.* Long non-coding RNAs: definitions, functions, challenges and recommendations. *Nat. Rev. Mol. Cell Biol.* **24**, 430–447 (2023).
4. Nemeth, K., Bayraktar, R., Ferracin, M. & Calin, G. A. Non-coding RNAs in disease: from mechanisms to therapeutics. *Nat. Rev. Genet.* (2023) doi:10.1038/s41576-023-00662-1.
5. Harrow, J. *et al.* GENCODE: the reference human genome annotation for The ENCODE Project. *Genome Res.* **22**, 1760–1774 (2012).
6. Tang, S., Lomsadze, A. & Borodovsky, M. Identification of protein coding regions in RNA transcripts. *Nucleic Acids Res.* **43**, e78 (2015).
7. Huang, K. K. *et al.* Long-read transcriptome sequencing reveals abundant promoter diversity in distinct molecular subtypes of gastric cancer. *Genome Biol.* **22**, 44 (2021).
8. Mitsui, Y. *et al.* Expression of the readthrough transcript CiDRE in alveolar macrophages boosts SARS-CoV-2 susceptibility and promotes COVID-19 severity. *Immunity* **56**, 1939–1954.e12 (2023).
9. Chang, S. *et al.* Uncovering the Localization and Function of a Novel Read-Through Transcript ‘TOMM40-APOE’. *Cells* **13**, 69 (2023).
10. Tardaguila, M. *et al.* SQANTI: extensive characterization of long-read transcript sequences for quality control in full-length transcriptome identification and quantification. *Genome Res.* **28**, 396–411 (2018).
11. Wang, H., McManus, J. & Kingsford, C. Isoform-level ribosome occupancy estimation guided by transcript abundance with Ribomap. *Bioinformatics* **32**, 1880–1882 (2016).
12. Reixachs-Solé, M., Ruiz-Orera, J., Albà, M. M. & Eyras, E. Ribosome profiling at isoform level reveals evolutionary conserved impacts of differential splicing on the proteome. *Nat. Commun.* **11**, 1768 (2020).
13. Dobin, A. *et al.* STAR: ultrafast universal RNA-seq aligner. *Bioinformatics* **29**, 15–21 (2013).
14. Wu, Q. & Bazzini, A. A. Translation and mRNA Stability Control. *Annu. Rev. Biochem.* **92**, 227–245 (2023).

REVIEWERS' COMMENTS

Reviewer #4 (Remarks to the Author):

Alternative splicing in human immune cells and disease genetics
by Inamo et al.

Inamo et al. present an in-depth analysis of the transcriptomes of 29 immune cell types using long-read sequencing, termed TRAILS.

In their revised version, the authors refined the definition of novel isoforms and now separately compare all detected transcripts and coding-transcripts. They also addressed all other concerns with this regard.

Regarding the observed cell type specificity, the authors answered most questions in the response but hardly changed the manuscript. Some of these considerations should also be mentioned in the manuscript. While I agree that using expression ratios rather than read counts should generally avoid biases from overall expression levels, the reliability of these measures is still expected to go down for genes with very low read counts.

Dear Reviewer,

Reviewer #4

>>>

Alternative splicing in human immune cells and disease genetics
by Inamo et al.

Inamo et al. present an in-depth analysis of the transcriptomes of 29 immune cell types using long-read sequencing, termed TRAILS.

In their revised version, the authors refined the definition of novel isoforms and now separately compare all detected transcripts and coding-transcripts. They also addressed all other concerns with this regard.

Regarding the observed cell type specificity, the authors answered most questions in the response but hardly changed the manuscript. Some of these considerations should also be mentioned in the manuscript. While I agree that using expression ratios rather than read counts should generally avoid biases from overall expression levels, the reliability of these measures is still expected to go down for genes with very low read counts

.>>>

We thank you for the constructive comments.

After considering your valuable feedback, we have added these points (1. concern about the reliability of low-expressed genes, and 2. Specific example for *IL23R*) in the Discussion section of the revised manuscript to acknowledge the reliability of very low-expressed genes.

Line 563, Page 10;

Regarding the observed cell type specificity, we primarily utilized isoform ratios rather than raw read counts to minimize biases associated with overall gene expression levels. However, it is important to acknowledge that the reliability of the isoform ratio may decrease for genes with very low read counts. In fact, for *IL23R*, which showed cell type-specific isoform expression in Th17 and Treg cells, we observed that certain T cell subsets, such as Tfh cells, did not have detectable expression of the gene in our dataset. Considering our previous research identifying associations between low-expressed isoforms and diseases¹¹, there is potential to uncover new insights by sequencing more reads and/or increasing the sample size, which could lead to the identification of additional cell type-specific isoforms and their relevance to disease mechanisms.